# True Knowledge Comes from Practice: Aligning LLMs with Embodied Environments via Reinforcement Learning

**Weihao Tan[1], Wentao Zhang[1], Shanqi Liu[2], Longtao Zheng[1], Xinrun Wang[1],\* Bo An[1, 3\*]**

[1] Nanyang Technological University, Singapore [2] Zhejiang University [3] Skywork AI
{weihao001, wt.zhang, longtao001, xinrun.wang, boan}@ntu.edu.sg
shanqiliu@zju.edu.cn

## Abstract

Despite the impressive performance across numerous tasks, large language models (LLMs) often fail in solving simple decision-making tasks due to the misalignment of the knowledge in LLMs with environments. On the contrary, reinforcement learning (RL) agents learn policies from scratch, which makes them always align with environments but difficult to incorporate prior knowledge for efficient explorations. To narrow the gap, we propose TWOSOME, a novel general online framework that deploys LLMs as decision-making agents to efficiently interact and align with embodied environments via RL without requiring any prepared datasets or prior knowledge of the environments. Firstly, we query the joint probabilities of each valid action with LLMs to form behavior policies. Then, to enhance the stability and robustness of the policies, we propose two normalization methods and summarize four prompt design principles. Finally, we design a novel parameter-efficient training architecture where the actor and critic share one frozen LLM equipped with low-rank adapters (LoRA) updated by PPO. We conduct extensive experiments to evaluate TWOSOME. i) TWOSOME exhibits significantly better sample efficiency and performance compared to the conventional RL method, PPO, and prompt tuning method, SayCan, in both classical decision-making environment, Overcooked, and simulated household environment, VirtualHome. ii) Benefiting from LLMs' open-vocabulary feature, TWOSOME shows superior generalization ability to unseen tasks. iii) Under our framework, there is no significant loss of the LLMs' original ability during online PPO finetuning. [1]

## 1 Introduction

LLMs have demonstrated remarkable success in natural language generation and understanding (Brown et al., 2020; OpenAI, 2023). Recent studies show that LLMs can manage other AI models and tools to address complex multimodal tasks (Shen et al., 2023; Lu et al., 2023), assist or play sophisticated games, such as TextWorld (Yao et al., 2023), Handbi (Hu & Sadigh, 2023), and MineCraft (Wang et al., 2023a), or be deployed on robots for real-world interactions (Brohan et al., 2023; Driess et al., 2023). While LLMs can provide insightful suggestions for complex tasks, they often fail in solving simple decision-making tasks due to misalignment issues (Brohan et al., 2023).

There are two main misalignment issues leading to the failures of LLMs in decision-making

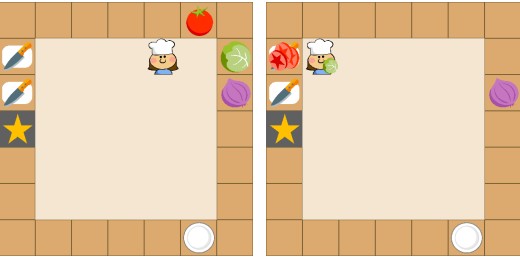

(a) Only tomato, lettuce and onion are provided in the game. LLMs may choose to pick up additional ingredients, such as cucumber and pepper to cook the dish.

(b) LLMs guide the agent to put the lettuce on the cutting board, which already contains tomato, without knowing that each cutting board can only contain one item at a time.

Figure 1: Two misalignment examples

---

\*Co-corresponding Authors
[1]Code is available at https://github.com/WeihaoTan/TWOSOME.

tasks. i) LLMs may generate invalid actions. As the example shown in Figure 1a, LLMs may keep adding cucumber and pepper when asked to make a tomato and lettuce salad, while these ingredients are not provided. ii) LLMs may not know accurately the dynamic transitions of the environments, especially when some specific constraints are introduced in the environments. This incorrect estimation will make LLMs tend to choose actions that fit their learned common sense, resulting in the failure to solve domain-specific tasks. As the example shown in Figure 1b, LLMs may keep trying to put both tomato and lettuce on the same cutting board, without knowing that only one item can be placed on the board in this environment. Addressing these two misalignment issues requires careful alignment between LLMs and environments.

On the contrary, reinforcement learning (RL) learns agents' policies from scratch through trial and error in environments (Sutton & Barto, 2018), which ensures that RL agents are well aligned with environments. Most RL methods start from random policies, updated according to the return from the environments, which leads to poor sample efficiency as most policies have poor performance at the early stage of learning. One way to improve the sample efficiency is to incorporate the prior knowledge with the initialization of the policy and the exploration during training (Kumar et al., 2023). LLMs are ideal sources of prior knowledge for RL agents as LLMs are trained with enormous data from the corpus. Therefore, by leveraging RL to align LLMs with embodied environments to solve decision-making tasks, we can address the misalignment issues in LLMs and the sample efficiency issue in RL simultaneously.

Motivated by this idea, we propose **T**rue kno**W**ledge c**O**me**S** fr**OM** practic**E** (TWOSOME), a general online framework that deploys LLMs as embodied agents to efficiently interact and align with environments via RL to solve decision-making tasks without requiring any prepared datasets or prior knowledge of the environments. Instead of letting LLMs directly generate actions, we use the loglikelihood scores of each token provided by LLMs to calculate the joint probabilities of each action and form valid behavior policies. This process eliminates the misalignment caused by invalid actions. Moreover, the LLM agents are optimized with proximal policy optimization (PPO) (Schulman et al., 2017) using rewards from the environments, which eliminates the misalignment caused by dynamic transitions. We observe that the formed behavior policies suffer a severe issue that longer actions tend to have lower joint probabilities, resulting in an unreasonable unbalance over the action distribution. To overcome this issue, we propose token normalization and word normalization in terms of the number of tokens and words in actions to rectify the unbalance. Furthermore, we design a novel architecture for efficient training, where both the actor and critic in RL methods share the same frozen LLaMA-7B model (Touvron et al., 2023), updated by parameter efficient finetuning methods, e.g., LoRA (Hu et al., 2022). During training, we observe that prompts for observations and actions can greatly influence the initial policies of LLM agents, therefore, we also summarize four principles for designing efficient prompts to enhance the reasoning ability of LLMs.

We first evaluate our methods on a classical RL decision-making environment, Overcooked, and a simulated physical environment, VirtualHome, with various tasks to show that our proposed word normalization method can remarkably improve stability and accelerate convergence during the training process. TWOSOME with word normalization exhibits significantly better sample efficiency and performance compared to the conventional RL method, PPO, and prompt tuning method, SayCan, in all tasks. Then, we test our trained TWOSOME agents in eight new unseen tasks and find that TWOSOME has superior generalization ability across unseen tasks. Finally, we evaluate our trained TWOSOME agents on traditional NLP benchmarks to demonstrate that under our framework, there is no significant loss of the LLMs' original ability during online PPO fine-tuning.

## 2 RELATED WORK

In this section, we present a brief overview of related work. More discussions are in Appendix A.

**Embodied Agents with LLMs.** Recent methods use LLMs to assist planning and reasoning in robot learning (Brohan et al., 2023; Liang et al., 2022; Zeng et al., 2022) and simulation environments (Fan et al., 2022; Wang et al., 2023a; Yao et al., 2023). LLMs are also applied to help robot navigation (Parisi et al., 2022; Majumdar et al., 2020) and manipulation (Jiang et al., 2022; Ren et al., 2023; Khandelwal et al., 2022). Among them, ReAct (Yao et al., 2023) uses chain-of-thought prompting by generating both reasoning traces and action plans with LLMs. SayCan (Brohan et al., 2023) leverages the ability of LLMs to understand human instructions to make plans for completing

tasks without finetuning LLMs. Voyager (Wang et al., 2023a) leverages GPT-4 to learn and continually discover skills during learning. While these works exhibit promising results, they rely too heavily on the inherent capabilities of powerful LLMs, which are difficult to apply to smaller LLMs with weaker reasoning abilities. Concurrent to our work, GLAM (Carta et al., 2023) uses RL to ground LLMs. However, they focus on simple primitive actions in toy environments without rich semantics, resulting in underutilizing the capabilities of LLMs, and failing to observe the impact of prompt design and address the unbalance over action space.

**Finetuning LLMs.** Parameter-efficient finetuning (PEFT) can significantly reduce the number of parameters to tune LLMs while with minor loss of the performance (Ding et al., 2023). *Prompt tuning* (Lester et al., 2021) and *prefix tuning* (Li & Liang, 2021) prepend additional tunable tokens to the input or hidden layers, and *adapter tuning* (Houlsby et al., 2019) introduces the adapter to LLMs, where only the introduced tokens and adapters are finetuned. Recently, low-rank adaption (LoRA) (Hu et al., 2022) introduces low-rank matrices to approximate parameter updates. Reinforcement learning from human feedback (RLHF) is effective in finetuning LLMs (Ouyang et al., 2022), where the reward for RL is learned from human feedback. Another concurrent work (Xiang et al., 2023) leverages embodied environments to provide feedback to finetune LLMs. Different from our method, they use supervised learning to finetune LLMs with pre-collected embodied experiences by random sampling in environments instead of doing decision-making tasks from scratch.

## 3 PRELIMINARIES

In this section, we provide a brief background on LLM and the RL problem formulation.

**LLMs** learn from text data using unsupervised learning. LLMs optimize the joint probabilities of variable-length symbol sequences as the product of conditional probabilities by $P(x) = \prod_{i=1}^{n} P(s_i|s_1, ..., s_{i-1})$, where $(s_1, s_2, ..., s_n)$ is variable length sequence of symbols.

**LoRA** is a parameter- and compute-efficient finetuning method that incorporates trainable rank decomposition matrices into each layer of an LLM. It allows indirect training of dense layers with weight matrix, $W_0 \in \mathbb{R}^{d \times k}$ by optimizing the rank-decomposition matrices by $W_0 + \Delta W = W_0 + BA$, where $B \in \mathbb{R}^{d \times r}$, $A \in \mathbb{R}^{r \times k}$, and the rank $r$ is much smaller than $d$ and $k$.

**RL** formulates decision-making problems as Markov Decision Processes (MDPs). An MDP is defined by the tuple $(S, A, T, R, \gamma)$, where $S$ is the state space, $A$ is the action space, $T$ is the transition dynamics, $R$ is the reward function and $\gamma$ is the discount factor. Agents select actions based on observations, aiming to maximize the expected discounted accumulative reward.

**PPO** is a state-of-the-art actor-critic RL method that optimizes the policy based on the accumulative reward with advantage function $A(s_t, a_t) = Q(s_t, a_t) - V(s_t)$, where $V(s_t)$ is the value function and $Q(s_t, a_t)$ is the action-value function. The objective of PPO can be expressed as:

$$J(\theta) = \mathbb{E}_{s,a}[\min(\frac{\pi_\theta(a|s)}{\pi_{\theta_{old}}(a|s)} A_{\pi_{\theta_{old}}}(s,a), \text{clip}(\frac{\pi_\theta(a \mid s)}{\pi_{\theta_{old}}(a \mid s)}, 1 \pm \epsilon) A_{\pi_{\theta_{old}}}(s,a))]$$

where $A_{\pi_{\theta_{old}}}$ represents the advantage function of the policy before updating, and the clip operator controls the size of policy updates.

**Aligning LLMs with Embodied Environments.** We intend to deploy LLMs as interactive agents in embodied environments, where LLMs receive textual observations and generate textual actions executed in the environments. The textual observations and actions can be transferred from images or vectors by vision-language modes (VLMs) or scripts. Compared to primitive actions, high-level actions, such as macro-actions (Theocharous & Kaelbling, 2003) and learned skills (Konidaris et al., 2011) usually have richer semantics, benefiting LLMs to leverage their prior knowledge.

## 4 TWOSOME

In this section, we present TWOSOME, a general online framework to align LLMs with embodied environments via RL. We first describe how to deploy LLMs as embodied agents to generate *valid* actions. Secondly, we investigate the influence of lengths of action prompts on the generated policies and propose two normalization techniques, i.e., token normalization and word normalization, to al-

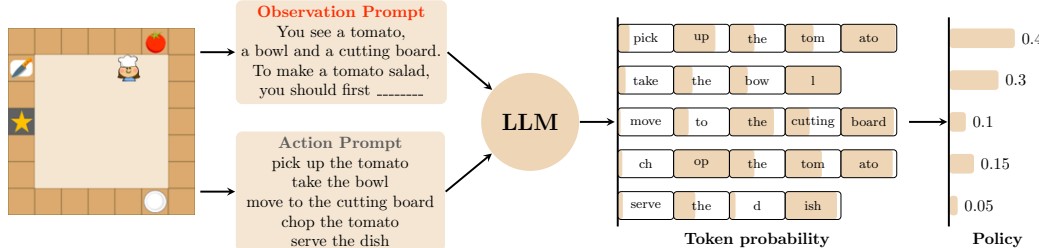

Figure 2: Overview of how TWOSOME generates a policy using joint probabilities of actions. The color areas in the token blocks indicate the probabilities of the corresponding token in the actions.

leviate the unbalance over actions. Then, we design a parameter-efficient training method under the PPO framework. Finally, we summarize four principles for efficient prompt design for TWOSOME.

## 4.1 VALID POLICY GENERATION

Actions directly generated by LLMs could be invalid in environments. This issue can be partially resolved by prompt design, i.e., adding the restrictions of actions in the prompt. However, most of the current LLMs cannot exactly follow the restrictions, especially for small and medium LLMs, i.e., less than 65B. Therefore, novel methods are required to let LLMs generate valid policies.

Instead of letting LLMs generate actions directly, TWOSOME queries the scores of all available actions from LLMs. These scores are used to determine the probabilities of executing the actions, which is similar to SayCan (Brohan et al., 2023). The process of the generation of policy is illustrated in Figure 2. Specifically, we associate a unique semantic prompt, e.g., *pick up the tomato*, to each action in the embodied environments, named *action prompt*, and *observation prompt* for the raw observation. The action prompt $a_k \in \mathcal{A}$ is a sequence of tokens $a_k = \{w_k^1, \ldots, w_k^{N_k}\}$ where $N_k$ is the length of the action prompt $k$. We note that the action prompts are not necessarily of the same length. At each step, the observation prompt $s \in \mathcal{S}$ is concatenated with each of the valid action prompts $a_k$, which forms the input of the LLM. We use the probabilities of tokens generated by the LLMs to compute the probability of the action prompt. Note that this probability differs from the probability of executing the action in the environment. For convenience, we call this probability token-level probability. The token-level probability of $a_k$ is

$$P_{token}(a_k|s) = P(w_k^1, \ldots, w_k^{N_k}|s) = \prod_{i=1}^{N_k} P(w_k^i|s, w_k^1, \ldots, w_k^{i-1}) \tag{1}$$

Normally, scores provided by LLMs are the loglikelihood of each token, namely the logits, $\log P_{token}$. We use softmax to normalize token-level probabilities over actions to get the policy:

$$P(a_k|s) = \frac{\exp(\log P_{token}(a_k|s))}{\sum_{a \in \mathcal{A}} \exp(\log P_{token}(a|s))} \tag{2}$$

There are two main advantages of this method: i) the generated policy is always valid for execution in the embodied environments and ii) leveraging the compositionability, our method can be applied to enormous actions representable by the vocabulary.

## 4.2 ACTION PROMPT NORMALIZATION

In this subsection, we will illustrate the issue in the method presented in the above section and present two normalization techniques to alleviate them.

**Issue of Eq. (1).** One key issue in Eq. (1) is that the probability of each token $P(w_k^i|\cdot)$ is always less than 1. Therefore, longer action prompts tend to have lower token-level probabilities, even though the longer action prompts may be more reasonable in the environment. For example, in Figure 2, *pick up the tomato* has more tokens than *serve the dish* but is the optimal action in terms of the given observation. A simple remedy for this issue is forcing all action prompts to have similar lengths, which is the case in GLAM (Carta et al., 2023) where all primitive actions have similar lengths. However, this will harm the applicability of the methods which makes the design of the

action prompts difficult, even impossible sometimes. Therefore, we propose two normalization techniques, token normalization and word normalization, to address this issue.

**Token and Word Normalization.** A simple idea to solve this issue is to normalize the token-level probabilities of the actions with the number of tokens, which can be defined as:

$$\log P_{token}^{tn}(a_k|s) = \log P_{token}(a_k|s)/N_k \qquad (3)$$

We note that $\log P_{token}(a_k|s)$ will always be negative, the longer action prompts will have smaller negative values, therefore, dividing by the number of tokens will make the token-level probabilities of longer action prompts fall in the same magnitude with short action prompts. Another option is that instead of dividing the number of tokens, we can normalize the token-level probability by dividing the number of words, i.e.,

$$\log P_{token}^{wn}(a_k|s) = \log P_{token}(a_k|s)/W_k \qquad (4)$$

where $W_k$ is the number of words in the action prompt $k$. For the example of *pick up the tomato*, $N_k = 5$ while $W_k = 4$.

**Comparing the Two Normalizations.** Though token normalization can eliminate the influence brought by the length of action prompts, it is slightly excessive. We observe that if a word is divided into several tokens, the first token usually has a relatively low probability, while the probabilities of the rest of the tokens tend to be remarkably high, which are often almost close to 100%. For example, another important action, *chop*, in the Overcooked environment, is made up of two tokens, *ch* and *op*. The probabilities of *ch* are usually in the range of $0 - 20\%$, depending on the priority given by the LLM agent according to the observation, however, once *ch* appears, the probability of *op* will boost to $90 - 99\%$ in the next word since there are no other words starting with *ch* in the observation prompts. The same phenomenon is also observed in *tomao* and *dish* as shown in Figure 2. LLMs discovered and learned this statistical law during the autoregressive training process. Thus, instead of normalizing the probabilities of actions according to the number of tokens, it is more reasonable to regard the several tokens made up of one word as an integrated symbol. Therefore, word normalization is a more suitable normalization, where the log probabilities of action get averaged according to the number of words in the action prompts.

### 4.3 PARAMETER-EFFICIENT PPO FINETUNING

In this section, we present the parameter-efficient finetuning method to align LLMs and embodied environments with the generated policies under the PPO framework. We first introduce the network design and then the training procedure.

**Architecture.** As shown in Figure 3, we add additional MLP layers to the last transformer block of LLaMA-7B model as the critic. The critic's MLPs use the last token of the observation prompt as input and output the estimated value of the observation prompt. On the other hand, the actor is formed by the frozen LLaMA-7B model with the augmentation of LoRA parameters. We also note that the dropout layers would bring additional training instabilities, because the randomness of the dropout may violate the KL divergence constraint in PPO, therefore, we do not use dropout in our LoRA modules. During training, only the MLPs for

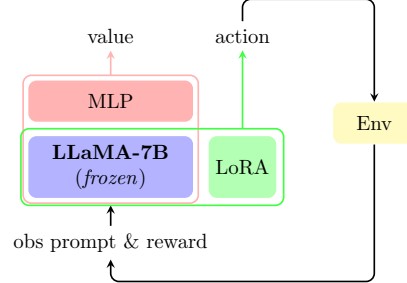

Figure 3: Parameter-efficient architecture

the critic and the LoRA parameters for the actor are updated, which makes the training efficient. The LLaMA-7B model can also serve as the reference model to regularize the update of the parameters, which is not explored in this work. Though we focus on the decoder-only models, our method can be seamlessly extended to the encoder-decoder architecture, e.g., T5 (Raffel et al., 2020).

**Training.** The training procedure generally follows PPO (Schulman et al., 2017), which is demonstrated to be effective in finetuning LLMs with human feedback (Ouyang et al., 2022). We observe the training instability when updating the actor multiple times with the same data, which is also

observed in DeepSpeed Chat blog[2]. Therefore, every sampled data is discarded after training once. Given the fact that the newly added MLPs in the critic are initialized randomly, while the LoRA parameters in the actor are initialized as zero, i.e., the output of the actor is exactly the same as the LLaMA-7B model, which is reasonable, therefore, a larger learning rate for the critic and a smaller learning rate for the actor is preferred for stable training and fast convergence.

**Inference.** During inference, the critic is discarded and only the actor is needed. Furthermore, the alignment of the LLMs and embodied environments is fully encoded in the LoRA parameters, which is normally 20 times less than the LLMs, e.g., 4.2M for LLaMA-7B. Therefore, the LoRA parameters can be a plug-and-play module of LLMs for generalizability across different environments.

### 4.4 PROMPT DESIGN

The prompts of observations and actions will significantly influence the generated policies by LLMs, which is a path orthogonal way to the finetuning for the alignment between LLMs and embodied environments. We summarize *four* principles for designing efficient prompts:

- Prompts of observations and actions should be cohesive for concatenation. Observation prompts end with *you should* and *the next step is to*, indicating the start of action prompts.
- Articles, i.e., *the*, *a*, and *an*, are important for action prompts. Most action prompts consist of a verb and a noun, e.g., *pick up the tomato*. As LLMs are trained with high-quality corpus, they are sensitive to the articles. Therefore, *pick up the tomato* is better than *pick up tomato*, where the latter leads to the extremely low probability on *tomato*.
- Preferred actions should appear in the observation prompt. As observed in (Xu et al., 2022; Fu et al., 2021), LLMs tend to assign higher probabilities to the repetitive tokens. Therefore, we can encourage LLMs to assign higher probability on preferred actions by emphasizing the nouns several times in the observation prompts. For example, if the observation prompt is *I see a tomato. My task is to make a tomato salad. I should*, then the *tomato* will have a relatively high probability.
- The same action can have different action prompts under different observations. For example, in Overcooked, when the agent carries a bowl in hand, *pick up the toamto* can be replaced with *put the tomato in the plate*, where both action prompts have the same function in the environment, the latter one better fits with the context and thus has higher probability.

The objective of the prompt design is to represent the observations and actions in the understandable manner of LLMs, thus improving the alignment between LLMs and embodied environments.

## 5 EXPERIMENTS

### 5.1 EXPERIMENTS SETUP

We deploy our methods on the LLaMA-7B model with half-precision and compare the performance among five methods: PPO (adapted from CleanRL (Huang et al., 2022a)), TWOSOME without finetuning (similar to SayCan (Brohan et al., 2023) with affordance value set to 1), TWOSOME without action prompt normalization (similar to GLAM (Carta et al., 2023) under decoder-only architecture and high-level actions) and TWOSOME with token normalization or word normalization. We mainly evaluate these methods in a typical decision-making environment, Overcooked, and a simulated physical household environment, VirtualHome.

**Overcooked** An agent is placed in the $7 \times 7$ Overcooked kitchen, aiming to make and serve a *tomato salad* and *tomato-lettuce salad* with the provided ingredients and tools in two tasks shown in Figure 4a and 4b. In the second task, we add an additional ingredient, onion as a disruptor, to show the robustness of our method. The agent needs to explore and learn the correct order to cook the dish with the provided macro-actions, such as *Chop*, *Get-Tomato*, and *Go-Cutting-Board*. The environment is partially observable. The agent only observes the objects within $5 \times 5$ square centered on the agent. The reward involves $+0.2$ for chopping a correct ingredient, $+1$ terminal reward for delivering the correct dish, $-0.1$ for delivering any wrong item, and $-0.001$ for every time step. The environment is adapted from Xiao et al. (2022) and Wu et al. (2021).

---

[2]`https://github.com/microsoft/DeepSpeedExamples/blob/master/
applications/DeepSpeed-Chat/training/step3_rlhf_finetuning/README.md`

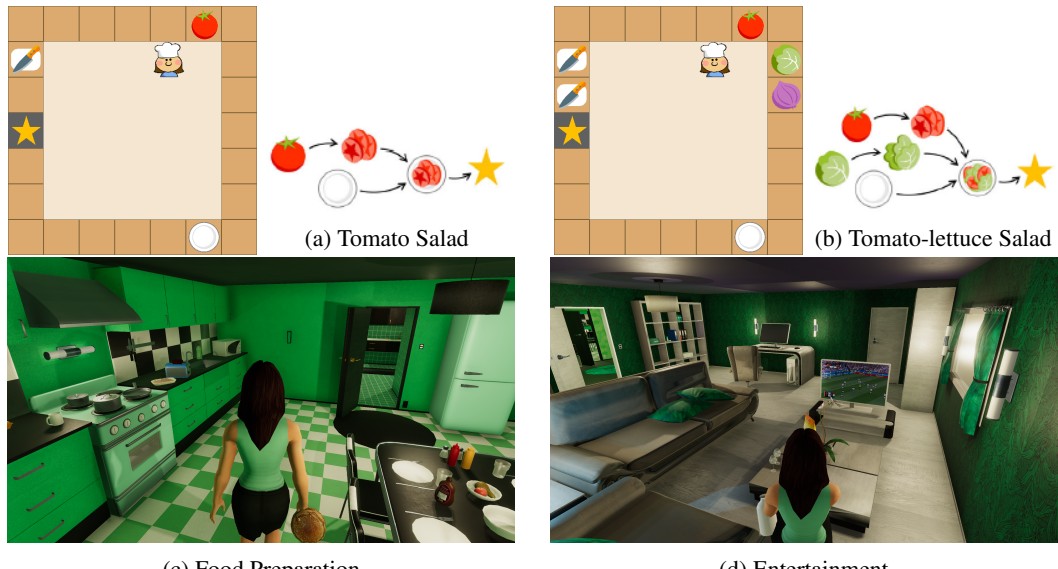

(a) Tomato Salad

(b) Tomato-lettuce Salad

(c) Food Preparation

(d) Entertainment

Figure 4: **Experimental environments.** Figure 4a and 4b show two tasks in Overcooked. Figure 4c and 4d show two tasks in VirtualHome.

**VirtualHome**    An agent is placed in a fully furnished household with various rich objects. Compared to Overcooked, this environment is more complex with a larger action space. The agent uses macro-actions to interact with the environment such as *walk to the living room*, *turn on the TV* and *sit on the sofa*. As shown in Figure 4c and 4d, we design two tasks. For the first task in Figure 4c, the agent needs to find the cold pancake on the table and heat it with the microwave in the kitchen. For the second task shown in Figure 4d, the agent plans to have some entertainment so it wants to find something to enjoy while watching TV. Thus, it needs to pick up chips and milk in the kitchen, bring them to the living room, turn on the TV, sit on the sofa and enjoy. The challenge arises when the agent already holding both the milk and chips, lacks an additional hand to turn on the TV. Consequently, it needs to learn to put at least one item on the nearby coffee table before operating the TV. Both tasks adopt a sparse reward setting, only when the task is finished, will the agent receive +1 reward. The environment is also partially observable. The agent can only see the objects in the current room. The environment is adapted from (Puig et al., 2018). A more detailed introduction of these two environments can be found in Appendix C. Our parameter-efficient framework enables us to complete all the experiments in one NVIDIA Tesla A100 40GB GPU. All the hyperparameters can be found in Appendix D.2. Finally, we provide the policy visualization and a detailed analysis of prompt design for all tasks in Appendix E and F.

## 5.2    THE IMPACT OF DIFFERENT NORMALIZATIONS IN TWOSOME

Figure 5a shows the performance of finetuned TWOSOMEs. TWOSOME with word normalization succeeds in learning the optimal policy among all tasks, exhibiting great performance, and high sample efficiency over other methods, which is consistent with the analysis in Section 4.2. Except for the second Overcooked task, which is more difficult, TWOSOME without normalization learns quickly at the beginning but suffers a severe sudden drop in the the other three tasks. Without normalization, the unbalance over action distribution can accelerate the training process if the shorter action prompts happen to be reasonable actions but introduce extra training instability. TWOSOME with token normalization alleviates this instability slightly excessively, resulting in less data efficiency and slow convergence, as shown in the two overcooked tasks.

## 5.3    TWOSOME VS. BASELINES

Figure 5b shows the performance among the typical RL method, PPO, prompt tuning method, TWOSOME without finetuning, and our best method, TWOSOME with word normalization.

**Comparison with Prompt Tuning.**    TWOSOME without finetuning fully relies on the capabilities of pre-trained LLMs with the designed prompts, demonstrating the LLM's decision-making ability

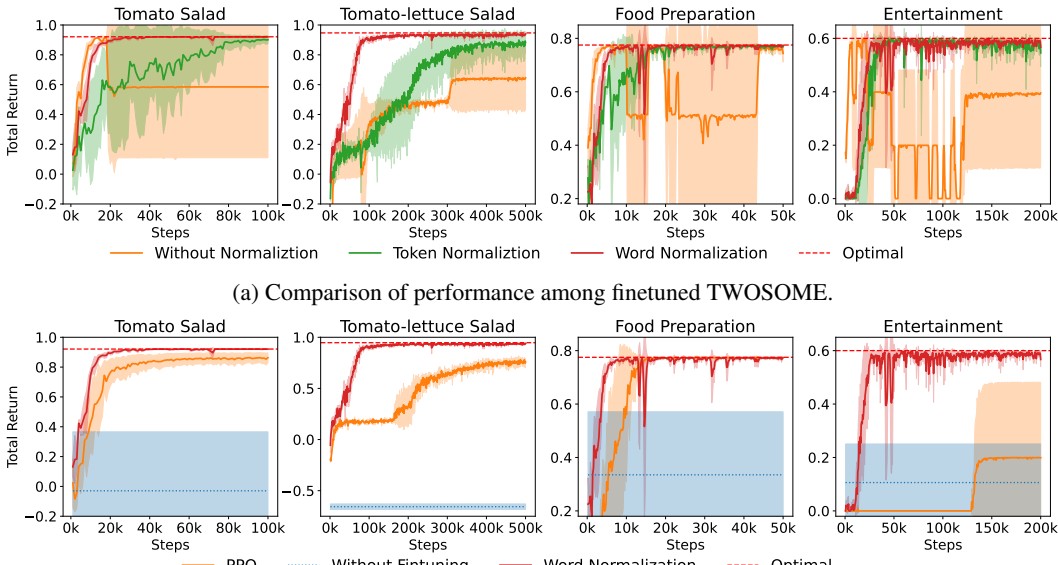

(a) Comparison of performance among finetuned TWOSOME.

(b) Comparison of performance among TWOSOME with word normalization and baselines.

Figure 5: Comparison of performance among PPO and four TWOSOME variants: i) without finetuning, ii) without normalization, iii) with token normalization, and iv) with word normalization. PPO masks unrelated actions in both VirtualHome tasks. The results are averaged over 3 seeds. The non-trained method is averaged over 100 episodes.

and the difficulty of the task. For the second task, which is the most difficult task, the non-finetuned LLM agent fails to finish the task completely. For the other three tasks, the low total returns with large variations indicate that the non-finetuned LLM agent can only finish these tasks with low probabilities. Prompt tuning can improve LLMs in solving easy decision-making tasks but difficult to solve complex and long-horizontal tasks with more interferents in the environment. We conclude that all training methods including PPO and all TWOSOME variants surpass the performance of the prompt tuning method, emphasizing the necessity of aligning LLMs with environments via RL.

**Comparison with PPO.** TWOSOME with word normalization succeeds in learning the optimal policies in both Overcooked tasks using 10k and 80k sampled data, respectively. In contrast, PPO fails to learn the optimal policy and gets stuck in the suboptimal due to partial observability. For the more complex second task, PPO even needs more than 500K steps to converge. As for the two tasks in VirtualHome, we find that vanilla PPO cannot deal with the large action space and learns nothing, as shown in Appendix C.3.1. So we mask all the unrelated actions in VirtualHome for PPO. The masked PPO manages to learn the optimal policy in the task of *Food Preparation* but still fails in the task of *Entertainment*. The results demonstrate that by leveraging the power of LLM, our method can defeat the traditional RL methods in solving decision-making tasks with large action spaces.

All finetuned TWOSOME methods, no matter whether they are normalized, manage to overcome partial observability and achieve optimal performance in at least two seeds in the two Overcooked tasks. Even though the agent does not see the target object, e.g., tomato, it can still appear in the goal part of the observation prompt, such as *your task is to make a salad consisting of tomato*, maintaining the high probabilities of tomato related actions, while other unrelated objects like onion, not appeared in neither the observation nor the observation prompt, remain relatively low probabilities. This feature greatly facilitates the exploration process and helps agents to sample good actions.

## 5.4 OPEN-VOCABULARY TASK GENERALIZATION

We also evaluate the generalization ability of TWOSOME in eight new unseen tasks. LLMs' open-vocabulary feature enables TWOSOME to transfer learned skills to different tasks, while traditional RL agents do not have such ability. We compare the performance between our finetuned TWOSOME and non-tuned TWOSOME. Figure 6 shows that for the first four tasks, which are similar to the agent's original trained task, though non-tuned TWOSOME can somehow finish the tasks, fine-

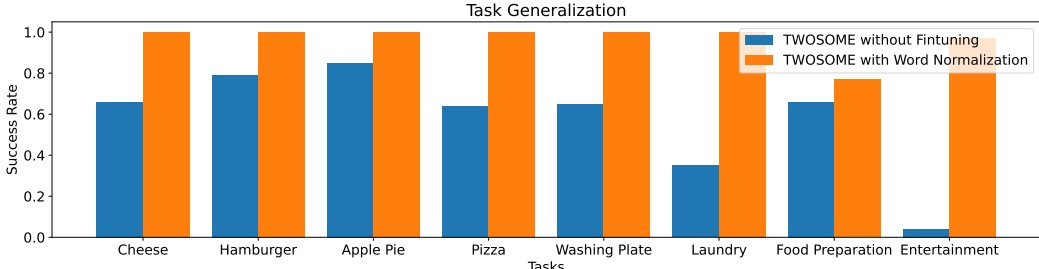

Figure 6: **Task Generalization Tests.** TWOSOME without Finetuning and TWOSOME with word normalization are tested in eight **unseen** tasks. Each task is tested over 100 episodes. For the first four tasks, *Cheese*, *Hamburger*, *Apple Pie* and *Pizza*, we replace the pancake in the original *Food Preparation* task with the corresponding food to see whether the agent can still finish the task. The following *Washing Plate* and *Laundry* tasks are more different but still have similar procedures to *Food Preparation* (agent needs to put dishes or clothes into the dishwasher or washing machine). For the last two crossover tests, TWOSOME agent trained in *Food Preparation* is tested in *Entertainment* and TWOSOME trained in *Entertainment* is tested in *Food Preparation*.

tuned TWOSOME achieves perfect performance. Replacing objects has little impact on finetuned TWOSOME. For the following two more different tasks, *Washing Plate* and *Laundry*, there is an obvious drop in the success rate of non-tuned TWOSOME, while finetuned TWOSOME still manages to complete the task. For the last two crossover tasks, where agents are tested in a completely different task, TWOSOME still exhibits remarkable performance. Especially in the *Entertainment* task, where non-tuned TWOSOME can barely finish the task and finetuned TWOSOME still maintains a 97% success rate. Total returns of each task are provided in Appendix B.2.

## 5.5  IMPACT OF ONLINE FINETUNING

To investigate the impacts of online finetuning on the LLM's abilities, we evaluate the models trained by TWOSOME with word normalization in Virtualhome on widely used NLP benchmarks (Gao et al., 2021). The models trained in *Food Preparation* and *Entertainment* are named TWOSOME-FP and TWOSOME-E, respectively. The four benchmarks are

Table 1: Zero-shot performance on Language Model Evaluation Harness (Gao et al., 2021).

|  | ARC_C | HellaSwag | PIQA | MMLU |
|---|---|---|---|---|
| LLaMA-7B-hf | 0.42 | 0.57 | 0.79 | 0.33 |
| TWOSOME-FP | 0.43 | 0.58 | 0.79 | 0.32 |
| TWOSOME-E | 0.43 | 0.58 | 0.79 | 0.30 |

ARC_C, HellaSwag, PIQA and MMLU, which are also reported in Touvron et al. (2023). Results of the zero-shot performance are displayed in Table 1, which demonstrates that there is no significant loss of the ability of the language understanding of LLMs after the aligning with embodied environments, and even sometimes brings minor improvements. The full results are in Appendix B.3.

## 6  DISCUSSION AND CONCLUSION

In this paper, we propose TWOSOME, a general online framework for efficiently aligning LLMs with embodied environments via RL to solve decision-making tasks without requiring any prepared datasets or prior knowledge of environments and without significant loss of LLMs' original ability. Instead of letting LLMs generate actions directly, TWOSOME is more controllable and feasible with better interpretability, since the impact of tuning prompts can be clearly reflected by the change of action probabilities. TWOSOME with word normalization exhibits significantly better sample efficiency and performance compared to traditional RL methods and prompt tuning methods. Moreover, TWOSOME exhibits a remarkable generalization ability to transfer learned skills and knowledge to unseen tasks. However, our method suffers a major limitation that training a PPO agent from scratch is far faster and cheaper than finetuning an LLM. TWOSOME needs to feed all the valid actions to the LLMs for every action sampling, resulting in multiple times the amount of computation and a small batch size. We hope this work can provide a step toward the general autonomous agent, where LLMs can self-improve by interacting with the world and harvesting true knowledge from practice.

REPRODUCIBILITY STATEMENT

This work does not use any dataset. All the TWOSOME experimental code and environment code of Overcooked and VirtualHome are included in the Supplementary Materials. We also provide videos for each task recorded by our best agent in the Supplementary Materials. All the hyperparameters and network architecture we use can be found in Appendix D.1 and D.2. We provide the policy visualization and a detailed analysis of prompt design for all tasks in Appendix E and F. Our parameter-efficient framework enables us to complete all the experiments in one NVIDIA Tesla A100 40GB GPU. Additional experimental results such as success rate and NLP benchmarks can be found in Appendix B. A detailed introduction to the Overcooked and VirtualHome environments, including observation space, action space, and macro actions we use can be found in Appendix C.

ACKNOWLEDGMENTS

This research is supported by the National Research Foundation Singapore and DSO National Laboratories under the AI Singapore Programme (AISGAward No: AISG2-GC-2023-009). This research is also supported by the National Research Foundation, Singapore under its Industry Alignment Fund – Pre-positioning (IAF-PP) Funding Initiative. Any opinions, findings and conclusions or recommendations expressed in this material are those of the author(s) and do not reflect the views of National Research Foundation, Singapore. The computational work for this work was partially performed on resources of the National Supercomputing Centre, Singapore (https://www.nscc.sg).

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

APPENDIX

## A MORE RELATED WORK

**Embodied Agent with LLMs.**    The successful integration of language as a semantically rich input for interactive decision-making highlights the crucial role of LLMs in facilitating interaction and decision-making processes (Abramson et al., 2020; Karamcheti et al., 2021; Li et al., 2022). LLMs are also applied in various environments to aid robot navigation (Parisi et al., 2022; Hong et al., 2021; Majumdar et al., 2020) and manipulation (Jiang et al., 2022; Ren et al., 2023; Khandelwal et al., 2022). Recently, there have been a large number of methods that utilize LLMs to enhance planning and reasoning capabilities in embodied agents. SayCan (Brohan et al., 2023) assesses the affordance of candidate actions by multiplying their probabilities under LLMs with a value function. Zeng et al. (2022) combine the LLM with a visual-language model and a pre-trained language-conditioned policy (Shridhar et al., 2022) to enable open vocabulary robotic tasks. Huang et al. (2022b) demonstrate that LLMs can be employed for planning and executing simple household tasks. They ground LLM-generated actions by comparing their embeddings with a predefined list of acceptable actions. To incorporate environment feedback, Inner Monologue (Huang et al., 2022c) extends SayCan using a closed-loop principle. This principle is also applied in related works such as (Yao et al., 2023; Huang et al., 2022c; Kim et al., 2023; Singh et al., 2022; Liang et al., 2022; Shinn et al., 2023; Wang et al., 2023b;a) to continuously monitor agent behaviors and refine and adjust plans accordingly for tasks such as computer automation, Minecraft, etc. Furthermore, there are approaches that prompt LLMs to generate temporal-abstracted actions (Zheng et al., 2023). Dasgupta et al. (2022) employ the LLM as a planner and success detector for an agent with their actor module necessitates pre-training with RL to enable the agent to follow natural language instructions. While these works demonstrate impressive results, they rely too heavily on the inherent capabilities of powerful LLMs, like GPT4 and PaLM (Chowdhery et al., 2022), which are difficult to apply to smaller LLMs with weaker reasoning abilities, like LLaMA-7B.

Concurrent to our work, GLAM (Carta et al., 2023) utilizes RL finetuning to achieve functional grounding of LLMs. However, they focus on simple primitive actions (turn left, turn right, go forward, etc.) evaluated in toy environments, BabyAI (Chevalier-Boisvert et al., 2018) with a much smaller encoder-decoder LLM, Flan-T5-780M. These primitive actions have a similar number of tokens and less meaningful semantics, resulting in underutilizing the capabilities of LLMs, and failing to observe the impact of prompt design and address the unbalance over action space, resulting in additional instability and poor robustness.

**Finetuning LLMs.**    Parameter-efficient finetuning (PEFT) can significantly reduce the number of parameters to tune LLMs while with minor loss of the performance (Ding et al., 2023). *Prompt tuning* (Lester et al., 2021) and *prefix tuning* (Li & Liang, 2021) prepend additional tunable tokens to the input or hidden layers, and *adapter tuning* (Houlsby et al., 2019) introduces the adapter, i.e., the bottleneck layers, to LLMs, where only the introduced tokens and adapters are finetuned. Recently, low-rank adaption (LoRA) (Hu et al., 2022) introduces low-rank matrices to approximate parameter updates.

Reinforcement learning from human feedback (RLHF) (Ouyang et al., 2022) proves that RL is an effective way to make alignments with human preferences and proposes a novel perspective regarding LLMs as actors under actor-critic framework. Our work is also motivated by this method. While RLHF needs to train a reward model to simulate human preferences, these signals are naturally present in most RL environments, which makes it possible to align LLMs with certain environments.

Another concurrent work (Xiang et al., 2023) leverages VirtualHome to collect embodied experiences by random sampling and use supervised learning to finetune LLMs to these embodied knowledge. They finally apply a simple RL method, MCTS, to plan without training, instead of directly generating plans. It is worth pointing out that our method also applies their supervised learning process to learn the embodied knowledge and get more familiar with the environments. Our online PPO sampling is much more efficient and feasible than their random sampling, e.g. in the overcooked tasks, it is almost impossible to complete the task by random sampling. In addition, an online learning PPO is obviously better than a simple MCTS without training. Compared with this work, our method is a simpler end-to-end online framework, which can automatically acquire new knowledge and solve decision-making tasks by interacting with environments without any prepared datasets.

# B  ADDITIONAL EXPERIMENTAL RESULTS

## B.1  SUCCESS RATE IN OVERCOOKED AND VIRTUALHOME

Besides the total reward provided in 5b, here we also provide the final success rates of all methods in Table 2, which tells almost the same story. Our TWOSOME with word normalization method achieves a 100% success rate across all tasks, exhibiting great stability and remarkable performance.

Table 2: Final success Rate of different methods in Overcooked and VirtualHome. Results are averaged over three seeds. Each seed is run for 100 episodes. Norm is short for normalization.

| Task | PPO | TWOSOME | | | |
| | | W/O Finetuning | W/O Norm | Token Norm | Word Norm |
|---|---|---|---|---|---|
| Tomato Salad | $1 \pm 0$ | $0.36 \pm 0.04$ | $0.67 \pm 0.47$ | $1 \pm 0$ | $\mathbf{1 \pm 0}$ |
| Tomato-lettuce Salad | $1 \pm 0$ | $0 \pm 0$ | $1 \pm 0$ | $1 \pm 0$ | $\mathbf{1 \pm 0}$ |
| Food Preparation | $1 \pm 0$ | $0.81 \pm 0.01$ | $1 \pm 0$ | $1 \pm 0$ | $\mathbf{1 \pm 0}$ |
| Entertainment | $0.33 \pm 0.47$ | $0.45 \pm 0.05$ | $0.67 \pm 0.47$ | $1 \pm 0$ | $\mathbf{1 \pm 0}$ |

## B.2  PERFORMANCE OF TWOSOME AND SAYCAN IN EIGHT UNSEEN TASKS

Table 3 and 4 show the final success rate and total return of our best method TWOSOME with word normalization and our baseline, SayCan(non-finetuned TWSOME) in eight unseen tasks. Each task is run for 100 episodes. TWOSOME exhibits superior generalization across all tasks.

Table 3: Final success rate of TWOSOME and SayCan in eight unseen tasks.

| Task | Cheese | Hamhurger | Applepie | Pizza | Washing Plate | Laundry | Food Preparation | Entertainment |
|---|---|---|---|---|---|---|---|---|
| TWOSOME | 1 | 1 | 1 | 1 | 1 | 1 | 0.77 | 0.97 |
| SayCan | 0.66 | 0.79 | 0.85 | 0.64 | 0.65 | 0.35 | 0.66 | 0.04 |

Table 4: Total return of TWOSOME and SayCan in eight unseen tasks.

| Task | Cheese | Hamhurger | Applepie | Pizza | Washing Plate | Laundry | Food Preparation | Entertainment |
|---|---|---|---|---|---|---|---|---|
| TWOSOME | 0.77±0.01 | 0.77±0.01 | 0.77±0.01 | 0.77±0.01 | 0.77±0.02 | 0.76±0.04 | 0.26±0.24 | 0.43±0.14 |
| SayCan | 0.18±0.17 | 0.21±0.17 | 0.23±0.17 | 0.17±0.18 | 0.18±0.17 | 0.08±0.13 | 0.20±0.16 | 0.01±005 |

## B.3  EVALUATION ON NLP BENCHMARKS

To investigate the impacts of the PPO online finetuning on the LLM's abilities, we evaluate the models trained by TWOSOME with word normalization in the virtual home tasks on widely used NLP benchmasrks (Gao et al., 2021). The models trained in *Food Preparation* and *Entertainment* are named TWOSOME-FP and TWOSOME-E, respectively. The four benchmarks are ARC_C, HellaSwag, PIQA and MMLU, which are also reported in (Touvron et al., 2023). The results on ARC_Challenge, HellaSwag and PIQA are displayed in Table 5 and the results of MMLU are displayed in Table 6. All results are calculated following the default configurations in (Gao et al., 2021).

Table 5: Zero-shot performance on Common Sense Reasoning tasks.

| | LLaMA | TWOSOME-FP | TWOSOME-E |
|---|---|---|---|
| ARC_C | $0.42 \pm 0.01$ | $0.43 \pm 0.01$ | $0.43 \pm 0.01$ |
| HellaSwag | $0.57 \pm 0.00$ | $0.58 \pm 0.00$ | $0.58 \pm 0.00$ |
| PIQA | $0.79 \pm 0.01$ | $0.79 \pm 0.01$ | $0.79 \pm 0.01$ |

Table 6: Zero-shot performance on Massive Multitask Language Understanding (MMLU).

| | LLaMA | TWOSOME-FP | TWOSOME-E |
|---|---|---|---|
| abstract_algebra | $0.33 \pm 0.05$ | $0.30 \pm 0.05$ | $0.28 \pm 0.05$ |
| anatomy | $0.38 \pm 0.04$ | $0.39 \pm 0.04$ | $0.36 \pm 0.04$ |
| astronomy | $0.34 \pm 0.04$ | $0.32 \pm 0.04$ | $0.24 \pm 0.03$ |
| business_ethics | $0.37 \pm 0.05$ | $0.31 \pm 0.05$ | $0.32 \pm 0.05$ |
| clinical_knowledge | $0.37 \pm 0.03$ | $0.35 \pm 0.03$ | $0.31 \pm 0.03$ |
| college_biology | $0.37 \pm 0.04$ | $0.35 \pm 0.04$ | $0.34 \pm 0.04$ |
| college_chemistry | $0.28 \pm 0.05$ | $0.30 \pm 0.05$ | $0.27 \pm 0.04$ |
| college_computer_science | $0.25 \pm 0.04$ | $0.26 \pm 0.04$ | $0.31 \pm 0.05$ |
| college_mathematics | $0.34 \pm 0.05$ | $0.35 \pm 0.05$ | $0.31 \pm 0.05$ |
| college_medicine | $0.31 \pm 0.04$ | $0.32 \pm 0.04$ | $0.27 \pm 0.03$ |
| college_physics | $0.25 \pm 0.04$ | $0.25 \pm 0.04$ | $0.20 \pm 0.04$ |
| computer_security | $0.38 \pm 0.05$ | $0.40 \pm 0.05$ | $0.40 \pm 0.05$ |
| conceptual_physics | $0.32 \pm 0.03$ | $0.34 \pm 0.03$ | $0.31 \pm 0.03$ |
| econometrics | $0.23 \pm 0.04$ | $0.20 \pm 0.04$ | $0.21 \pm 0.04$ |
| electrical_engineering | $0.26 \pm 0.04$ | $0.26 \pm 0.04$ | $0.26 \pm 0.04$ |
| elementary_mathematics | $0.24 \pm 0.02$ | $0.23 \pm 0.02$ | $0.23 \pm 0.02$ |
| formal_logic | $0.34 \pm 0.04$ | $0.32 \pm 0.04$ | $0.27 \pm 0.04$ |
| global_facts | $0.32 \pm 0.05$ | $0.29 \pm 0.05$ | $0.26 \pm 0.04$ |
| high_school_biology | $0.32 \pm 0.03$ | $0.32 \pm 0.03$ | $0.27 \pm 0.03$ |
| high_school_chemistry | $0.21 \pm 0.03$ | $0.21 \pm 0.03$ | $0.19 \pm 0.03$ |
| high_school_computer_science | $0.24 \pm 0.04$ | $0.27 \pm 0.04$ | $0.28 \pm 0.05$ |
| high_school_european_history | $0.40 \pm 0.04$ | $0.38 \pm 0.04$ | $0.41 \pm 0.04$ |
| high_school_geography | $0.28 \pm 0.03$ | $0.29 \pm 0.03$ | $0.26 \pm 0.03$ |
| high_school_government_and_politics | $0.30 \pm 0.03$ | $0.30 \pm 0.03$ | $0.27 \pm 0.03$ |
| high_school_macroeconomics | $0.28 \pm 0.02$ | $0.26 \pm 0.02$ | $0.24 \pm 0.02$ |
| high_school_mathematics | $0.24 \pm 0.03$ | $0.25 \pm 0.03$ | $0.27 \pm 0.03$ |
| high_school_microeconomics | $0.26 \pm 0.03$ | $0.26 \pm 0.03$ | $0.24 \pm 0.03$ |
| high_school_physics | $0.26 \pm 0.04$ | $0.27 \pm 0.04$ | $0.28 \pm 0.04$ |
| high_school_psychology | $0.33 \pm 0.02$ | $0.32 \pm 0.02$ | $0.27 \pm 0.02$ |
| high_school_statistics | $0.22 \pm 0.03$ | $0.19 \pm 0.03$ | $0.17 \pm 0.03$ |
| high_school_us_history | $0.40 \pm 0.03$ | $0.41 \pm 0.03$ | $0.40 \pm 0.03$ |
| high_school_world_history | $0.38 \pm 0.03$ | $0.40 \pm 0.03$ | $0.42 \pm 0.03$ |
| human_aging | $0.43 \pm 0.03$ | $0.40 \pm 0.03$ | $0.41 \pm 0.03$ |
| human_sexuality | $0.34 \pm 0.04$ | $0.32 \pm 0.04$ | $0.34 \pm 0.04$ |
| international_law | $0.47 \pm 0.05$ | $0.43 \pm 0.05$ | $0.36 \pm 0.04$ |
| jurisprudence | $0.36 \pm 0.05$ | $0.35 \pm 0.05$ | $0.35 \pm 0.05$ |
| logical_fallacies | $0.36 \pm 0.04$ | $0.35 \pm 0.04$ | $0.26 \pm 0.03$ |
| machine_learning | $0.27 \pm 0.04$ | $0.28 \pm 0.04$ | $0.33 \pm 0.04$ |
| management | $0.28 \pm 0.04$ | $0.31 \pm 0.05$ | $0.22 \pm 0.04$ |
| marketing | $0.38 \pm 0.03$ | $0.38 \pm 0.03$ | $0.39 \pm 0.03$ |
| medical_genetics | $0.42 \pm 0.05$ | $0.38 \pm 0.05$ | $0.36 \pm 0.05$ |
| miscellaneous | $0.45 \pm 0.02$ | $0.45 \pm 0.02$ | $0.42 \pm 0.02$ |
| moral_disputes | $0.35 \pm 0.03$ | $0.34 \pm 0.03$ | $0.29 \pm 0.02$ |
| moral_scenarios | $0.24 \pm 0.01$ | $0.24 \pm 0.01$ | $0.24 \pm 0.01$ |
| nutrition | $0.33 \pm 0.03$ | $0.33 \pm 0.03$ | $0.30 \pm 0.03$ |
| philosophy | $0.37 \pm 0.03$ | $0.34 \pm 0.03$ | $0.27 \pm 0.03$ |
| prehistory | $0.39 \pm 0.03$ | $0.34 \pm 0.03$ | $0.31 \pm 0.03$ |
| professional_accounting | $0.27 \pm 0.03$ | $0.28 \pm 0.03$ | $0.26 \pm 0.03$ |
| professional_law | $0.28 \pm 0.01$ | $0.28 \pm 0.01$ | $0.29 \pm 0.01$ |
| professional_medicine | $0.25 \pm 0.03$ | $0.24 \pm 0.03$ | $0.25 \pm 0.03$ |
| professional_psychology | $0.34 \pm 0.02$ | $0.31 \pm 0.02$ | $0.30 \pm 0.02$ |
| public_relations | $0.35 \pm 0.05$ | $0.33 \pm 0.04$ | $0.32 \pm 0.04$ |
| security_studies | $0.27 \pm 0.03$ | $0.27 \pm 0.03$ | $0.23 \pm 0.03$ |
| sociology | $0.41 \pm 0.03$ | $0.37 \pm 0.03$ | $0.37 \pm 0.03$ |
| us_foreign_policy | $0.42 \pm 0.05$ | $0.41 \pm 0.05$ | $0.41 \pm 0.05$ |
| virology | $0.36 \pm 0.04$ | $0.36 \pm 0.04$ | $0.33 \pm 0.04$ |
| world_religions | $0.46 \pm 0.04$ | $0.48 \pm 0.04$ | $0.44 \pm 0.04$ |

## C    DETAILS OF EMBODIED ENVIRONMENTS

In this section, we present the details of the embodied environments Overcooked and VirtualHome.

### C.1    OVERCOOKED

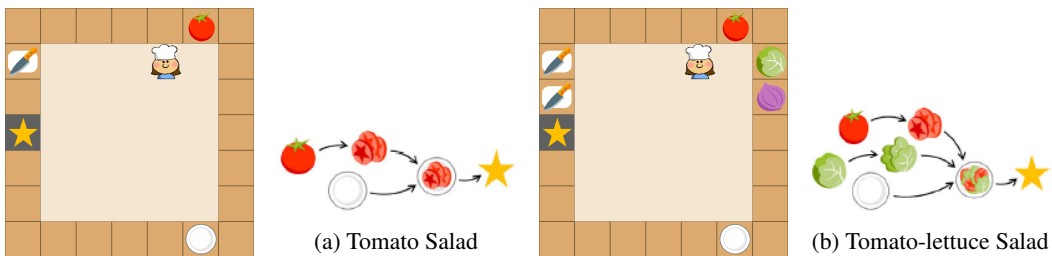

(a) Tomato Salad          (b) Tomato-lettuce Salad

Figure 7: Overcooked environments.

**Goal**. An agent is placed in the Overcooked kitchen and aims to cook a certain dish with the provided ingredients and tools and deliver it to the 'star' cell as soon as possible. Agents have to learn the correct procedure in terms of picking up raw vegetables, chopping them, and merging them in a bowl before delivering. Figure 7a and 7b show the recipes for making tomato salad and tomato-lettuce salad.

**Observation Space**. The environment is a 7×7 grid world involving ingredients, bowls, cutting boards, and a delivery counter. For the task of making tomato salad, there is only one tomato, one bowl, one cutting board, and one delivery counter available in the environment. For the task of making a tomato-lettuce salad, one tomato, one lettuce, one onion, one bowl, two cutting boards, and one delivery cell are provided in the environment. The onion serves as an interferent, though it does not appear in the recipe. The global state information consists of the positions of the agent and the above objects, and the status of each ingredient: chopped or unchopped. The environment is partially observable. For the observation, the agent only observes the *positions* and *status* of the entities within a $5 \times 5$ square centered on the agent. Other unseen objects are masked in the observation. And the initial positions of all the objects are known to agents. The symbolic raw observations are directly fed into PPO as input and converted to prompts with scripts to serve as LLMs' input.

**Primitive-Action Space**. Agents use five primitive actions: *up*, *down*, *left*, *right*, and *stay* to interact with the environment. Agents can move around and achieve picking, placing, chopping, and delivering by standing next to the corresponding cell and moving against it. All the macro-actions are based on these primitive actions.

**Macro-Action Space**. In this work, we mainly focus on using high-level macro-actions, since they usually have richer semantics. Every macro-action may take several time steps to complete. The main function of each macro-action and the corresponding termination conditions is exactly the same as the setting in (Xiao et al., 2022) despite that we only have one agent. Here we list and briefly describe all the macro-actions we use: ***Chop***, chops a raw ingredient into pieces when the agent stands next to a cutting board with an unchopped ingredient on it. ***Get-Tomato***, ***Get-Lettuce***, ***Get-Onion***, ***Get-Bowl***, ***Go-Cutting-Board-1/2*** and ***Deliver***, which navigate the agent to the location of the corresponding object and execute the corresponding action.

In the first task of making tomato salad, ***Get-Tomato***, ***Get-Bowl***, ***Go-Cutting-Board-1*** and ***Deliver*** are available. In the second task of making tomato-lettuce salad all the macro-actions are valid.

**Dynamics**: The transition in this task is deterministic. If an agent delivers any wrong item, the item will be reset to its initial position.

**Reward**: $+0.2$ for chopping a correct ingredient, $+1$ terminal reward for delivering the correct dish, $-0.1$ for delivering any wrong dish, and $-0.001$ for every timestep.

**Episode Termination**: Each episode terminates either when the agent successfully delivers the target dish to the delivery counter or reaches the maximal time steps, 200.

## C.2 VIRTUALHOME

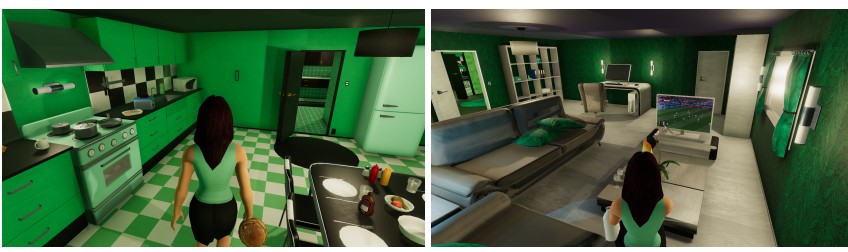

(a) Food Preparation           (b) Entertainment

Figure 8: VirtualHome environments.

**Goal.** An agent, represented as a humanoid avatar, is placed in a fully furnished household with various rich objects to interact with. Figure 8a and 8b show two tasks we design for the agent to complete. For the first task of food preparation, the agent needs to find the pancake on the table in the kitchen and put it in the microwave to heat. For the second task of entertainment, the agent needs to find and bring chips and milk, from the kitchen to the living room and succeed sitting on the sofa with the TV open and the milk and chips are nearby. The challenge arises when the agent, already holding both the milk and chips, lacks an additional hand to turn on the TV. Consequently, it needs to learn to put at least one item on the nearby coffee table before operating the TV.

**Observation Space**. The environment is also partially observable. The agent can only see the objects in the current room and does not the objects in the other room. The observation consists of a set of bool values, representing whether the agent sees the relative object, whether these objects are close to the agent and the status of the objects, such as whether the TV is turned on and whether the milk is on the coffee table. The symbolic raw observations are directly fed into PPO as input and converted to prompts with scripts to serve as LLMs' input.

## C.3 ACTION SPACE

To simulate the daily activities, all the actions in VirtualHome are macro-actions. Every macro-action takes only time step for execution. Here, we list and briefly describe all the macro-actions as Table 7.

Table 7: Actions in VirtualHome and their corresponding descriptions

| Action | Description |
| --- | --- |
| [STANDUP] | Stand up |
| [SIT] <Object> <Object> (object_id) | Sit on the *Object(object_id)* |
| [WALK] <Object> (object_id) | Walk to the *Object(object_id)* |
| [GRAB] <Object> (object_id) | Grab the *Object(object_id)* |
| [OPEN] <Object> (object_id) | Open the *Object(object_id)* |
| [CLOSE] <Object> (object_id) | Close the *Object(object_id)* |
| [SWITCHON] <Object> (object_id) | Switch/Turn on the *Object(object_id)* |
| [SWITCHOFF] <Object> (object_id) | Switch/Turn off the *Object(object_id)* |
| [PUTIN] <Object1> (object1_id) <Object2> (object2_id) | Put the *Object1(object1_id)* in the *Object2(object2_id)* |
| [PUTBACK] <Object1> (object1_id) <Object2> (object2_id) | Put the *Object1(object1_id)* on the *Object2(object2_id)* |

For the first task of heating pancake, there are four rooms and two items in the environment and their corresponding object id are as follows: *livingroom(267), kitchen(11), bathroom(172), bedroom(210), pancake(62), microwave(109).* For the second task of watching TV, there are four rooms and five items in the environment and their corresponding object id are as follows: *livingroom(267), kitchen(11), bathroom(172), bedroom(210), chips(61), milk(46), TV(297), sofa(276), coffeetable(268).* For their actions, corresponding action prompts and abbreviated labels, please refer to Table 8 and Table 9.

Only when the agent is close to the object, can the agent operate the object. For example, the agent needs to walk to the TV before turning it on.

Table 8: Actions, action prompts and abbreviated labels in the heating pancake task

| Action | Action Prompt | Label |
|---|---|---|
| [WALK] *<livingroom> (267)* | walk to the living room | Walk-Livingroom |
| [WALK] *<kitchen> (11)* | walk to the kitchen | Walk-Kitchen |
| [WALK] *<bathroom> (172)* | walk to the bathroom | Walk-Bathroom |
| [WALK] *<bedroom> (210)* | walk to the bedroom | Walk-Bedroom |
| [WALK] *<pancake> (62)* | reach for the pancake | Walk-Pancake |
| [WALK] *<microwave> (109)* | move to the microwave | Walk-Microwave |
| [GRAB] *<pancake> (62)* | grab the pancake | Grab-Pancake |
| [OPEN] *<microwave> (109)* | open the microwave | Open-Microwave |
| [CLOSE] *<microwave> (109)* | close the microwave | Close-Microwave |
| [PUTIN] *<pancake> (62) <microwave> (109)* | put the pancake in the microwave | Putin-Pancake-Microwave |

Table 9: Actions, action prompts and abbreviated labels in the watching TV task

| Action | Action Prompt | Label |
|---|---|---|
| [WALK] *<livingroom> (267)* | walk to the living room | Walk-Livingroom |
| [WALK] *<kitchen> (11)* | walk to the kitchen | Walk-Kitchen |
| [WALK] *<bathroom> (172)* | walk to the bathroom | Walk-Bathroom |
| [WALK] *<bedroom> (210)* | walk to the bedroom | Walk-Bedroom |
| [WALK] *<chips> (61)* | reach for the chips | Walk-Chips |
| [WALK] *<milk> (46)* | reach for the milk | Walk-Milk |
| [WALK] *<coffeetable> (268)* | move to the coffee table | Walk-Coffeetable |
| [WALK] *<TV> (297)* | move to the TV | Walk-TV |
| [WALK] *<sofa> (276)* | move to the sofa | Walk-Sofa |
| [GRAB] *<chips> (61)* | grab the chips | Grab-Chips |
| [GRAB] *<milk> (46)* | grab the milk | Grab-Milk |
| [SWITCHON] *<TV> (297)* | turn on the TV | Switchon-TV |
| [SWITCHOFF] *<TV> (297)* | turn off the TV | Switchoff-TV |
| [SIT] *<sofa> (276)* | take a seat on the sofa | Sit-Sofa |
| [STANDUP] | stand up from the sofa | Standup-Sofa |
| [PUTBACK] *<chips> (61) <coffeetable> (268)* | put the chips on the coffee table | Putback-Chips-Coffeetable |
| [PUTBACK] *<milk> (46) <coffeetable> (268)* | put the milk on the coffee table | Putback-Milk-Coffeetable |

**Dynamics**: The transition in this task is deterministic.

**Reward**: We adopt a sparse reward setting, where only when the task is finished will the agent receive +1 reward.

**Episode Termination**: Each episode terminates either when the agent successfully finishes the task or reaches the maximal time steps, 50 since every macro-action takes only time step for execution. For the task of heating pancake, the agent succeeds when it places the pancake in the microwave and closes the microwave. For the task of watching TV, the agent succeeds when it sits on the sofa with TV turned on and chips and milk are on the coffee table or hold in hand.

### C.3.1 VANILLA PPO VS PPO WITH ACTION MASK

As mentioned in Section 5.3, in VirtualHome environment, vanilla PPO without action mask cannot learn anything for the two tasks due to the large action space. Figure 9 shows the performance of vanilla PPO without action mask and PPO with action mask.

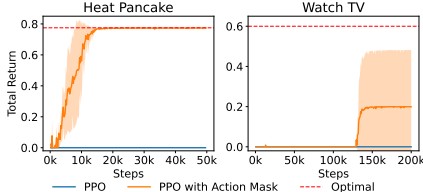

Figure 9: Performance of PPO without action mask and PPO with action mask in VirtualHome tasks.

# D    TRAINING DETAILS

Our results are mainly generated on a single NVIDIA Tesla A100 and NVIDIA RTX A6000 GPU. For all domains, we use the same architecture, which is shown in Figure 10. Throughout all the experiments, we conducted training and testing using the half-precision *float16*.

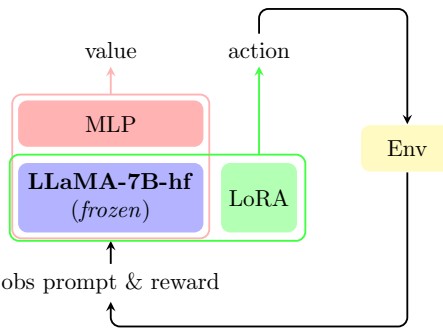

Figure 10: Parameter efficient architecture

## D.1    NETWORK ARCHITECTURE

The same neural network architecture is applied to both the actor and critic networks across TWO-SOME method. In the LLaMA-7B model, additional MLP layers are incorporated into the last transformer block to serve as the critic. The critic's MLPs take the last token of the observation prompt as input and output the estimated value of the observation prompt. The critic utilizes a 3-layer MLP structure, with the number of neurons in each layer being 1024, 512, and 1, respectively. ReLU is employed as the activation function. On the other hand, the actor consists of the frozen LLaMA-7B model with the augmentation of LoRA parameters. The policy optimizer uses AdamW with an epsilon value of 1e-5 and a weight decay of 0. The critic optimizer uses Adam with an epsilon value of 1e-5 and the default weight decay of 0. In our experiments, we found that the default epsilon value of 1e-8 in the policy optimizer's AdamW can lead to unstable training and potential "nan" outputs in Lora.

In the PPO method, the critic network remains a 3-layer MLP architecture with 64 neurons in each layer and a single neuron as the final output. However, the main distinction from TWOSOME lies in the choice of activation function, which is Tanh. Meanwhile, the actor in the PPO method also adopts a 3-layer MLP architecture. The number of neurons in the first two layers is the same as the critic, which is 64. The activation function used is Tanh. The final layer's output depends on the number of actions in the task. For example, in the *Entertainment* task, there are 17 actions, while in the *Food Preparation* task, there are 10 actions. The optimizer uses Adam with an epsilon value of 1e-5 and a weight decay of 0.

## D.2    HYPERPARAMETERS FOR TWOSOME AND PPO

In following subsections, we first list the hyper-parameter candidates used for training TWOSOME via grid search in the corresponding task, and then show the hyper-parameter table with the parameters used by TWOSOME and PPO achieving the best performance. We choose the best performance of each method depending on its final convergence value and convergence efficiency.

Table 10: Overcooked hyper-parameter candidates of TWOSOME for grid search training

| | First group parameters | Second group parameters |
|---|---|---|
| Learning rate pair (policy,critic) | (1e-6, 5e-5), (5e-7, 5e-5), (5e-7, 1e-5) | |
| | (3e-6, 5e-5), (1e-6, 1e-5) | |
| Max episode steps and gamma | (200, 0.99), (50, 0.95) | |
| Total time steps | 1e5, 5e5 | |
| Number of environments | 4 | 4 |
| Number of steps per environment policy rollout | 32 | 128 |
| Number of mini-batches of policy | 32 | 128 |
| Number of mini-batches of critic | 4 | 16 |
| Number of epochs to update policy | 1 | 1 |
| Gradient checkpoint steps | 8 | 32 |
| Surrogate clipping coefficient | 0.2 | 0.2 |
| Coefficient of the entropy | 0.01 | 0.01 |
| Coefficient of the value function | 0.5 | 0.5 |
| Maximum norm for the gradient clipping | 0.5 | 0.5 |
| Target KL divergence threshold | 0.02 | 0.02 |

Table 11: Overcooked hyper-parameter used of TWOSOME in Tomato Salad task

| | |
|---|---|
| Learning rate pair (policy,critic) | (5e-7, 1e-5) |
| Max episode steps and gamma | (200, 0.99) |
| Total time steps | 5e5 |
| Number of environments | 4 |
| Number of steps per environment policy rollout | 32 |
| Number of mini-batches of policy | 32 |
| Number of mini-batches of critic | 4 |
| Number of epochs to update policy | 1 |
| Gradient checkpoint steps | 8 |
| Surrogate clipping coefficient | 0.2 |
| Coefficient of the entropy | 0.01 |
| Coefficient of the value function | 0.5 |
| Maximum norm for the gradient clipping | 0.5 |
| Target KL divergence threshold | 0.02 |

Table 12: Overcooked hyper-parameter used of TWOSOME in Tomato Lettuce Salad task

| | |
|---|---|
| Learning rate pair (policy,critic) | (5e-7, 1e-5) |
| Max episode steps and gamma | (200, 0.99) |
| Total time steps | 5e5 |
| Number of environments | 4 |
| Number of steps per environment policy rollout | 32 |
| Number of mini-batches of policy | 32 |
| Number of mini-batches of critic | 4 |
| Number of epochs to update policy | 1 |
| Gradient checkpoint steps | 8 |
| Surrogate clipping coefficient | 0.2 |
| Coefficient of the entropy | 0.01 |
| Coefficient of the value function | 0.5 |
| Maximum norm for the gradient clipping | 0.5 |
| Target KL divergence threshold | 0.02 |

Table 13: Overcooked hyper-parameter candidates of PPO for grid search training

| | |
|---|---|
| Learning rate | 1e-4, 2.5e-4, 5e-4, 1e-3 |
| Max episode steps and gamma | (200, 0.99), (50, 0.95) |
| Total time steps | 1e5, 5e5 |
| Number of environments | 4 |
| Number of steps per environment policy rollout | 128 |
| Number of mini-batches of policy | 4 |
| Number of epochs to update policy | 4 |
| Surrogate clipping coefficient | 0.2 |
| Coefficient of the entropy | 0.01 |
| Coefficient of the value function | 0.5 |
| Maximum norm for the gradient clipping | 0.5 |
| Target KL divergence threshold | 0.02 |

Table 14: Overcooked hyper-parameter used of PPO in Tomato Salad task

| | |
|---|---|
| Learning rate | 1e-4 |
| Max episode steps and gamma | (200, 0.99) |
| Total time steps | 5e5 |
| Number of environments | 4 |
| Number of steps per environment policy rollout | 128 |
| Number of mini-batches of policy | 4 |
| Number of epochs to update policy | 4 |
| Surrogate clipping coefficient | 0.2 |
| Coefficient of the entropy | 0.01 |
| Coefficient of the value function | 0.5 |
| Maximum norm for the gradient clipping | 0.5 |
| Target KL divergence threshold | 0.02 |

Table 15: Overcooked hyper-parameter used of PPO in Tomato Lettuce Salad task

| | |
|---|---|
| Learning rate | 5e-4 |
| Max episode steps and gamma | (200, 0.99) |
| Total time steps | 5e5 |
| Number of environments | 4 |
| Number of steps per environment policy rollout | 128 |
| Number of mini-batches of policy | 4 |
| Number of epochs to update policy | 4 |
| Surrogate clipping coefficient | 0.2 |
| Coefficient of the entropy | 0.01 |
| Coefficient of the value function | 0.5 |
| Maximum norm for the gradient clipping | 0.5 |
| Target KL divergence threshold | 0.02 |

Table 16: Virtualhome hyper-parameter candidates of TWOSOME for grid search training

| | First | Second | Third | Forth |
|---|---|---|---|---|
| Learning rate pair (policy,critic) | (1e-6, 5e-5), (5e-7, 5e-5), (5e-7, 1e-5) (3e-6, 5e-5), (1e-6, 1e-5) | | | |
| Max episode steps and gamma | (200, 0.99), (50, 0.95) | | | |
| Total time steps | 5e4, 1e5, 2e5 | | | |
| Number of environments | 4 | 4 | 4 | 4 |
| Number of steps per environment policy rollout | 32 | 32 | 32 | 128 |
| Number of mini-batches of policy | 32 | 64 | 64 | 128 |
| Number of mini-batches of critic | 4 | 8 | 4 | 16 |
| Number of epochs to update policy | 1 | 1 | 1 | 1 |
| Gradient checkpoint steps | 8 | 16 | 16 | 32 |
| Surrogate clipping coefficient | 0.2 | 0.2 | 0.2 | 0.2 |
| Coefficient of the entropy | 0.01 | 0.01 | 0.01 | 0.01 |
| Coefficient of the value function | 0.5 | 0.5 | 0.5 | 0.5 |
| Maximum norm for the gradient clipping | 0.5 | 0.5 | 0.5 | 0.5 |
| Target KL divergence threshold | 0.02 | 0.02 | 0.02 | 0.02 |

Table 17: Virtualhome hyper-parameter used of TWOSOME in the task of *Food Preparation*

| | |
|---|---|
| Learning rate pair (policy,critic) | (1e-6, 5e-5) |
| Max episode steps and gamma | (50, 0.95) |
| Total time steps | 5e4 |
| Number of environments | 4 |
| Number of steps per environment policy rollout | 32 |
| Number of mini-batches of policy | 32 |
| Number of mini-batches of critic | 4 |
| Number of epochs to update policy | 1 |
| Gradient checkpoint steps | 8 |
| Surrogate clipping coefficient | 0.2 |
| Coefficient of the entropy | 0.01 |
| Coefficient of the value function | 0.5 |
| Maximum norm for the gradient clipping | 0.5 |
| Target KL divergence threshold | 0.02 |

Table 18: Virtualhome hyper-parameter used of TWOSOME in the task of *Entertainment*

| | |
|---|---|
| Learning rate pair (policy,critic) | (1e-6, 5e-5) |
| Max episode steps and gamma | (50, 0.95) |
| Total time steps | 2e5 |
| Number of environments | 4 |
| Number of steps per environment policy rollout | 32 |
| Number of mini-batches of policy | 32 |
| Number of mini-batches of critic | 4 |
| Number of epochs to update policy | 1 |
| Gradient checkpoint steps | 8 |
| Surrogate clipping coefficient | 0.2 |
| Coefficient of the entropy | 0.01 |
| Coefficient of the value function | 0.5 |
| Maximum norm for the gradient clipping | 0.5 |
| Target KL divergence threshold | 0.02 |

Table 19: Virtualhome hyper-parameter candidates of PPO for grid search training

| | |
|---|---|
| Learning rate | 1e-4, 2.5e-4, 5e-4, 1e-3 |
| Max episode steps and gamma | (200, 0.99), (50, 0.95) |
| Total time steps | 5e4, 1e5, 2e5 |
| Number of environments | 4 |
| Number of steps per environment policy rollout | 128 |
| Number of mini-batches of policy | 4 |
| Number of epochs to update policy | 4 |
| Surrogate clipping coefficient | 0.2 |
| Coefficient of the entropy | 0.01 |
| Coefficient of the value function | 0.5 |
| Maximum norm for the gradient clipping | 0.5 |
| Target KL divergence threshold | 0.02 |

Table 20: Virtualhome hyper-parameter used of PPO in the task of *Food Preparation*

| | |
|---|---|
| Learning rate | 1e-3 |
| Max episode steps and gamma | (50, 0.95) |
| Total time steps | 5e4 |
| Number of environments | 4 |
| Number of steps per environment policy rollout | 128 |
| Number of mini-batches of policy | 4 |
| Number of epochs to update policy | 4 |
| Surrogate clipping coefficient | 0.2 |
| Coefficient of the entropy | 0.01 |
| Coefficient of the value function | 0.5 |
| Maximum norm for the gradient clipping | 0.5 |
| Target KL divergence threshold | 0.02 |

Table 21: Virtualhome hyper-parameter used of PPO in the task of *Entertainment*

| | |
|---|---|
| Learning rate | 1e-3 |
| Max episode steps and gamma | (50, 0.95) |
| Total time steps | 2e5 |
| Number of environments | 4 |
| Number of steps per environment policy rollout | 128 |
| Number of mini-batches of policy | 4 |
| Number of epochs to update policy | 4 |
| Surrogate clipping coefficient | 0.2 |
| Coefficient of the entropy | 0.01 |
| Coefficient of the value function | 0.5 |
| Maximum norm for the gradient clipping | 0.5 |
| Target KL divergence threshold | 0.02 |

# E BEHAVIOR VISUALIZATION

In this section, we display the optimal behaviors learned by TWOSOME with word normalization under all considered domains. We also provide corresponding videos for each task in the Supplementary Materials.

## E.1 VISUALIZATION OF TOMATO SALAD

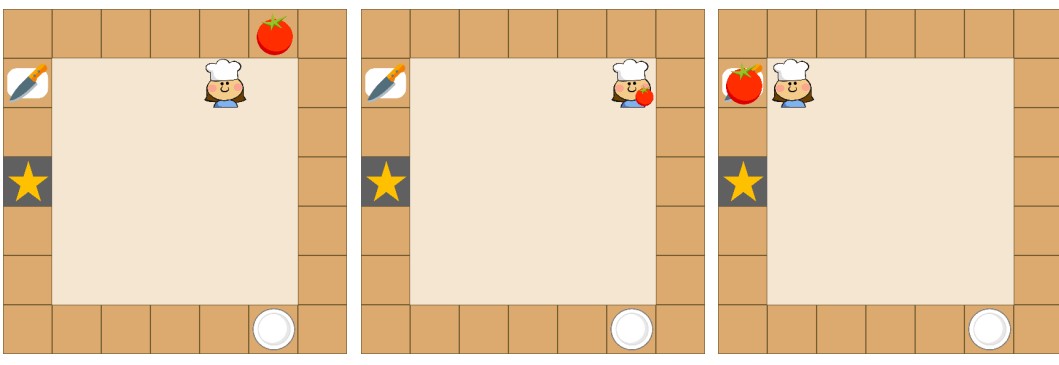

(a) The initial state of the environment, the agent is in the kitchen without grabbing anything.

(b) The agent executes **Get-Tomato**.

(c) After picking up the tomato, the agent executes **Go-Cutting-Board-1**.

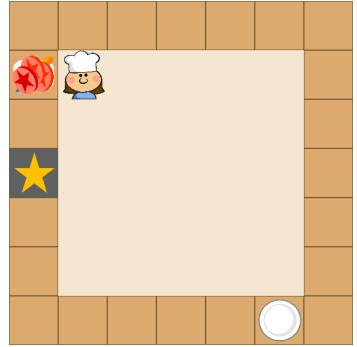 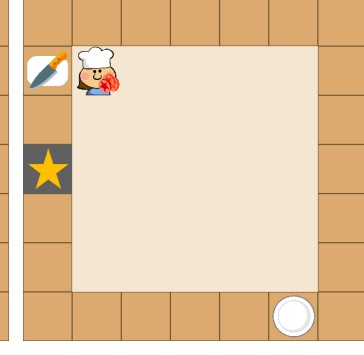 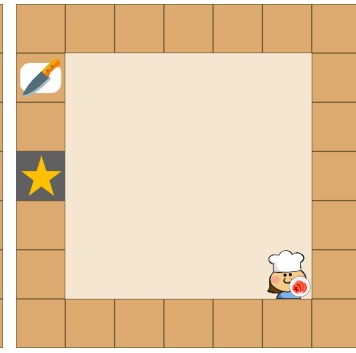

(d) After approaching the cutting board, the agent executes the action **Chop** to chop the tomato.

(e) After chopping the tomato, the agent executes the action **Get-Tomato** to pick up the chopped tomato.

(f) After picking up the chopped tomato, the agent executes the action **Get-Bowl** to take a bowl and place the chopped tomato inside it.

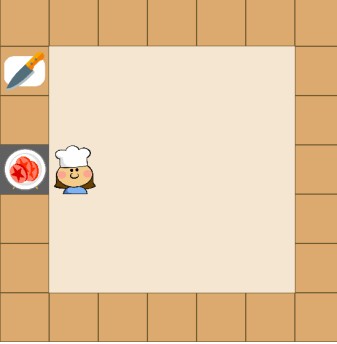

(g) After placing the chopped tomato inside the bowl, the agent executes the action **Deliver** to complete the task.

### E.2 VISUALIZATION OF TOMATO LETTUCE SALAD

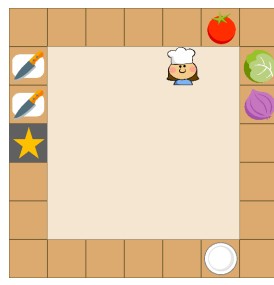

(a) The initial state of the environment, the agent is in the kitchen without grabbing anything.

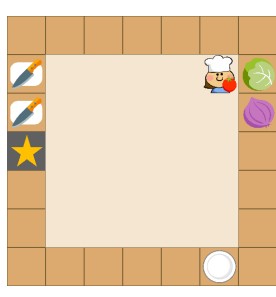

(b) The agent executes *Get-Tomato*.

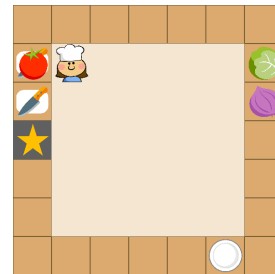

(c) After picking up the tomato, the agent executes *Go-Cutting-Board-1*.

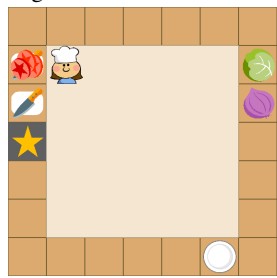

(d) After approaching the cutting board 1, the agent executes the action *Chop* to chop the tomato.

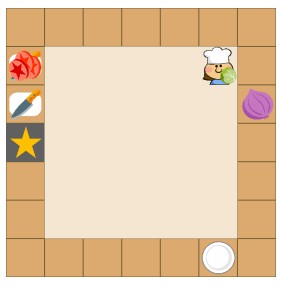

(e) After chopping the tomato, the agent executes *Get-Lettuce*.

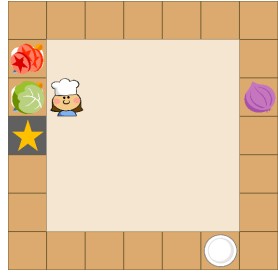

(f) After picking up the lettuce, the agent executes *Go-Cutting-Board-2*.

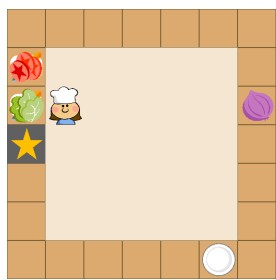

(g) After approaching the cutting board 2, the agent executes the action *Chop* to chop the lettuce.

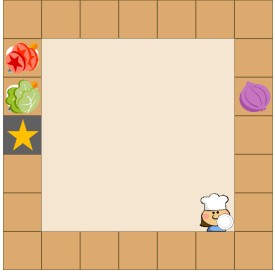

(h) After chopping the lettuce, the agent executes the action *Get-Bowl* to take a bowl.

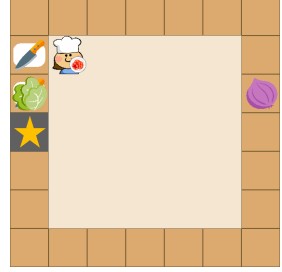

(i) After taking a bowl, the agent executes the action *Get-Tomato* to place the chopped tomato inside the bowl.

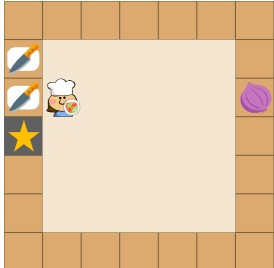

(j) After picking up the chopped tomato, the agent executes *Get-Lettuce* to place the chopped lettuce inside the bowl.

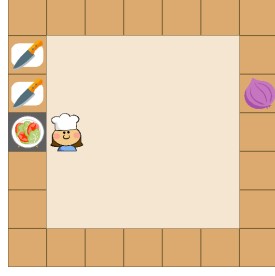

(k) After placing the chopped lettuce inside the bowl, the agent executes the action *Deliver* to complete the task.

### E.3 VISUALIZATION OF *Food Preparation*

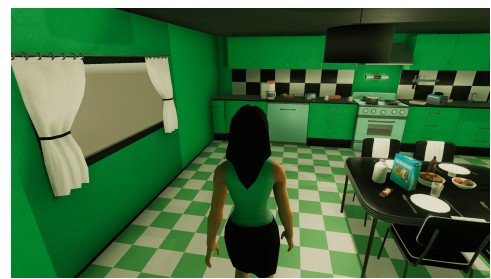

(a) The initial state of the environment, the agent is in the kitchen without grabbing the pancake.

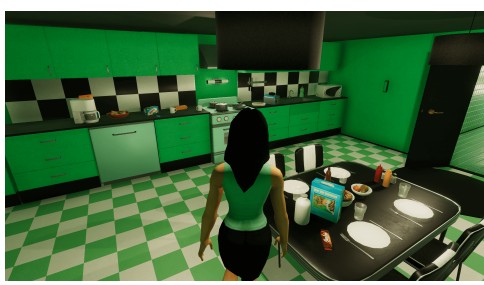

(b) The agent executes ***Walk-Pancake***.

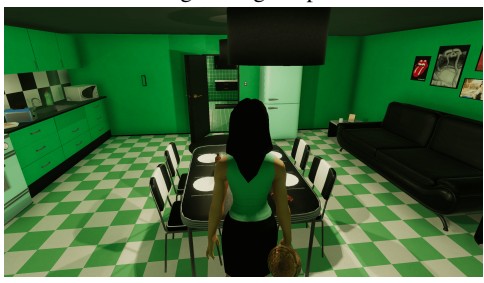

(c) After approaching the pancake, the agent executes ***Grab-Pancake***.

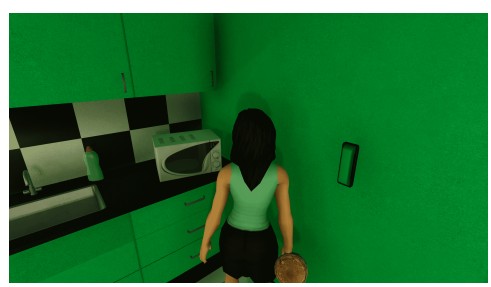

(d) After grabbing the pancake, the agent executes ***Walk-Microwave***.

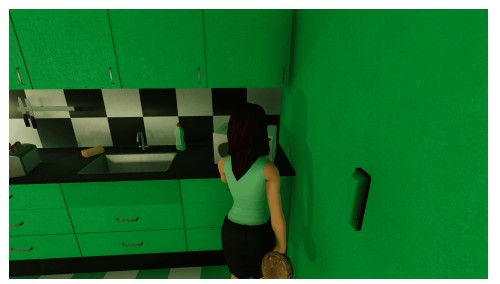

(e) After approaching the microwave, the agent executes ***Open-Microwave***.

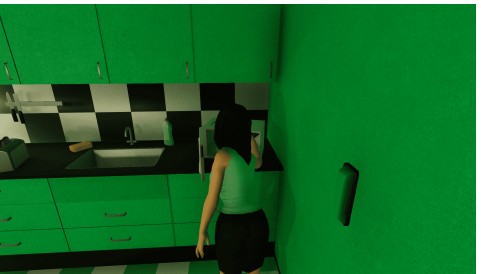

(f) After opening the microwave, the agent executes ***Putin-Pancake-Microwave***.

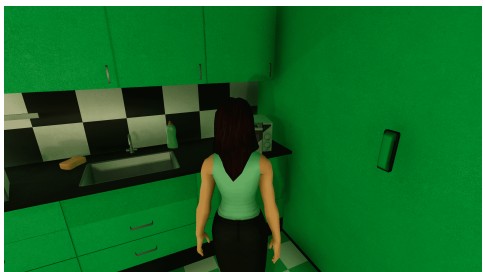

(g) After placing the pancake into the microwave, the agent executes ***Close-Microwave***.

### E.4 VISUALIZATION OF THE *Entertainment*

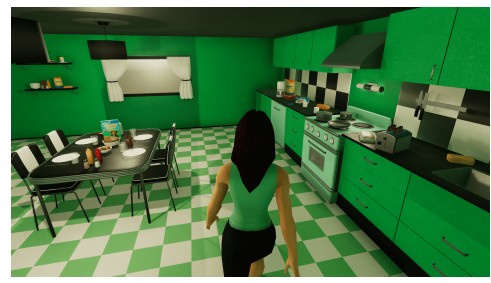

(a) The initial state of the environment, the agent is in the kitchen without grabbing anything.

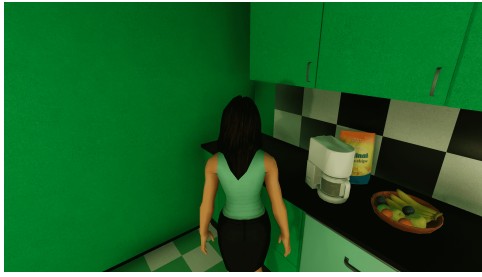

(b) The agent executes **Walk-Chips**.

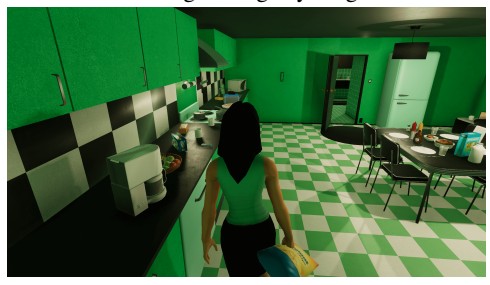

(c) After approaching the chips, the agent executes **Grab-Chips**.

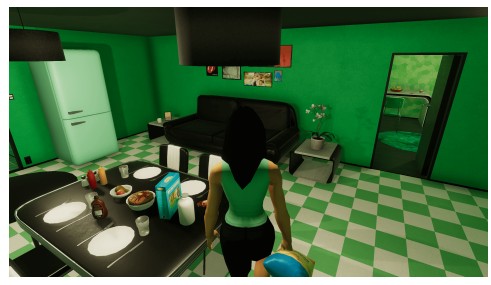

(d) After grabbing the chips, the agent executes **Walk-Milk**.

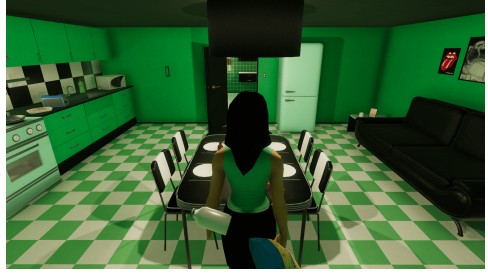

(e) After approaching the milk, the agent executes **Grab-Milk**.

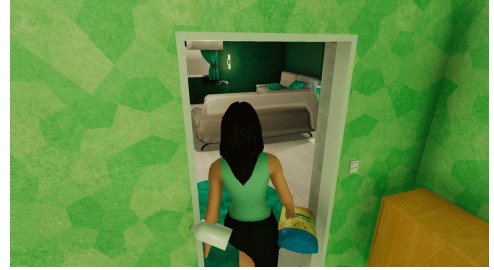

(f) After grabbing the milk, the agent executes **Walk-Livingroom**.

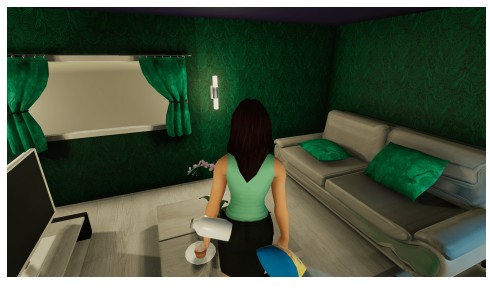

(g) The agent leaves the kitchen and enters the living room, then executes *Walk-Coffeetable*.

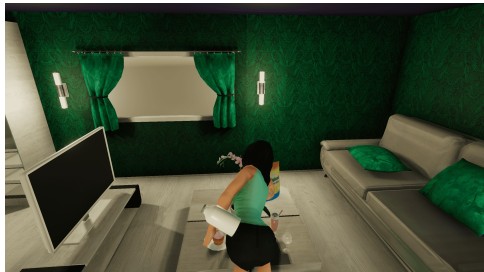

(h) After approaching the coffee table, the agent executes *Putback-Chips-Coffeetable*.

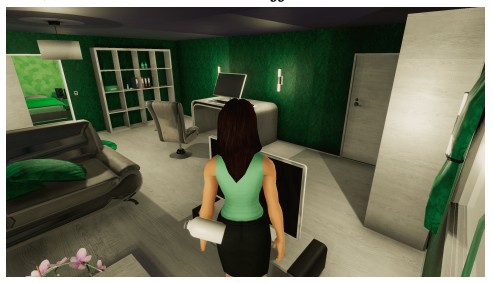

(i) With the chips on the coffee table, the agent executes *Walk-TV*.

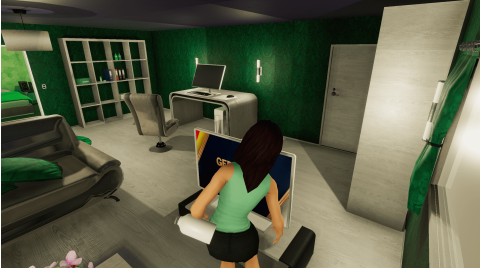

(j) After approaching the television, the agent executes *Switchon-TV*.

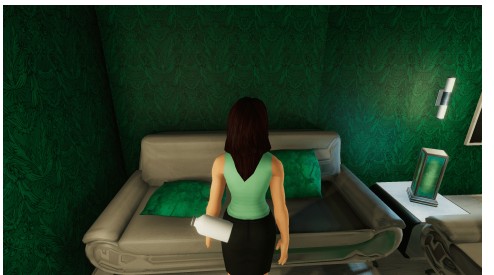

(k) After turning on the television, the agent executes *Walk-Sofa*.

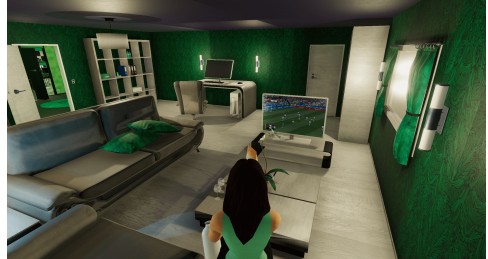

(l) After approaching the sofa, the agent executes the action *Sit-Sofa* to complete the task.

## F    PROMPT DESIGN

In this section, we provide a detailed analysis of the design of the observation prompt and action prompt and their impacts on the probabilities of each action. All the prompts are converted from raw observations and actions by scripts. For each observation shown in the visualization, we first present the observation prompt of the corresponding observation and then show the action prompts, tokens that make up the action prompts, the probability of each token, and the policy of TWOSOME without normalization, TWOSOME with token normalization and TWOSOME with word normalization, sequentially.

### F.1    TOMATO SALAD

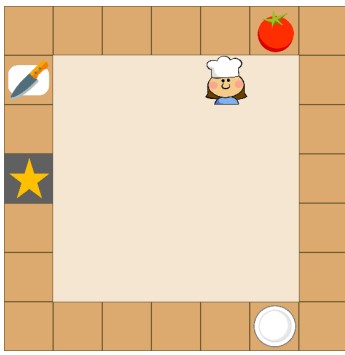

(a) The initial state of the environment, the agent is in the kitchen without grabbing anything.

**Observation prompt in step (a):** There is a fixed cutting board in the room. You notice a tomato on the table. Currently you don't have anything in hand. To serve the dish of a bowl only containing chopped tomato, you should first

Table 22: Action Prompt in Step (a)

| Action | Token | Token Probability | Action Probability of TWOSOME | | |
|---|---|---|---|---|---|
| | | | W/O Norm | Token Norm | Word Norm |
| pick up the tomato | [pick, up, the, tom, ato] | [13.73, 77.67, 72.82, 28.54, 98.87] | 55.24 | 48.34 | 49.61 |
| take the bowl | [take, the, bow, l] | [14.81, 48.69, 24.65, 99.72] | 44.66 | 37.87 | 33.61 |
| walk to the cutting board | [walk, to, the, cutting, board] | [0.06, 42.28, 89.96, 15.79, 96.56] | 0.09 | 13.33 | 16.56 |
| serve nothing | [serve, nothing] | [4.65, 0.0] | 0.01 | 0.16 | 0.19 |
| chop nothing | [ch, op, nothing] | [0.53, 98.64, 0.0] | 0.00 | 0.31 | 0.02 |

Analysis: To ensure a high probability of items that are useful for completing the task like tomato and bowl, we need to mention them in the observation prompt. However, due to the partial observability, the agent cannot see bowl in the current position. So we need to mention the bowl in the task by mentioning *To serve the dish of a bowl*, which can increase the probability of *serve*, too. Because currently there is nothing for the agent to serve and chop, so we can use *nothing* to lower the probability of these two actions. These two actions are actually not necessary to appear in the action prompt list, which is what we do in the VirtualHome environment. To show the performance of our method, which can solve typical decision-making tasks, we do not mask these actions here, for a fair comparison with PPO.

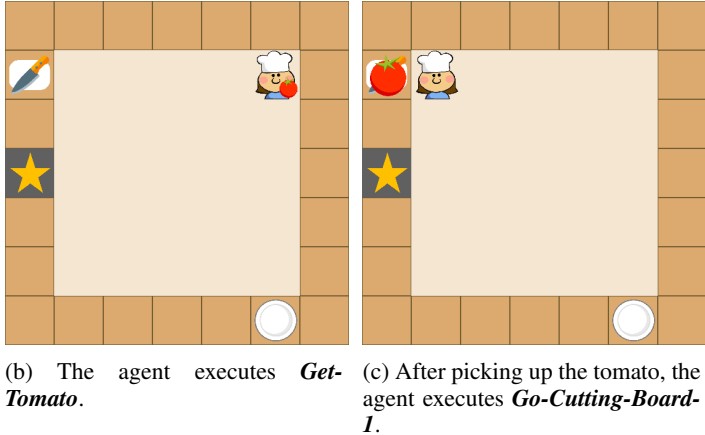

(b) The agent executes **Get-Tomato**.

(c) After picking up the tomato, the agent executes **Go-Cutting-Board-1**.

**Observation prompt step (b):** There is a fixed cutting board in the room. Currently you are carrying an unchopped tomato in hand. To serve the dish of a bowl only containing chopped tomato, you should first

Table 23: Action Prompt in Step (b)

| Action | Token | Token Probability | Action Probability of TWOSOME | | |
| --- | --- | --- | --- | --- | --- |
| | | | W/O Norm | Token Norm | Word Norm |
| pick up the tomato | [pick, up, the, tom, ato] | [0.31, 72.1, 64.94, 20.24, 97.16] | 3.28 | 17.67 | 15.73 |
| take the bowl | [take, the, bow, l] | [3.46, 37.33, 5.29, 99.34] | 7.73 | 14.56 | 10.61 |
| put the tomato on the cutting board | [put, the, tom, ato, on, the, cutting, board] | [11.44, 60.03, 36.08, 96.9, 46.14, 87.94, 53.74, 96.81] | 57.69 | 46.60 | 56.71 |
| serve the dish | [serve, the, d, ish] | [2.98, 57.96, 16.23, 97.9] | 31.29 | 20.66 | 16.90 |
| chop nothing | [ch, op, nothing] | [6.24, 99.45, 0.0] | 0.00 | 0.51 | 0.05 |

**Observation prompt step (c):** There is a fixed cutting board in the room. An unchopped tomato is on the cutting board. Currently you are standing in front of the cutting board without anything in hand. To serve the dish of a bowl only containing chopped tomato, you should first

Table 24: Action Prompt in Step (c)

| Action | Token | Token Probability | Action Probability of TWOSOME | | |
| --- | --- | --- | --- | --- | --- |
| | | | W/O Norm | Token Norm | Word Norm |
| pick up the tomato | [pick, up, the, tom, ato] | [11.26, 78.92, 81.57, 30.95, 98.76] | 38.32 | 32.52 | 36.21 |
| take the bowl | [take, the, bow, l] | [13.38, 52.9, 19.33, 99.75] | 23.59 | 23.81 | 22.42 |
| walk to the cutting board | [walk, to, the, cutting, board] | [0.08, 37.29, 89.38, 37.08, 98.33] | 0.18 | 11.11 | 14.97 |
| serve nothing | [serve, nothing] | [3.23, 0.01] | 0.00 | 0.10 | 0.14 |
| chop the tomato | [ch, op, the, tom, ato] | [4.55, 99.57, 57.2, 88.8, 95.28] | 37.91 | 32.45 | 26.27 |

Analysis: In Step (b), because the agent has already picked up the tomato, the third action *walk to the cutting board* can be replaced to *put the tomato on the cutting board*, which is more contextually appropriate and thus has a high probability. And the agent now can serve the tomato, so the fourth action is replaced with *serve the dish*, though it is not a correct action. LLMs need to learn which dish is the correct dish, namely the alignment. It is worth noting that for TWOSOME without normalization, *serve the dish* also has a high probability, if the agent happens to sample this action several times, it may learn to avoid entering this state and refuse to pick up the tomato, resulting in the divergence.

In Step (c), the tomato is put on the cutting board, so the final action is replaced with *chop the tomato*.

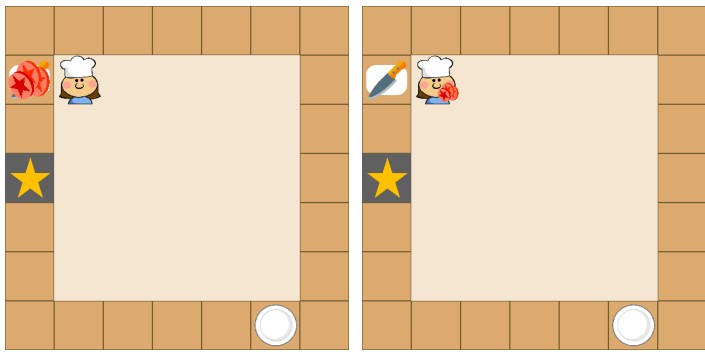

(d) After approaching the cutting board, the agent executes the action **_Chop_** to chop the tomato.

(e) After chopping the tomato, the agent executes the action **_Get-Tomato_** to pick up the chopped tomato.

**Observation prompt step (d):** There is a fixed cutting board in the room. A chopped tomato is on the cutting board. Currently you are standing in front of the cutting board without anything in hand. To serve the dish of a bowl only containing chopped tomato, you should first

Table 25: Action Prompt in Step (d)

| Action | Token | Token Probability | Action Probability of TWOSOME | | |
| --- | --- | --- | --- | --- | --- |
| | | | W/O Norm | Token Norm | Word Norm |
| pick up the tomato | [pick, up, the, tom, ato] | [16.68, 73.75, 82.4, 19.69, 98.68] | 38.27 | 33.49 | 37.31 |
| take the bowl | [take, the, bow, l] | [15.05, 54.08, 34.69, 99.76] | 54.75 | 30.09 | 30.31 |
| walk to the cutting board | [walk, to, the, cutting, board] | [0.12, 36.22, 89.51, 39.64, 98.38] | 0.30 | 12.69 | 17.21 |
| serve nothing | [serve, nothing] | [3.28, 0.01] | 0.00 | 0.11 | 0.15 |
| chop the tomato | [ch, op, the, tom, ato] | [0.76, 98.79, 54.96, 90.12, 92.33] | 6.68 | 23.62 | 15.03 |

**Observation prompt step (e):** There is a fixed cutting board in the room. Currently you are standing in front of the cutting board, carrying a chopped tomato in hand. To serve the dish of a bowl only containing chopped tomato, you should first

Table 26: Action Prompt in Step (e)

| Action | Token | Token Probability | Action Probability of TWOSOME | | |
| --- | --- | --- | --- | --- | --- |
| | | | W/O Norm | Token Norm | Word Norm |
| pick up the tomato | [pick, up, the, tom, ato] | [0.94, 76.54, 73.57, 20.2, 98.13] | 20.00 | 31.26 | 31.87 |
| take the bowl | [take, the, bow, l] | [4.99, 42.45, 13.73, 99.54] | 55.44 | 28.62 | 25.26 |
| walk to the cutting board | [walk, to, the, cutting, board] | [0.19, 38.87, 87.37, 6.89, 95.86] | 0.84 | 16.57 | 23.80 |
| serve the dish | [serve, the, d, ish] | [1.25, 63.08, 15.9, 98.87] | 23.72 | 23.15 | 19.04 |
| chop nothing | [ch, op, nothing] | [0.77, 97.96, 0.0] | 0.00 | 0.40 | 0.03 |

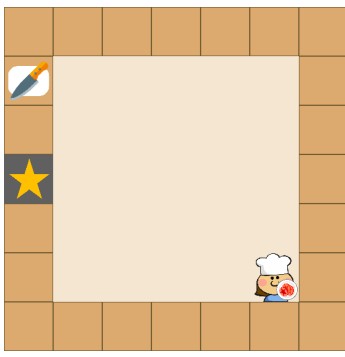

(f) After picking up the chopped tomato, the agent executes the action ***Get-Bowl*** to take a bowl and place the chopped tomato inside it.

**Observation prompt step (f):** There is a fixed cutting board in the room. Currently you are carrying a bowl containing chopped tomato. To serve the dish of a bowl only containing chopped tomato, you should first

Table 27: Action Prompt in Step (f)

| Action | Token | Token Probability | Action Probability of TWOSOME | | |
|---|---|---|---|---|---|
| | | | W/O Norm | Token Norm | Word Norm |
| put the tomato in the bowl | [put, the, tom, ato, in, the, bow, l] | [11.87, 51.27, 5.92, 89.86, 21.92, 70.82, 61.58, 99.5] | 1.16 | 24.12 | 22.45 |
| take the bowl | [take, the, bow, l] | [5.83, 52.13, 51.05, 99.67] | 58.43 | 23.36 | 21.52 |
| put the bowl on the cutting board | [put, the, bow, l, on, the, cutting, board] | [11.87, 51.27, 58.39, 99.63, 28.88, 88.37, 59.89, 97.23] | 19.87 | 34.38 | 40.82 |
| serve the dish | [serve, the, d, ish] | [3.71, 69.93, 21.5, 97.56] | 20.54 | 17.99 | 15.19 |
| chop nothing | [ch, op, nothing] | [0.17, 94.86, 0.0] | 0.00 | 0.15 | 0.01 |

Analysis: Even for the final step, *serve the dish* still has a not high probability, which reflects the ability of LLaMA 7B. LLMs need to learn what is the correct dish to serve and make alignment with the environment.

### F.2 TOMATO LETTUCE SALAD

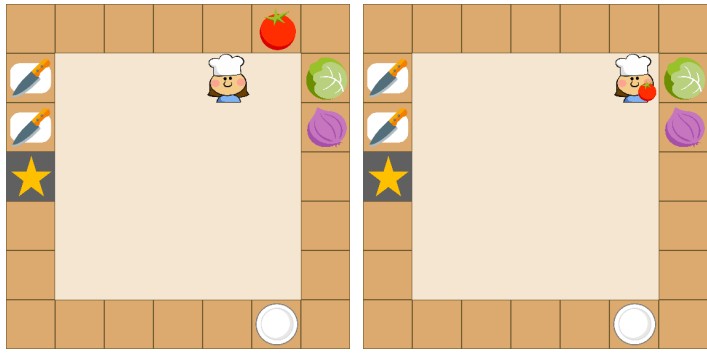

(a) The initial state of the environment, the agent is in the kitchen without grabbing anything.

(b) The agent executes *Get-Tomato*.

**Observation prompt step (a):** There are two fixed cutting boards in the room. You notice a tomato, a lettuce and an onion on the different tables. Currently you don't have anything in hand. To serve the dish of a bowl only containing chopped tomato and lettuce, you should first

Table 28: Action Prompt in Step (a)

| Action | Token | Token Probability | Action Probability of TWOSOME W/O Norm | Token Norm | Word Norm |
|---|---|---|---|---|---|
| pick up the tomato | [pick, up, the, tom, ato] | [7.09, 61.35, 62.92, 31.68, 95.21] | 73.79 | 29.66 | 30.23 |
| pick up the lettuce | [pick, up, the, lett, uce] | [7.09, 61.35, 62.92, 6.1, 99.72] | 14.88 | 21.54 | 20.26 |
| pick up the onion | [pick, up, the, on, ion] | [7.09, 61.35, 62.92, 4.68, 92.45] | 10.58 | 20.11 | 18.60 |
| take the empty bowl | [take, the, empty, bow, l] | [15.35, 50.98, 0.14, 75.08, 98.35] | 0.70 | 11.69 | 9.44 |
| walk to the first cutting board | [walk, to, the, first, cutting, board] | [0.08, 40.31, 84.18, 2.55, 39.84, 94.17] | 0.02 | 9.11 | 11.81 |
| walk to the second cutting board | [walk, to, the, second, cutting, board] | [0.08, 40.31, 84.18, 0.66, 41.69, 94.06] | 0.01 | 7.32 | 9.49 |
| serve nothing | [serve, nothing] | [2.97, 0.01] | 0.01 | 0.10 | 0.13 |
| chop nothing | [ch, op, nothing] | [0.96, 99.17, 0.0] | 0.00 | 0.46 | 0.05 |

**Observation prompt step (b):** There are two fixed cutting boards in the room. You notice a lettuce and an onion on the different tables. Currently you are carrying an unchopped tomato in hand. To serve the dish of a bowl only containing chopped tomato and lettuce, you should first

Table 29: Action Prompt in Step (b)

| Action | Token | Token Probability | Action Probability of TWOSOME W/O Norm | Token Norm | Word Norm |
|---|---|---|---|---|---|
| pick up the tomato | [pick, up, the, tom, ato] | [1.07, 66.26, 68.49, 15.88, 98.79] | 16.23 | 15.69 | 15.24 |
| pick up the lettuce | [pick, up, the, lett, uce] | [1.07, 66.26, 68.49, 30.14, 99.8] | 31.12 | 17.87 | 17.93 |
| pick up the onion | [pick, up, the, on, ion] | [1.07, 66.26, 68.49, 3.2, 97.0] | 3.21 | 11.35 | 10.16 |
| take the empty bowl | [take, the, empty, bow, l] | [6.38, 46.73, 0.06, 65.47, 98.34] | 0.24 | 6.75 | 5.31 |
| put the tomato on the first cutting board | [put, the, tom, ato, on, the, first, cutting, board] | [8.52, 62.29, 15.2, 97.76, 35.8, 81.18, 0.43, 56.07, 96.25] | 0.11 | 17.12 | 20.12 |
| put the tomato on the second cutting board | [put, the, tom, ato, on, the, second, cutting, board] | [8.52, 62.29, 15.2, 97.76, 35.8, 81.18, 0.28, 57.29, 96.58] | 0.07 | 16.34 | 19.09 |
| serve the dish | [serve, the, d, ish] | [4.47, 71.08, 7.44, 97.06] | 49.00 | 14.44 | 12.11 |
| chop nothing | [ch, op, nothing] | [5.07, 99.63, 0.0] | 0.01 | 0.43 | 0.05 |

Analysis: In Step (a), to ensure a high probability of items that are useful for completing the task like tomato, lettuce and bowl, we need to mention them in the observation prompt. However, due to the partial observability, the agent cannot see bowl in the current position. So we need to mention the bowl in the task by mentioning *To serve the dish of a bowl*, which can increase the probability of *serve*, too. Compared with onion, tomato and lettuce appear twice in the observation prompt, thus having higher probabilities. We add the status *empty* to the bowl to lower the probability of holding the plate at the beginning, where the agent can do nothing but to serve the plate to the delivery counter and receive a negative reward.

In Step (b), because the agent has already picked up the tomato, the third action *walk to the first cutting board* can be replaced to *put the tomato on the first cutting board*, which is more contextually appropriate and thus has a high probability. It is worth noting that TWOSOME without the normalization method has an extremely low probability of long action prompts, such as *walk to the first cutting board* , and *put the tomato on the first cutting board*, which is consistent with the disadvantage mentioned in the main paper. Thus, this method has a poor performance in this task.

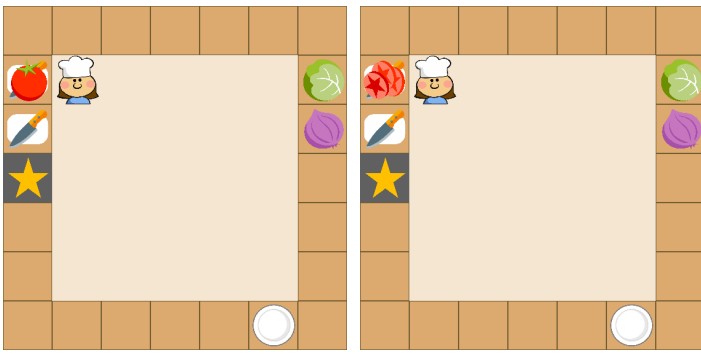

(c) After picking up the tomato, the agent executes **Go-Cutting-Board-1**.

(d) After approaching the cutting board 1, the agent executes the action **Chop** to chop the tomato.

**Observation prompt step (c):** There are two fixed cutting boards in the room. An unchopped tomato is on the first cutting board. Currently you are standing in front of the first cutting board without anything in hand. To serve the dish of a bowl only containing chopped tomato and lettuce, you should first

Table 30: Action Prompt in Step (c)

| Action | Token | Token Probability | Action Probability of TWOSOME | | |
|---|---|---|---|---|---|
| | | | W/O Norm | Token Norm | Word Norm |
| pick up the tomato | [pick, up, the, tom, ato] | [5.33, 77.33, 71.0, 15.17, 97.87] | 25.69 | 19.24 | 19.46 |
| pick up the lettuce | [pick, up, the, lett, uce] | [5.33, 77.33, 71.0, 10.77, 99.9] | 18.61 | 18.04 | 17.95 |
| pick up the onion | [pick, up, the, on, ion] | [5.33, 77.33, 71.0, 0.01, 56.95] | 0.01 | 4.05 | 2.78 |
| take the empty bowl | [take, the, empty, bow, l] | [13.95, 48.59, 0.16, 75.06, 99.64] | 0.47 | 8.64 | 7.16 |
| walk to the first cutting board | [walk, to, the, first, cutting, board] | [0.3, 42.93, 93.27, 5.1, 92.08, 97.85] | 0.33 | 11.16 | 14.82 |
| walk to the second cutting board | [walk, to, the, second, cutting, board] | [0.3, 42.93, 93.27, 55.23, 92.63, 97.89] | 3.60 | 16.62 | 22.07 |
| serve nothing | [serve, nothing] | [4.7, 0.01] | 0.04 | 0.14 | 0.19 |
| chop the tomato | [ch, op, the, tom, ato] | [2.48, 99.61, 63.38, 61.1, 90.68] | 51.26 | 22.09 | 15.57 |

**Observation prompt step (d):** There are two fixed cutting boards in the room. A chopped tomato is on the first cutting board. Currently you are standing in front of the first cutting board without anything in hand. To serve the dish of a bowl only containing chopped tomato and lettuce, you should first

Table 31: Action Prompt in Step (d)

| Action | Token | Token Probability | Action Probability of TWOSOME | | |
|---|---|---|---|---|---|
| | | | W/O Norm | Token Norm | Word Norm |
| pick up the tomato | [pick, up, the, tom, ato] | [7.99, 75.02, 73.47, 9.43, 97.66] | 33.89 | 19.71 | 19.72 |
| pick up the lettuce | [pick, up, the, lett, uce] | [7.99, 75.02, 73.47, 13.72, 99.89] | 50.44 | 21.34 | 21.79 |
| pick up the onion | [pick, up, the, on, ion] | [7.99, 75.02, 73.47, 0.01, 53.76] | 0.02 | 4.45 | 3.07 |
| take the empty bowl | [take, the, empty, bow, l] | [15.99, 51.55, 0.13, 77.09, 99.6] | 0.68 | 9.03 | 7.43 |
| walk to the first cutting board | [walk, to, the, first, cutting, board] | [0.36, 41.54, 92.93, 9.12, 92.14, 97.87] | 0.97 | 13.10 | 17.26 |
| walk to the second cutting board | [walk, to, the, second, cutting, board] | [0.36, 41.54, 92.93, 44.85, 91.56, 97.92] | 4.74 | 17.06 | 22.49 |
| serve nothing | [serve, nothing] | [3.67, 0.01] | 0.04 | 0.13 | 0.17 |
| chop the tomato | [ch, op, the, tom, ato] | [0.47, 99.03, 65.17, 41.12, 88.77] | 9.23 | 15.19 | 8.08 |

Analysis: In Step (c) and (d), TWOSOME without normalization shows high probabilities on the optimal actions for both steps, namely *chop the tomato* and *pick up the lettuce*, due to the fewer tokens these actions have. So if all the optimal actions happen to have very few tokens, this method can achieve very good performance.

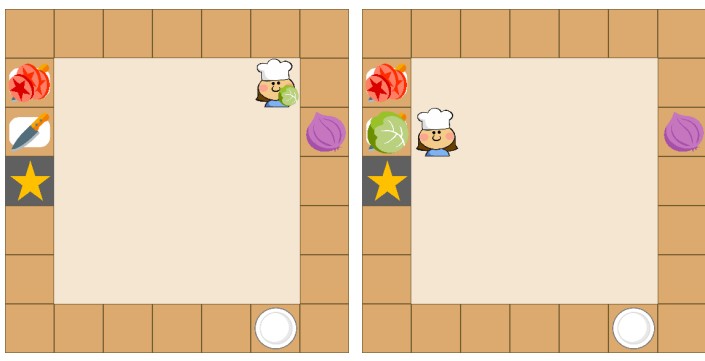

(e) After chopping the tomato, the agent executes **Get-Lettuce**.

(f) After picking up the lettuce, the agent executes **Go-Cutting-Board-2**.

**Observation prompt step (e):** There are two fixed cutting boards in the room. You notice an onion on the table. Currently you are carrying an unchopped lettuce in hand. To serve the dish of a bowl only containing chopped tomato and lettuce, you should first

Table 32: Action Prompt in Step (e)

| Action | Token | Token Probability | Action Probability of TWOSOME | | |
|---|---|---|---|---|---|
| | | | W/O Norm | Token Norm | Word Norm |
| pick up the tomato | [pick, up, the, tom, ato] | [0.7, 69.47, 67.35, 5.15, 94.95] | 2.09 | 11.48 | 10.30 |
| pick up the lettuce | [pick, up, the, lett, uce] | [0.7, 69.47, 67.35, 19.88, 99.87] | 8.50 | 15.20 | 14.63 |
| pick up the onion | [pick, up, the, on, ion] | [0.7, 69.47, 67.35, 8.1, 98.17] | 3.40 | 12.66 | 11.64 |
| take the empty bowl | [take, the, empty, bow, l] | [4.77, 47.86, 0.05, 69.81, 98.91] | 0.10 | 6.23 | 4.80 |
| put the lettuce on the first cutting board | [put, the, lett, uce, on, the, first, cutting, board] | [6.93, 57.59, 47.31, 99.9, 34.26, 85.28, 0.12, 69.79, 97.24] | 0.06 | 16.79 | 19.66 |
| put the lettuce on the second cutting board | [put, the, lett, uce, on, the, second, cutting, board] | [6.93, 57.59, 47.31, 99.9, 34.26, 85.28, 0.27, 69.9, 97.55] | 0.13 | 18.38 | 21.77 |
| serve the dish | [serve, the, d, ish] | [4.6, 70.07, 20.74, 97.9] | 85.71 | 18.76 | 17.14 |
| chop nothing | [ch, op, nothing] | [10.62, 99.71, 0.0] | 0.01 | 0.51 | 0.06 |

**Observation prompt step (f):** There are two fixed cutting boards in the room. A chopped tomato is on the first cutting board. An unchopped lettuce is on the second cutting board. Currently you are standing in front of the second cutting board without anything in hand. To serve the dish of a bowl only containing chopped tomato and lettuce, you should first

Table 33: Action Prompt in Step (f)

| Action | Token | Token Probability | Action Probability of TWOSOME | | |
|---|---|---|---|---|---|
| | | | W/O Norm | Token Norm | Word Norm |
| pick up the tomato | [pick, up, the, tom, ato] | [4.97, 76.56, 67.6, 7.96, 97.22] | 8.06 | 16.67 | 16.20 |
| pick up the lettuce | [pick, up, the, lett, uce] | [4.97, 76.56, 67.6, 16.6, 99.87] | 17.27 | 19.41 | 19.60 |
| pick up the onion | [pick, up, the, on, ion] | [4.97, 76.56, 67.6, 0.01, 30.74] | 0.00 | 3.43 | 2.25 |
| take the empty bowl | [take, the, empty, bow, l] | [15.65, 46.8, 0.08, 73.45, 99.64] | 0.17 | 7.70 | 6.17 |
| walk to the first cutting board | [walk, to, the, first, cutting, board] | [0.18, 49.84, 94.62, 61.28, 93.27, 98.43] | 1.93 | 16.15 | 21.43 |
| walk to the second cutting board | [walk, to, the, second, cutting, board] | [0.18, 49.84, 94.62, 4.99, 93.69, 98.46] | 0.16 | 10.64 | 14.12 |
| serve nothing | [serve, nothing] | [4.77, 0.01] | 0.02 | 0.14 | 0.18 |
| chop the lettuce | [ch, op, the, lett, uce] | [3.39, 98.94, 69.9, 76.36, 99.83] | 72.38 | 25.85 | 20.05 |

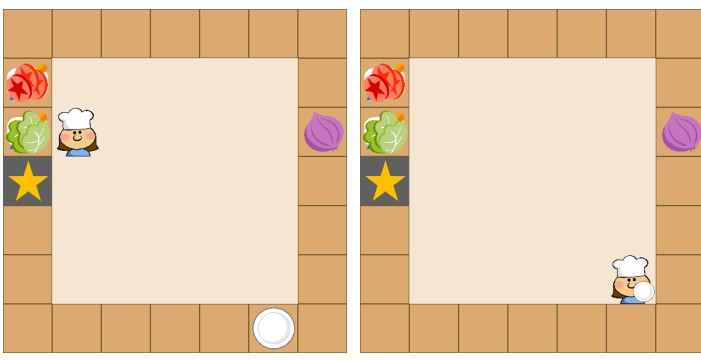

(g) After approaching the cutting board 2, the agent executes the action ***Chop*** to chop the lettuce.

(h) After chopping the lettuce, the agent executes the action ***Get-Bowl*** to take a bowl.

**Observation prompt step (g):** There are two fixed cutting boards in the room. A chopped tomato is on the first cutting board. A chopped lettuce is on the second cutting board. Currently you are standing in front of the second cutting board without anything in hand. To serve the dish of a bowl only containing chopped tomato and lettuce, you should first

Table 34: Action Prompt in Step (g)

| Action | Token | Token Probability | Action Probability of TWOSOME | | |
|---|---|---|---|---|---|
| | | | W/O Norm | Token Norm | Word Norm |
| pick up the tomato | [pick, up, the, tom, ato] | [6.34, 77.54, 68.8, 10.22, 97.71] | 31.69 | 19.38 | 19.38 |
| pick up the lettuce | [pick, up, the, lett, uce] | [6.34, 77.54, 68.8, 12.52, 99.88] | 39.69 | 20.27 | 20.50 |
| pick up the onion | [pick, up, the, on, ion] | [6.34, 77.54, 68.8, 0.01, 31.62] | 0.01 | 3.74 | 2.48 |
| take the empty bowl | [take, the, empty, bow, l] | [18.07, 45.56, 0.09, 77.65, 99.69] | 0.54 | 8.59 | 7.01 |
| walk to the first cutting board | [walk, to, the, first, cutting, board] | [0.27, 49.33, 94.7, 57.71, 93.07, 98.53] | 6.29 | 17.89 | 23.78 |
| walk to the second cutting board | [walk, to, the, second, cutting, board] | [0.27, 49.33, 94.7, 5.63, 93.91, 98.55] | 0.62 | 12.16 | 16.16 |
| serve nothing | [serve, nothing] | [3.9, 0.01] | 0.03 | 0.11 | 0.14 |
| chop the lettuce | [ch, op, the, lett, uce] | [0.56, 98.19, 61.88, 66.44, 99.82] | 21.12 | 17.87 | 10.54 |

**Observation prompt step (h):** There are two fixed cutting boards in the room. Currently you are carrying a bowl in hand. To serve the dish of a bowl only containing chopped tomato and lettuce, you should first

Table 35: Action Prompt in Step (h)

| Action | Token | Token Probability | Action Probability of TWOSOME | | |
|---|---|---|---|---|---|
| | | | W/O Norm | Token Norm | Word Norm |
| put the tomato in the bowl | [put, the, tom, ato, in, the, bow, l] | [20.52, 56.65, 1.22, 92.99, 7.5, 66.35, 56.18, 99.37] | 0.90 | 15.01 | 13.52 |
| put the lettuce in the bowl | [put, the, lett, uce, in, the, bow, l] | [20.52, 56.65, 2.39, 99.84, 23.52, 68.19, 43.22, 99.31] | 4.68 | 18.44 | 17.80 |
| put the onion in the bowl | [put, the, on, ion, in, the, bow, l] | [20.52, 56.65, 0.03, 65.18, 10.4, 73.74, 34.36, 99.26] | 0.02 | 9.05 | 6.89 |
| take the empty bowl | [take, the, empty, bow, l] | [9.0, 36.86, 0.3, 79.57, 98.48] | 1.89 | 8.09 | 6.95 |
| put the bowl on the first cutting board | [put, the, bow, l, on, the, first, cutting, board] | [20.72, 56.65, 59.31, 99.41, 52.52, 86.8, 0.42, 64.3, 95.3] | 1.99 | 18.89 | 22.85 |
| put the bowl on the second cutting board | [put, the, bow, l, on, the, second, cutting, board] | [20.72, 56.65, 59.31, 99.41, 52.52, 86.8, 0.18, 62.49, 95.96] | 0.85 | 17.19 | 20.55 |
| serve the dish | [serve, the, d, ish] | [2.96, 64.42, 21.04, 91.06] | 89.67 | 13.23 | 11.43 |
| chop nothing | [ch, op, nothing] | [0.12, 95.65, 0.0] | 0.00 | 0.10 | 0.01 |

Analysis: In Step (g), because the word *empty* has an extremely low probability, TWOSOME without normalization agent struggles with getting the bowl.

In Step (h), because other action prompts happen to be relatively long, though the probability of *serve* is still low, the probability of *serve the dish* is extremely high compared with other actions for TWOSOME without the normalization method. This step also explains why this method cannot have good performance in this task.

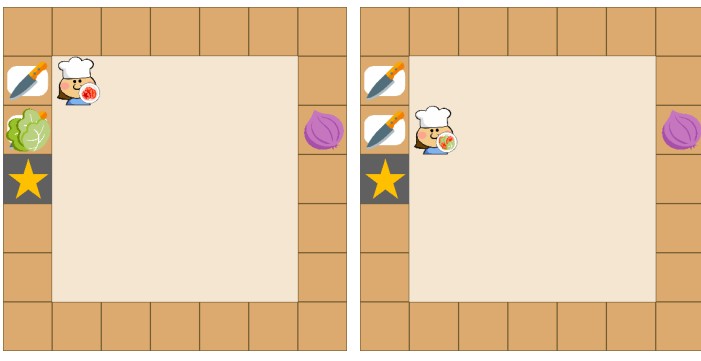

(i) After taking a bowl, the agent executes the action **Get-Tomato** to place the chopped tomato inside the bowl.

(j) After picking up the chopped tomato, the agent executes **Get-Lettuce** to place the chopped lettuce inside the bowl.

**Observation prompt step (i):** There are two fixed cutting boards in the room. A chopped lettuce is on the second cutting board. Currently you are standing in front of the first cutting board, carrying a bowl containing chopped tomato in hand. To serve the dish of a bowl only containing chopped tomato and lettuce, you should first

Table 36: Action Prompt in Step (i)

| Action | Token | Token Probability | Action Probability of TWOSOME | | |
|---|---|---|---|---|---|
| | | | W/O Norm | Token Norm | Word Norm |
| put the tomato in the bowl | [put, the, tom, ato, in, in, the, bow, l] | [10.16, 69.12, 4.23, 96.87, 25.63, 81.21, 65.79, 99.59] | 1.29 | 14.67 | 13.56 |
| put the lettuce in the bowl | [put, the, lett, uce, in, in, the, bow, l] | [10.16, 69.12, 27.55, 99.92, 9.05, 73.3, 55.06, 99.6] | 2.31 | 15.79 | 14.95 |
| put the onion in the bowl | [put, the, on, ion, in, in, the, bow, l] | [10.16, 69.12, 0.01, 30.38, 11.58, 75.65, 42.65, 99.54] | 0.00 | 4.87 | 3.12 |
| take the bowl | [take, the, bow, l] | [8.55, 54.69, 39.66, 99.76] | 60.81 | 14.43 | 13.26 |
| put the bowl on the first cutting board | [put, the, bow, l, on, the, first, cutting, board] | [10.16, 69.42, 42.7, 99.75, 36.04, 90.13, 19.9, 93.96, 98.81] | 5.92 | 19.39 | 22.75 |
| put the bowl on the second cutting board | [put, the, bow, l, on, the, second, cutting, board] | [10.16, 69.42, 42.7, 99.75, 36.04, 90.13, 19.29, 92.64, 98.65] | 5.65 | 19.29 | 22.62 |
| serve the dish | [serve, the, d, ish] | [4.54, 76.52, 21.19, 99.23] | 24.01 | 11.44 | 9.73 |
| chop nothing | [ch, op, nothing] | [0.49, 98.79, 0.0] | 0.00 | 0.12 | 0.01 |

**Observation prompt step (j):** There are two fixed cutting boards in the room. Currently you are standing in front of the second cutting board, carrying a bowl containing chopped tomato and lettuce in hand. To serve the dish of a bowl only containing chopped tomato and lettuce, you should first

Table 37: Action Prompt in Step (j)

| Action | Token | Token Probability | Action Probability of TWOSOME | | |
|---|---|---|---|---|---|
| | | | W/O Norm | Token Norm | Word Norm |
| put the tomato in the bowl | [put, the, tom, ato, in, in, the, bow, l] | [12.59, 61.53, 2.4, 97.88, 3.03, 68.96, 58.64, 99.42] | 0.08 | 11.26 | 9.45 |
| put the lettuce in the bowl | [put, the, lett, uce, in, in, the, bow, l] | [12.59, 61.53, 3.25, 99.9, 20.56, 71.54, 52.21, 99.49] | 0.69 | 14.76 | 13.55 |
| put the onion in the bowl | [put, the, on, ion, in, in, the, bow, l] | [12.59, 61.53, 0.01, 60.24, 7.82, 71.34, 34.39, 99.43] | 0.00 | 5.65 | 3.77 |
| take the bowl | [take, the, bow, l] | [7.17, 47.24, 53.19, 99.74] | 64.48 | 15.75 | 14.78 |
| put the bowl on the first cutting board | [put, the, bow, l, on, the, first, cutting, board] | [12.69, 61.52, 59.05, 99.71, 39.62, 87.89, 6.33, 83.94, 98.12] | 2.99 | 19.57 | 23.26 |
| put the bowl on the second cutting board | [put, the, bow, l, on, the, second, cutting, board] | [12.69, 61.52, 59.05, 99.71, 39.62, 87.89, 7.82, 87.65, 98.27] | 3.87 | 20.13 | 24.01 |
| serve the dish | [serve, the, d, ish] | [3.39, 70.08, 33.37, 98.12] | 27.89 | 12.77 | 11.18 |
| chop nothing | [ch, op, nothing] | [0.15, 98.0, 0.0] | 0.00 | 0.10 | 0.01 |

Analysis: In both Step (i) and Step (j), short actions prompts, *take the bowl* and *serve the dish* still dominates the probabilities, making it difficult for TWOSOME without normalization to sample optimal action and learn good policy.

### F.3 FOOD PREPARATION

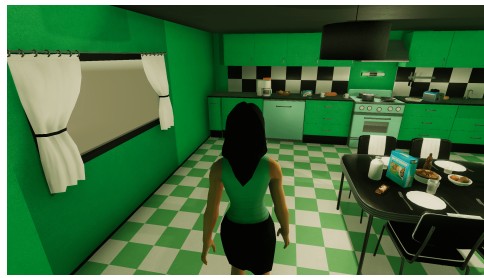 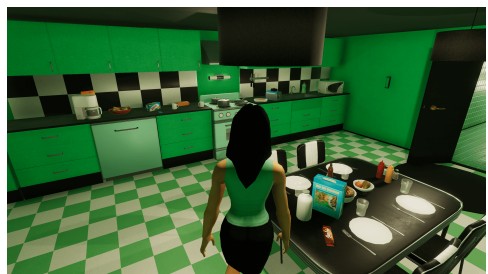

(a) The initial state of the environment, the agent is in the kitchen without grabbing the pancake.

(b) The agent executes ***Walk-Pancake***.

**Observation prompt step (a):** There are four rooms: the kitchen, bathroom, bedroom, and living room. You are in the kitchen. You notice pancake and microwave. Currently, you are not grabbing anything in hand. The pancake and the microwave are not within your immediate reach. The microwave is not opened. In order to heat up the pancake in the microwave, your next step is to

Table 38: Action Prompt in Step (a)

| Action | Token | Token Probability | Action Probability of TWOSOME | | |
|---|---|---|---|---|---|
| | | | W/O Norm | Token Norm | Word Norm |
| walk to the living room | [walk, to, the, living, room] | [0.06, 41.87, 87.8, 4.58, 97.74] | 2.06 | 13.84 | 24.51 |
| walk to the bathroom | [walk, to, the, bath, room] | [0.06, 41.87, 87.8, 1.19, 99.3] | 0.54 | 10.61 | 9.84 |
| walk to the bedroom | [walk, to, the, bed, room] | [0.06, 41.87, 87.8, 0.59, 98.43] | 0.27 | 9.20 | 8.24 |
| reach for the pancake | [reach, for, the, pan, ca, ke] | [1.31, 24.77, 76.49, 14.72, 99.0, 100.0] | 80.76 | 37.56 | 34.36 |
| move to the microwave | [move, to, the, mic, row, ave] | [0.19, 8.96, 85.27, 50.8, 99.85, 99.04] | 16.37 | 28.79 | 23.05 |

**Observation prompt step (b):** There are four rooms: the kitchen, bathroom, bedroom, and living room. You are in the kitchen. You notice pancake and microwave. Currently, you are not grabbing anything in hand. The pancake is within your immediate reach. The microwave is not opend. In order to heat up the pancake in the microwave, your next step is to

Table 39: Action Prompt in Step (b)

| Action | Token | Token Probability | Action Probability of TWOSOME | | |
|---|---|---|---|---|---|
| | | | W/O Norm | Token Norm | Word Norm |
| walk to the living room | [walk, to, the, living, room] | [0.05, 37.46, 86.09, 5.58, 98.02] | 0.12 | 12.14 | 21.74 |
| walk to the bathroom | [walk, to, the, bath, room] | [0.05, 37.46, 86.09, 0.93, 99.3] | 0.02 | 8.51 | 7.80 |
| walk to the bedroom | [walk, to, the, bed, room] | [0.05, 37.46, 86.09, 0.42, 98.47] | 0.01 | 7.26 | 6.39 |
| move to the microwave | [move, to, the, mic, row, ave] | [0.17, 9.25, 84.83, 60.45, 99.8, 99.12] | 1.04 | 25.63 | 20.86 |
| grab the pancake | [grab, the, pan, ca, ke] | [2.08, 74.95, 47.42, 99.49, 99.99] | 98.81 | 46.46 | 43.22 |

Analysis: In the *Food Preparation* task, to ensure a high probability of the items necessary for task completion, we increase the loglikelihood of their occurrence by mentioning task-related objects observed in the current room. For example, in the kitchen, we use *You notice a pancake and a microwave* to raise the chances of the pancake and microwave appearing. To distinguish the probability of each action, our design utilizes different phrases. For approaching a room, we use *walk to*. For approaching food, we use *reach for*. And for approaching furniture, we use *move to*. This differentiation helps to convey the varying probabilities associated with each action.

In Step (a), because the agent is currently not next to the pancake, there is no action available in the options for grabbing objects. To increase the probability of approaching the pancake, we emphasize in the observation prompt that it is *not within your immediate reach* and we also use the word *reach* in the available actions to raise the probability.

In Step (b), as the agent is currently close to the pancake, the action *grab the pancake* is included, while the action *reach for the pancake* is removed from the options.

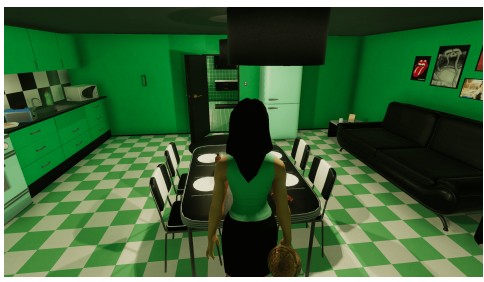 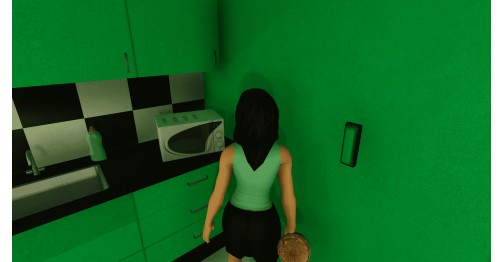

(c) After approaching the pancake, the agent executes *Grab-Pancake*.

(d) After grabbing the pancake, the agent executes *Walk-Microwave*.

**Observation prompt step (c):** There are four rooms: the kitchen, bathroom, bedroom, and living room. You are in the kitchen. You notice pancake and microwave. Currently, you have grabbed the pancake in hand. The microwave is not within your immediate reach. The microwave is not opend. In order to heat up the pancake in the microwave, your next step is to

Table 40: Action Prompt in Step (c)

| Action | Token | Token Probability | Action Probability of TWOSOME | | |
| --- | --- | --- | --- | --- | --- |
| | | | W/O Norm | Token Norm | Word Norm |
| walk to the living room | [walk, to, the, living, room] | [0.06, 42.72, 89.77, 5.7, 98.03] | 5.99 | 15.80 | 28.45 |
| walk to the bathroom | [walk, to, the, bath, room] | [0.06, 42.72, 89.77, 0.77, 99.35] | 0.82 | 10.63 | 9.85 |
| walk to the bedroom | [walk, to, the, bed, room] | [0.06, 42.72, 89.77, 0.48, 98.72] | 0.51 | 9.66 | 8.74 |
| reach for the pancake | [reach, for, the, pan, ca, ke] | [1.31, 21.15, 80.7, 2.52, 98.01, 99.99] | 26.78 | 29.56 | 23.51 |
| move to the microwave | [move, to, the, mic, row, ave] | [0.19, 14.16, 87.89, 59.33, 99.87, 99.29] | 65.89 | 34.35 | 29.45 |

**Observation prompt step (d):** There are four rooms: the kitchen, bathroom, bedroom, and living room. You are in the kitchen. You notice pancake and microwave. Currently, you have grabbed the pancake in hand. The microwave is within your immediate reach. The microwave is not opend. In order to heat up the pancake in the microwave, your next step is to

Table 41: Action Prompt in Step (d)

| Action | Token | Token Probability | Action Probability of TWOSOME | | |
| --- | --- | --- | --- | --- | --- |
| | | | W/O Norm | Token Norm | Word Norm |
| walk to the living room | [walk, to, the, living, room] | [0.03, 36.68, 89.09, 1.12, 98.06] | 0.00 | 3.15 | 4.43 |
| walk to the bathroom | [walk, to, the, bath, room] | [0.03, 36.68, 89.09, 0.16, 99.3] | 0.00 | 2.14 | 1.37 |
| walk to the bedroom | [walk, to, the, bed, room] | [0.03, 36.68, 89.09, 0.08, 98.13] | 0.00 | 1.84 | 1.14 |
| reach for the pancake | [reach, for, the, pan, ca, ke] | [0.45, 24.79, 79.23, 2.25, 97.99, 99.99] | 0.00 | 8.12 | 4.63 |
| put the pancake in the microwave | [put, the, pan, ca, ke, in, the, mic, row, ave] | [0.3, 81.66, 91.57, 99.41, 99.99, 55.36, 90.36, 96.73, 99.77, 99.27] | 0.23 | 25.02 | 22.34 |
| open the microwave | [open, the, mic, row, ave] | [59.42, 82.24, 96.79, 99.74, 99.23] | 98.70 | 42.55 | 54.14 |
| close the microwave | [close, the, mic, row, ave] | [0.67, 81.83, 92.87, 99.8, 99.1] | 1.06 | 17.19 | 11.96 |

Analysis: In Step (c) and Step (d), we control the probability of the agent approaching the microwave and opening it by describing the current state of whether the agent is grabbing the pancake or not.

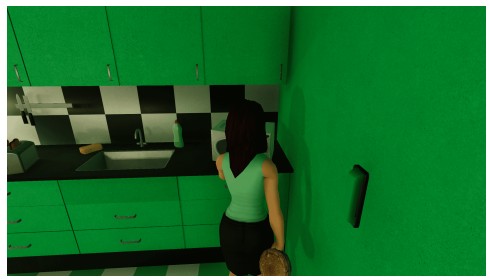
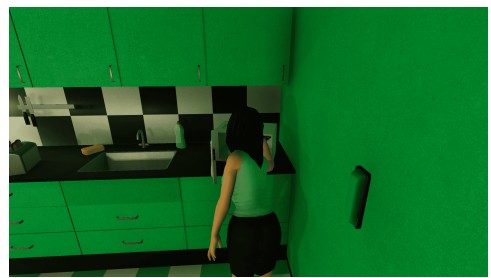

(e) After approaching the microwave, the agent executes ***Open-Microwave***.

(f) After opening the microwave, the agent executes ***Put-in-Pancake-Microwave***.

**Observation prompt step (e):** There are four rooms: the kitchen, bathroom, bedroom, and living room. You are in the kitchen. You notice pancake and microwave. Currently, you have grabbed the pancake in hand. The microwave is within your immediate reach. The microwave is opened. In order to heat up the pancake in the microwave, your next step is to

Table 42: Action Prompt in Step (e)

| Action | Token | Token Probability | Action Probability of TWOSOME | | |
|---|---|---|---|---|---|
| | | | W/O Norm | Token Norm | Word Norm |
| walk to the living room | [walk, to, the, living, room] | [0.14, 33.01, 90.28, 0.35, 98.68] | 0.00 | 3.18 | 4.65 |
| walk to the bathroom | [walk, to, the, bath, room] | [0.14, 33.01, 90.28, 0.06, 99.67] | 0.00 | 2.25 | 1.55 |
| walk to the bedroom | [walk, to, the, bed, room] | [0.14, 33.01, 90.28, 0.04, 97.63] | 0.00 | 2.05 | 1.37 |
| reach for the pancake | [reach, for, the, pan, ca, ke] | [0.54, 31.01, 81.89, 10.92, 99.39, 100.0] | 0.10 | 10.71 | 7.51 |
| put the pancake in the microwave | [put, the, pan, ca, ke, in, the, mic, row, ave] | [2.96, 83.64, 92.77, 99.74, 100.0, 56.31, 93.23, 95.9, 99.93, 99.74] | 7.33 | 29.73 | 32.27 |
| open the microwave | [open, the, mic, row, ave] | [16.93, 86.19, 85.44, 99.92, 99.66] | 79.13 | 30.61 | 33.88 |
| close the microwave | [close, the, mic, row, ave] | [2.56, 87.98, 94.12, 99.92, 99.62] | 13.44 | 21.47 | 18.76 |

**Observation prompt step (f):** There are four rooms: the kitchen, bathroom, bedroom, and living room. You are in the kitchen. You notice pancake and microwave. The microwave is opened. In order to heat up the pancake in the microwave, your next step is to

Table 43: Action Prompt in Step (f)

| Action | Token | Token Probability | Action Probability of TWOSOME | | |
|---|---|---|---|---|---|
| | | | W/O Norm | Token Norm | Word Norm |
| walk to the living room | [walk, to, the, living, room] | [0.05, 35.97, 86.86, 7.82, 98.96] | 0.01 | 7.44 | 10.70 |
| walk to the bathroom | [walk, to, the, bath, room] | [0.05, 35.97, 86.86, 2.31, 99.56] | 0.00 | 5.84 | 4.49 |
| walk to the bedroom | [walk, to, the, bed, room] | [0.05, 35.97, 86.86, 1.3, 99.07] | 0.00 | 5.20 | 3.89 |
| open the microwave | [open, the, mic, row, ave] | [14.45, 81.0, 64.52, 99.9, 99.4] | 60.07 | 42.43 | 43.21 |
| close the microwave | [close, the, mic, row, ave] | [6.47, 85.02, 91.19, 99.88, 99.45] | 39.91 | 39.10 | 37.71 |

Analysis: In Step (e), despite the microwave being already open, the action *open the microwave* still has the highest probability, while the optimal action *put the pancake in the microwave* is just slightly lower. During the training process, the agent will gradually increase the probability of sampling the optimal action. It is worth noting that for TWOSOME without normalization, *open the microwave* remains excessively high, if the agent happens to sample this action several times, it may learn to avoid entering this state and refuse to *put the pancake in the microwave*, resulting in the divergence.

In Step (f), *open the microwave* still has the highest probability, while the optimal action *close the microwave* is the second highest, which reflects the ability of LLaMA 7B. LLMs need to learn what is the correct open and close and make alignment with the environment.

## F.4 ENTERTAINMENT

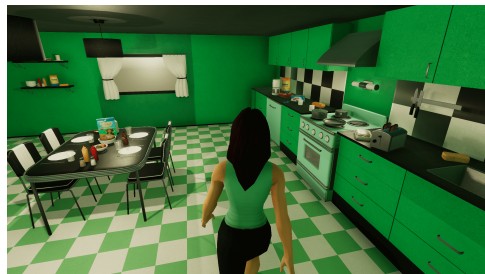

(a) The initial state of the environment, the agent is in the kitchen without grabbing anything.

**Observation prompt step (a):** There are four rooms: the kitchen, bathroom, bedroom, and living room. You are in the kitchen and notice chips and milk. But they are not within your immediate reach. Currently, you are not grabbing anything in hand. In order to enjoy the chips and the milk while watching TV, your next step is to

Table 44: Action Prompt in Step (a)

| Action | Token | Token Probability | Action Probability of TWOSOME | | |
|---|---|---|---|---|---|
| | | | W/O Norm | Token Norm | Word Norm |
| walk to the living room | [walk, to, the, living, room] | [3.79, 48.38, 81.0, 11.0, 98.36] | 4.13 | 17.78 | 20.89 |
| walk to the bathroom | [walk, to, the, bath, room] | [3.79, 48.38, 81.0, 20.1, 99.5] | 7.62 | 20.11 | 17.66 |
| walk to the bedroom | [walk, to, the, bed, room] | [3.79, 48.38, 81.0, 5.98, 98.92] | 2.26 | 15.76 | 13.02 |
| reach for the chips | [reach, for, the, ch, ips] | [16.71, 48.0, 53.21, 72.95, 99.96] | 79.94 | 32.17 | 31.77 |
| reach for the milk | [reach, for, the, milk] | [16.7, 48.12, 52.9, 5.54] | 6.05 | 14.19 | 16.66 |

**Ablation Observation prompt step (a):** There are four rooms: the kitchen, bathroom, bedroom, and living room. You are in the kitchen and notice chips and milk. But they are not close to you. Currently, you are not grabbing anything in hand. In order to enjoy the chips and the milk while watching TV, your next step is to

Table 45: Ablation Action Prompt in Step (a)

| Action | Token | Token Probability | Action Probability of TWOSOME | | |
|---|---|---|---|---|---|
| | | | W/O Norm | Token Norm | Word Norm |
| walk to the living room | [walk, to, the, living, room] | [3.87, 44.14, 80.22, 9.86, 98.81] | 11.26 | 20.19 | 24.45 |
| walk to the bathroom | [walk, to, the, bath, room] | [3.87, 44.14, 80.22, 14.11, 99.4] | 16.21 | 21.71 | 19.24 |
| walk to the bedroom | [walk, to, the, bed, room] | [3.87, 44.14, 80.22, 5.53, 98.5] | 6.30 | 17.97 | 15.19 |
| reach for the chips | [reach, for, the, ch, ips] | [6.48, 23.32, 64.79, 74.24, 99.94] | 61.26 | 28.32 | 26.82 |
| reach for the milk | [reach, for, the, milk] | [6.49, 23.32, 64.82, 5.99] | 4.96 | 11.81 | 14.31 |

Analysis: In the task of *Entertainment*, to ensure a high probability of the items necessary for task completion, we increase the loglikelihood of their occurrence by mentioning task-related objects observed in the current room. For example, in the kitchen, we use *notice chips and milk* to raise the chances of the chips and milk appearing, in the living room, we use *notice a coffee table, a TV and a sofa* to raise the chances of the coffee table, TV and sofa appearing. To distinguish the probability of each action, our design utilizes different phrases. For approaching a room, we use *walk to*. For approaching food, we use *reach for*. And for approaching furniture, we use *move to*. This differentiation helps to convey the varying probabilities associated with each action.

In Step (a), we have found that the combination of *within your immediate reach* and *reach for* has a higher probability of occurrence compared to the combination of *close to* and *reach for*.

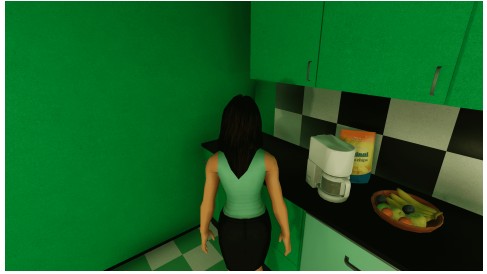

(b) The agent executes *Walk-Chips*.

**Observation prompt step (b):** There are four rooms: the kitchen, bathroom, bedroom, and living room. You are in the kitchen and notice chips and milk. The chips are within your immediate reach. But you have not grabbed the chips. Currently, you are not grabbing anything in hand. In order to enjoy the chips and the milk while watching TV, your next step is to

Table 46: Action Prompt in Step (b)

| Action | Token | Token Probability | Action Probability of TWOSOME | | |
|---|---|---|---|---|---|
| | | | W/O Norm | Token Norm | Word Norm |
| walk to the living room | [walk, to, the, living, room] | [0.08, 50.83, 87.12, 29.02, 99.06] | 0.50 | 16.55 | 21.19 |
| walk to the bathroom | [walk, to, the, bath, room] | [0.08, 50.83, 87.12, 20.9, 99.24] | 0.36 | 15.50 | 12.37 |
| walk to the bedroom | [walk, to, the, bed, room] | [0.08, 50.83, 87.12, 9.27, 97.89] | 0.16 | 13.14 | 10.06 |
| reach for the milk | [reach, for, the, milk] | [3.32, 29.07, 76.17, 7.37] | 2.49 | 15.68 | 20.07 |
| grab the chips | [grab, the, ch, ips] | [3.45, 75.04, 81.22, 99.96] | 96.50 | 39.13 | 36.31 |

Analysis: In Step (b), as the agent is currently close to the chips, the action *grab the chips* is included, while the action *reach for the chips* is removed from the options. However, despite the presence of the option *reach for the milk*, *grab the chips* still obtains the highest probability.

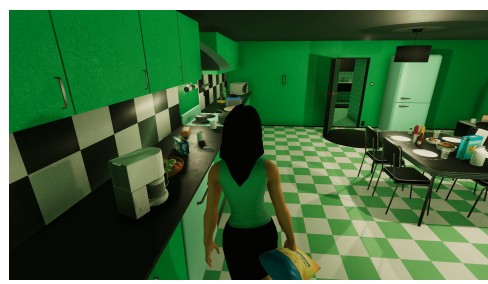 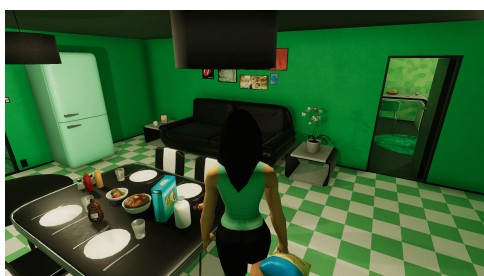

(c) After approaching the chips, the agent executes **Walk-Grab-Chips**.

(d) After grabbing the chips, the agent executes **Walk-Milk**.

**Observation prompt step (c):** There are four rooms: the kitchen, bathroom, bedroom, and living room. You are in the kitchen and notice chips and milk. The milk is not within your immediate reach. Currently, you have grabbed the chips in hand. In order to enjoy the chips and the milk while watching TV, your next step is to

Table 47: Action Prompt in Step (c)

| Action | Token | Token Probability | Action Probability of TWOSOME | | |
|---|---|---|---|---|---|
| | | | W/O Norm | Token Norm | Word Norm |
| walk to the living room | [walk, to, the, living, room] | [0.3, 44.83, 87.01, 40.11, 98.8] | 15.45 | 27.58 | 32.75 |
| walk to the bathroom | [walk, to, the, bath, room] | [0.3, 44.83, 87.01, 17.79, 99.44] | 6.90 | 23.47 | 18.24 |
| walk to the bedroom | [walk, to, the, bed, room] | [0.3, 44.83, 87.01, 10.79, 98.43] | 4.14 | 21.20 | 16.05 |
| reach for the milk | [reach, for, the, milk] | [1.26, 29.88, 88.35, 66.42] | 73.51 | 27.75 | 32.95 |

**Observation prompt step (d):** There are four rooms: the kitchen, bathroom, bedroom, and living room. You are in the kitchen and notice chips and milk. The milk is within your immediate reach. But you have not grabbed the milk. Currently, you have grabbed the chips in hand. In order to enjoy the chips and the milk while watching TV, your next step is to

Table 48: Action Prompt in Step (d)

| Action | Token | Token Probability | Action Probability of TWOSOME | | |
|---|---|---|---|---|---|
| | | | W/O Norm | Token Norm | Word Norm |
| walk to the living room | [walk, to, the, living, room] | [4.93, 52.67, 93.82, 46.07, 99.14] | 20.09 | 29.30 | 33.09 |
| walk to the bathroom | [walk, to, the, bath, room] | [4.93, 52.67, 93.82, 19.81, 99.4] | 8.66 | 24.76 | 21.41 |
| walk to the bedroom | [walk, to, the, bed, room] | [4.93, 52.67, 93.82, 10.94, 98.6] | 4.74 | 21.96 | 18.42 |
| grab the milk | [grab, the, milk] | [5.67, 88.74, 73.16] | 66.51 | 23.97 | 27.07 |

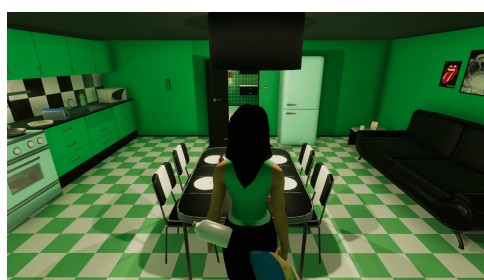 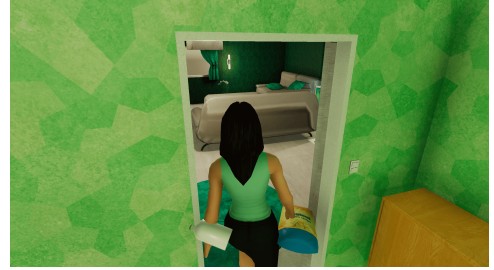

(e) After approaching the milk, the agent executes **Walk-Grab-Milk**.

(f) After grabbing the milk, the agent executes **Walk-Livingroom**.

**Observation prompt step (e):** There are four rooms: the kitchen, bathroom, bedroom, and living room. You are in the kitchen and notice chips and milk. Currently, you have grabbed the chips and the milk in hand. In order to enjoy the chips and the milk while watching TV, your next step is to

Table 49: Action Prompt in Step (e)

| Action | Token | Token Probability | Action Probability of TWOSOME | | |
|---|---|---|---|---|---|
| | | | W/O Norm | Token Norm | Word Norm |
| walk to the living room | [walk, to, the, living, room] | [2.18, 46.82, 92.81, 65.93, 98.53] | 79.68 | 43.25 | 52.18 |
| walk to the bathroom | [walk, to, the, bath, room] | [2.18, 46.82, 92.81, 11.11, 99.4] | 13.55 | 30.34 | 25.98 |
| walk to the bedroom | [walk, to, the, bed, room] | [2.18, 46.82, 92.81, 5.59, 98.8] | 6.77 | 26.41 | 21.84 |

**Observation prompt step (f):** There are four rooms: the kitchen, bathroom, bedroom, and living room. You are in the living room and notice a coffee table, a TV and a sofa. They are not close to you. Currently, you have grabbed the chips and the milk in hand. To enjoy the chips and the milk while watching TV, your next step is to

Table 50: Action Prompt in Step (f)

| Action | Token | Token Probability | Action Probability of TWOSOME | | |
|---|---|---|---|---|---|
| | | | W/O Norm | Token Norm | Word Norm |
| walk to the kitchen | [walk, to, the, kitchen] | [3.66, 47.61, 90.77, 46.41] | 40.70 | 19.59 | 22.68 |
| walk to the bathroom | [walk, to, the, bath, room] | [3.67, 47.91, 90.77, 0.7, 99.49] | 0.62 | 10.85 | 7.97 |
| walk to the bedroom | [walk, to, the, bed, room] | [3.67, 47.91, 90.77, 1.82, 98.62] | 1.59 | 13.10 | 10.09 |
| move to the coffee table | [move, to, the, coffee, table] | [11.49, 28.79, 89.56, 3.26, 98.59] | 5.28 | 16.65 | 19.28 |
| move to the TV | [move, to, the, TV] | [11.44, 28.82, 89.53, 21.56] | 35.32 | 18.91 | 21.89 |
| move to the sofa | [move, to, the, so, fa] | [11.49, 28.79, 89.56, 10.04, 99.96] | 16.49 | 20.91 | 18.10 |

Analysis: In Step (e), since the agent now has grabbed the chips and milk, its only choice is to leave the kitchen and move to another room.

In Step (f), currently, the agent is in the living room. Since the agent is not close to the coffee table, TV, or sofa, it does not have any actions related to interacting with those furniture pieces. It is worth mentioning that the action *walk to the kitchen* has the highest probability. However, the difference in probabilities between the action *move to "furniture"* and *walk to the kitchen* is not significant.

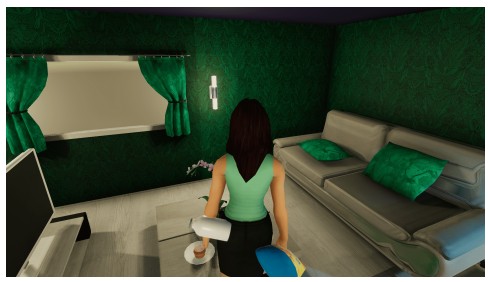 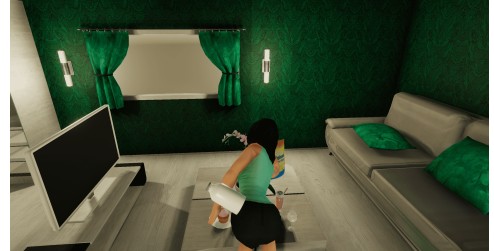

(g) The agent leaves the kitchen and enters the living room, then executes *Walk-Coffeetable*.

(h) After approaching the coffee table, the agent executes *Putback-Chips-Coffeetable*.

**Observation prompt step (g):** There are four rooms: the kitchen, bathroom, bedroom, and living room. You are in the living room and notice a coffee table, a TV and a sofa. The coffee table is close to you. Currently, you have grabbed the chips and the milk in hand. In order to enjoy the chips and the milk while watching TV, your next step is to

Table 51: Action Prompt in Step (g)

| Action | Token | Token Probability | Action Probability of TWOSOME | | |
|---|---|---|---|---|---|
| | | | W/O Norm | Token Norm | Word Norm |
| walk to the kitchen | [walk, to, the, kitchen] | [1.96, 40.29, 92.74, 15.44] | 4.70 | 9.24 | 10.40 |
| walk to the bathroom | [walk, to, the, bath, room] | [1.96, 40.27, 92.72, 0.23, 99.61] | 0.07 | 5.59 | 3.63 |
| walk to the bedroom | [walk, to, the, bed, room] | [1.96, 40.27, 92.72, 0.33, 97.82] | 0.10 | 6.00 | 3.97 |
| move to the TV | [move, to, the, TV] | [9.41, 23.75, 90.57, 19.12] | 16.11 | 12.57 | 14.15 |
| move to the sofa | [move, to, the, so, fa] | [9.36, 23.75, 90.54, 26.99, 99.98] | 22.60 | 17.76 | 15.39 |
| put the chips on the coffee table | [put, the, ch, ips, on, the, coffee, table] | [20.56, 78.62, 84.28, 99.98, 11.51, 95.85, 81.33, 99.17] | 50.42 | 29.03 | 30.19 |
| put the milk on the coffee table | [put, the, milk, on, the, coffee, table] | [20.56, 78.63, 4.54, 28.88, 95.2, 72.08, 99.16] | 6.00 | 19.80 | 22.28 |

**Observation prompt step (h):** There are four rooms: the kitchen, bathroom, bedroom, and living room. You are in the living room. You notice a coffee table, a TV and a sofa. The TV is not close to you. Currently, you have the chips on the coffee table and the milk in your hand. In order to enjoy the chips and the milk while watching TV, your next step is to

Table 52: Action Prompt in Step (h)

| Action | Token | Token Probability | Action Probability of TWOSOME | | |
|---|---|---|---|---|---|
| | | | W/O Norm | Token Norm | Word Norm |
| walk to the kitchen | [walk, to, the, kitchen] | [1.68, 44.48, 92.17, 9.36] | 14.16 | 18.49 | 22.91 |
| walk to the bathroom | [walk, to, the, bath, room] | [1.68, 44.44, 92.2, 0.2, 99.68] | 0.30 | 12.34 | 8.73 |
| walk to the bedroom | [walk, to, the, bed, room] | [1.68, 44.44, 92.2, 0.22, 98.18] | 0.33 | 12.57 | 8.93 |
| move to the TV | [move, to, the, TV] | [5.69, 12.39, 89.18, 42.93] | 59.25 | 26.45 | 32.77 |
| move to the sofa | [move, to, the, so, fa] | [5.73, 12.39, 89.32, 18.64, 99.97] | 25.97 | 30.15 | 26.66 |

Analysis: In Step (g), since the agent is next to the coffee table, there is now actions available to place chips and milk on the coffee table.

In Step (h), we emphasize that the chips are already on the coffee table, reminding the agent to move to other furniture nearby.

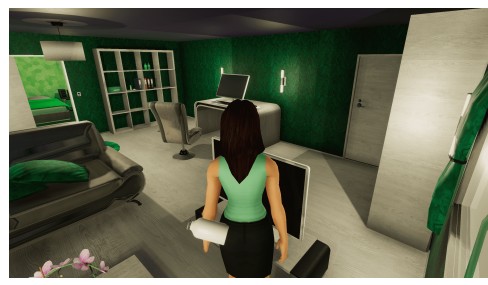 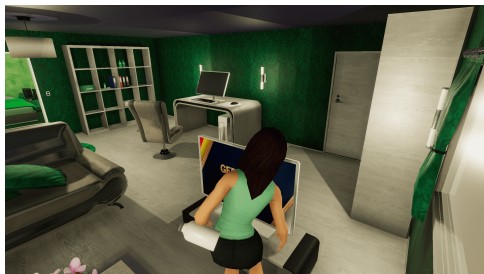

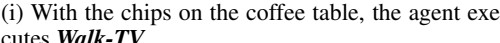

(i) With the chips on the coffee table, the agent executes **Walk-TV**.

(j) After approaching the television, the agent executes **Switchon-TV**.

**Observation prompt step (i):** There are four rooms: the kitchen, bathroom, bedroom, and living room. You are in the living room. You notice a coffee table, a TV and a sofa. The TV is close to you. Currently, you have the chips on the coffee table and the milk in your hand. In order to enjoy the chips and the milk while watching TV, your next step is to

Table 53: Action Prompt in Step (i)

| Action | Token | Token Probability | Action Probability of TWOSOME | | |
|---|---|---|---|---|---|
| | | | W/O Norm | Token Norm | Word Norm |
| walk to the kitchen | [walk, to, the, kitchen] | [0.59, 41.18, 93.12, 15.95] | 5.90 | 12.24 | 14.14 |
| walk to the bathroom | [walk, to, the, bath, room] | [0.59, 41.2, 93.13, 0.17, 99.72] | 0.06 | 7.32 | 4.54 |
| walk to the bedroom | [walk, to, the, bed, room] | [0.59, 41.2, 93.13, 0.19, 98.0] | 0.07 | 7.46 | 4.64 |
| move to the coffee table | [move, to, the, coffee, table] | [3.92, 15.52, 91.54, 13.48, 99.22] | 12.11 | 21.00 | 24.27 |
| move to the sofa | [move, to, the, so, fa] | [3.92, 15.52, 91.54, 23.29, 99.98] | 21.08 | 23.46 | 19.45 |
| turn on the TV | [turn, on, the, TV] | [0.53, 78.05, 94.08, 94.97] | 60.27 | 21.88 | 25.29 |
| turn off the TV | [turn, off, the, TV] | [0.53, 0.97, 94.56, 64.39] | 0.51 | 6.64 | 7.67 |

**Observation prompt step (j):** There are four rooms: the kitchen, bathroom, bedroom, and living room. You are in the living room. You notice a coffee table, a TV and a sofa. The sofa is not close to you. Currently, the TV is turned on, you have the chips on the coffee table and the milk in your hand. In order to enjoy the chips and the milk while watching TV, your next step is to

Table 54: Action Prompt in Step (j)

| Action | Token | Token Probability | Action Probability of TWOSOME | | |
|---|---|---|---|---|---|
| | | | W/O Norm | Token Norm | Word Norm |
| walk to the kitchen | [walk, to, the, kitchen] | [1.72, 45.72, 91.41, 6.41] | 5.86 | 9.67 | 11.83 |
| walk to the bathroom | [walk, to, the, bath, room] | [1.71, 45.34, 91.44, 0.09, 99.68] | 0.08 | 5.98 | 4.02 |
| walk to the bedroom | [walk, to, the, bed, room] | [1.71, 45.34, 91.44, 0.16, 97.46] | 0.14 | 6.76 | 4.68 |
| move to the coffee table | [move, to, the, coffee, table] | [5.62, 16.76, 87.12, 8.05, 99.14] | 8.34 | 15.24 | 18.64 |
| move to the sofa | [move, to, the, so, fa] | [5.62, 16.76, 87.12, 57.68, 99.98] | 60.23 | 22.64 | 21.19 |
| take a seat on the sofa | [take, a, seat, on, the, so, fa] | [7.79, 14.31, 27.5, 72.98, 95.85, 90.65, 99.97] | 24.72 | 27.06 | 28.53 |
| stand up from the sofa | [stand, up, from, the, so, fa] | [0.21, 71.61, 6.97, 56.23, 82.01, 99.97] | 0.63 | 12.64 | 11.11 |

Analysis: In Step (i), the agent is next to the TV, and it is emphasized that the TV is not turned on, increasing the probability of turning on the TV.

In Step (j), we found that the probability of *take a seat on the sofa* is higher than the probability of *move to the sofa*, which reflects the ability of LLaMA 7B. However, we believe that TWOSOME will gradually learn the optimal action for this step, and experimental evidence supports this belief.

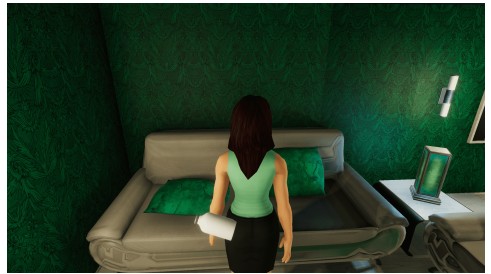

(k) After turning on the television, the agent executes
*Walk-Sofa*.

**Observation prompt step (k):** There are four rooms: the kitchen, bathroom, bedroom, and living room. You are in the living room. You notice a coffee table, a TV and a sofa. The sofa is close to you. Currently, the TV is turned on, you have the chips on the coffee table and the milk in your hand. In order to enjoy the chips and the milk while watching TV, your next step is to

Table 55: Action Prompt in Step (k)

| Action | Token | Token Probability | Action Probability of TWOSOME | | |
|---|---|---|---|---|---|
| | | | W/O Norm | Token Norm | Word Norm |
| walk to the kitchen | [walk, to, the, kitchen] | [0.6, 40.01, 93.07, 6.71] | 2.38 | 8.60 | 9.98 |
| walk to the bathroom | [walk, to, the, bath, room] | [0.6, 40.03, 93.07, 0.06, 99.69] | 0.02 | 5.27 | 3.12 |
| walk to the bedroom | [walk, to, the, bed, room] | [0.6, 40.03, 93.07, 0.11, 96.73] | 0.04 | 5.80 | 3.52 |
| move to the coffee table | [move, to, the, coffee, table] | [3.77, 21.04, 90.28, 6.26, 99.11] | 7.09 | 16.62 | 19.27 |
| move to the TV | [move, to, the, TV] | [3.74, 21.02, 90.39, 11.53] | 13.08 | 13.16 | 15.27 |
| take a seat on the sofa | [take, a, seat, on, the, so, fa] | [7.06, 18.35, 53.63, 76.32, 97.3, 93.12, 99.97] | 76.79 | 36.30 | 37.08 |
| stand up from the sofa | [stand, up, from, the, so, fa] | [0.16, 72.24, 6.41, 59.92, 85.86, 99.98] | 0.60 | 14.24 | 11.76 |

