# OpenReview forum: "True Knowledge Comes from Practice: Aligning Large Language Models with Embodied Environments via Reinforcement Learning"
_ICLR.cc/2024/Conference — ICLR 2024 poster_

### Official Review · Reviewer_HnZU · 2023-10-16

**Soundness:** 3 good
**Presentation:** 4 excellent
**Contribution:** 3 good
**Rating:** 6
**Confidence:** 4

**Summary:**

The paper introduces TWOSOME, a novel finetuning method designed to ground Large Language Models (LLMs) in embodied environments. TWOSOME comprises three main components: action-selection, where the LLM identifies the best action from a list of actions on criteria such as likelihood, token normalization, and word normalization, LoRA updates and prompt design. The authors conducted empirical experiments using TWOSOME on four fine-tuning tasks within Overcooked/VirtualHome environments. Their results demonstrate that TWOSOME's word normalization component outperforms others. Moreover, the fine-tuned policies exhibit superior generalization to unseen tasks in zero-shot scenarios. Remarkably, TWOSOME's capabilities in NLP benchmarks remain unaffected after the fine-tuning process.

**Strengths:**

- The authors noticed the limitation of considering plain action likelihood (a problem known in NLP when selecting different texts based on likelihood of the full sentences). To my knowledge, the word normalization is new and I really liked the idea when noticing that dividing by the sequence length is not enough to mitigate the end of words being more likely.
- The authors worked in a compute budgeted setting so the work can be reproduced easily.
- I really liked the task generalization section.
- Results on showing little to no catastrophic forgetting on NLP tasks (possibly coming from LoRA?) are really promising.

**Weaknesses:**

- As for now it is hard to consider LoRA to be part of the proposed method: hard to see the benefits of the LoRA contribution to sample efficiency and no catastrophic forgetting without an ablation on with and without LoRA.
The contribution of the method seems to be the action normalization and some prompt design (qualitatively assessed by the authors in 4.4).  Could you add experiments on these points?

- Paper mentions that (Carta et al, 2023) “focus on primitive actions in toy environments without rich semantics“. I am not sure actions are primitive in (Carta et al, 2023) but rather semantically described actions, e.g. “go forward”, “turn left”, “drop and toggle”. What makes them more “primitive” seems to be the environment they chose for experimenting (BabyAI vs Overcooked/VirtualHome).

- Regarding the choice of environments. I am afraid there are too few fine-tuning tasks. I might be wrong, but the task generalization seems almost too-good from 4 tasks only. BabyAI was procedurally generated which would have enabled experimentation on more finetuning tasks.

**Questions:**

- Can you elaborate on the difference between (Carta et al, 2019) and unnormalized action selection.
- Can you explain the PPO baseline: state/action space and architecture and initialization here (and in the main paper)? This would help in understanding the contribution of pretrained initialization.
- Discussion question (not required to increase my score):
     - Do you have insights on the use of embodied environments to **improve** LLMs, e.g. dealing with safety concerns?  How to design such environments?
    - It would be nice to see how multi-modal text generation could be used to remove the assumption that the agent can extract a textual description of its state.

---

> ### Author Response · Authors · 2023-11-15
>
> We thank the reviewers for their time and comments. Please see the above official comment to all the reviewers first and then our answers to your questions. We hope our answers will clear up the doubts about our work, and please let us know if there is any other clarification we can provide.
>
> ***
>
> Q1: As for now it is hard to consider LoRA to be part of the proposed method: hard to see the benefits of the LoRA contribution to sample efficiency and no catastrophic forgetting without an ablation on with and without LoRA. The contribution of the method seems to be the action normalization and some prompt design (qualitatively assessed by the authors in 4.4). Could you add experiments on these points?
>
> A1: Thanks for your valuable comments. **We want to point out that we use LoRA mainly for parameter efficiency instead of sample efficiency.** Even using LoRA, TWOSOME still has about 500M parameters to update. In contrast, the PPO network only has about 10K parameters to update, which is 50,000X less than TWOSOME. But TWOSOME still exhibits much better sample efficiency than PPO with the same amount of data, we attribute this to the prior knowledge of LLM, instead of LoRA.
>
> As mentioned in the first point of the above official comment, in the section of **Parameter-efficient Fine-tuning Architecture**, one of our main baselines, GLAM requires 8 NVIDIA A100 80GB GPUs to train their framework with a much smaller LLM, Flan-T5 780M without using LoRA. They report using 18880 GPU hours on Nvidia A100 80GB in total. Our model is 9X larger than theirs. **We estimate that at least 32 NVIDIA A100 80GB GPUs are required to conduct the experiments to train LLaMA-7B without using LoRA, otherwise, there is even no way to load the model and do a complete backpropagation.** Training LLaMA-7B without using LoRA is significantly far beyond most research labs’ ability, including us. It also shows the importance of our non-trivial parameter-efficient architecture, which enables the whole framework to run in a single A100 40G GPU. It greatly lowers the threshold for researchers to conduct the following research.
>
> ***
>
> Q2: Paper mentions that (Carta et al, 2023) “focus on primitive actions in toy environments without rich semantics“. I am not sure actions are primitive in (Carta et al, 2023) but rather semantically described actions, e.g. “go forward”, “turn left”, “drop and toggle”. What makes them more “primitive” seems to be the environment they chose for experimenting (BabyAI vs Overcooked/VirtualHome).
>
> A2: Yes, you are correct. That is one of the main reasons why we do not think BabyAI is a suitable environment to exhibit the performance of LLM-based agents. **We argue that LLM-based agents should be deployed in environments as close to daily life or their learned specific domains as possible, like SayCan with high-level daily skills (household robotics).**  Then they can make full use of the prior knowledge learned in the training process. If LLMs are not familiar with the observation and actions, not only will they fail to apply their learned prior knowledge but also dramatically slow down the training process for their tremendous parameters. It also emphasizes the importance of prompt design.
>
> As we mentioned in the above official comment, both the observation and action in BabyAI have relatively poor semantics. Here is an example prompt used by GLAM in BabyAI, “You carry a blue ball, You see a wall 4 steps forward, You see a wall 2 steps left, You see a grey key 1 step right and 1 step forward, You see a red box 2 steps forward”. It is rare for LLMs to see a corpus like this in the training process and we all know that even GPT-4 struggles with simple navigation tasks. It might partially explain why GLAM spent as much as 18880 GPU hours on Nvidia A100 80GB to get the results.
> LLM-based agents should not be applied to solve tasks in these environments under primitive settings. They are exactly the strong areas of RL agents. This is the reason why we insist on deploying our framework in daily environments with rich semantics for observation and actions.
>
> ***

---

> > ### Author Response · Authors · 2023-11-15
> >
> > Q3: Regarding the choice of environments. I am afraid there are too few fine-tuning tasks. I might be wrong, but the task generalization seems almost too good from 4 tasks only. BabyAI was procedurally generated which would have enabled experimentation on more finetuning tasks.
> >
> > A3: In the previous question Q2, we explained that we do not think BabyAI is a suitable environment to show the performance of our method. Here we provide another perspective.
> > Compared to our representative environments, BabyAI is a relatively easy environment, which can be totally solved by PPO as shown in [1]. According to Figure 4 and Table 1 shown in GLAM, even with a random policy, the agent can still have as high as 30% success rate.  In both of our two environments, the agent with random policy can never manage to finish the task with a success rate of 0. Even SayCan still has an extremely low success rate. We argue that BabyAI is not a suitable environment to show the performance of our methods.
> >
> > We believe our current representative environments and tasks are enough to show the robustness and generalization of our methods. Please refer to the second point of the above official comment, **About Evaluations on More Tasks and Environments**, for a clear explanation.
> >
> >
> > [1] Chevalier-Boisvert M, Bahdanau D, Lahlou S, et al. Babyai: A platform to study the sample efficiency of grounded language learning[J]. arXiv preprint arXiv:1810.08272, 2018.
> >
> > ***
> >
> > Q4: Can you elaborate on the difference between (Carta et al, 2019) and unnormalized action selection.
> >
> > A4: Sorry for the confusion. Due to the limitation of pages, we had to put the extended related work in Appendix A, where there are more discussions, especially the difference from previous works, like SayCan and GLAM.
> >
> > We assume the work that the reviewer refers to is GLAM (Carta et al, 2023). The main differences between TWOSOME without normalization and GLAM are the architecture and prompt design. 1) GLAM uses a much smaller encoder-decoder LLM, Flan-T5-780M, which requires 8 Nvidia A100 80GB GPUs to train. All TWOSOME agents use a 9X larger decoder-only LLM, LLaMA-7B, which only requires a single A100 40GB GPU to train, benefiting from our parameter-efficient framework. 2) GLAM focuses on simple primitive actions (turn left, turn right, go forward, etc.) evaluated in toy environments, BabyAI. These primitive actions have a similar number of tokens and less meaningful semantics, resulting in underutilizing the capabilities of LLMs, and failing to observe the impact of prompt design and address the unbalance over action space, resulting in additional instability and poor robustness. Though TWOSOME without normalization does not apply normalization, either, the design of observation and action prompts are more natural and robust.
> >
> > ***
> >
> > Q5: Can you explain the PPO baseline: state/action space and architecture and initialization here (and in the main paper)? This would help in understanding the contribution of pretrained initialization.
> >
> > A5: Sorry for the confusion. Due to the limitation of pages, we had to put the description of state and action space in Appendix C and the architecture of PPO in Appendix D.1. We will move them to the main paper in the camera-ready version since there will be more space provided. PPO adopts symbolic observations, which consist of the positions and status of the agent and all the objects in the observations. Note that both two environments are partially observable. All the unseen objects will be masked. For action space, PPO adopts the same action space as TWOSOME. In the PPO method, both the actor and critic networks adopt a 3-layer MLP architecture with 64 neurons in each layer. PPO is initialized with orthogonal initialization.
> >
> > ***

---

> > > ### Author Response · Authors · 2023-11-15
> > >
> > > Q6: Do you have insights on the use of embodied environments to improve LLMs, e.g. dealing with safety concerns? How to design such environments?
> > >
> > > A6: Thanks for pointing this out. We have been very interested in deploying LLMs in various embodied environments to solve different complex tasks and improve LLMs themselves. Designing such environments from scratch might be costly and time-consuming. We believe that games are the perfect environments for LLMs to explore and self-improve. Games contain many rules and laws, which are carefully designed by the developers. These rules actually represent the values of humans, like what we should do, what we should not do, what is good, what is bad, where is safe and where is dangerous.
> > >
> > > There are many simulation games, which replicate certain situations in "real" life. **The Sims 4** might be one of the most famous games among them. Players can determine the appearances, personalities, and aspirations of the characters and need to help them pursue careers, develop skills, form relationships, and achieve life goals, like being a doctor, teacher and lawyer. This game provides tremendous alignment with reality. If we can make LLMs able to play the game well and manage to achieve life goals or good careers, then it means that LLMs have a good understanding of human society and that they can strictly follow the rules and laws to act as a good citizen according to the instructions of humans. We hope this may answer your questions on the safety concerns.
> > >
> > > There are already some works using LLMs to play Minecraft, like Voyager[1], GITM[2] and DEPS[3]. However, all of them heavily rely on GPT-4 to do the planning without updating and improving LLMs themselves. That is also the intuition of TWOSOME. Our method can online interact with the environments and keep improving LLMs with the experiences. We hope our method can be applied to more complex and realistic environments if more computational resources are provided.
> > >
> > > [1] Wang, Guanzhi, et al. "Voyager: An open-ended embodied agent with large language models." arXiv preprint arXiv:2305.16291 (2023).
> > > [2] Zhu, Xizhou, et al. "Ghost in the Minecraft: Generally Capable Agents for Open-World Enviroments via Large Language Models with Text-based Knowledge and Memory." arXiv preprint arXiv:2305.17144 (2023).
> > > [3] Wang, Zihao, et al. "Describe, explain, plan and select: Interactive planning with large language models enables open-world multi-task agents." arXiv preprint arXiv:2302.01560 (2023).
> > >
> > > ***
> > >
> > > Q7: It would be nice to see how multi-modal text generation could be used to remove the assumption that the agent can extract a textual description of its state.
> > >
> > > A7: Yes, you are correct. A powerful VLM, like GPT-4V, can definitely greatly improve the performance and make our framework more complete. It will also facilitate the research in more complex environments as we mentioned in the last question, Q6. We are also thrilled to dig out GPT-4V’s ability, including image understanding and decision making, and imagine the infinite possibilities brought by GPT-4V.

---

> > > > ### Comment · Reviewer_HnZU · 2023-11-17
> > > > **Final remarks.**
> > > >
> > > > **Framework.** I have a hard time convincing myself that the proposed method is a "framework". I consider the method contribution to be only the action normalization (and prompt design). The LoRA part cannot be considered part of the method as its contribution is not evaluated.  That said, I totally understand that LoRA was the only solution to finetune larger models given limited compute budgets. I won't ask for extra experiments and am happy with the current ones but probably be careful with the formulation of a "framework". I also believe the model/critic architecture is not really ablated so hard to consider it part of the method. As for now, without more ablations on both LoRA and mode/critic, I think a "framework" is not a good formulation and the authors could improve their writing to make LoRA an important experimental detail that is required to finetune larger pre-trained models.
> > > >
> > > > **Environments.** I understand that BabyAI is a poorer environment in terms of semantics compared to overcooked/virtualhome, however my main point was that the choice of overcooked/virtualhome comes at the expense of the number of finetuning tasks (which could easily be procedurally generated). "BabyAI is a relatively easy environment" is that true for all scenarios? Couldn't you come with hard scenarios? Once again I understand that this would become more compute intensive (for which I ask a justification later).
> > > >
> > > > **Computational consumption of experiments.** Could you please describe why finetuning on downstream tasks is compute intensive? What is the main bottleneck (slowness of the environment, computing actions with a large model...)? I don't know how overcooked/virtualhome are implemented but I would expect the inference of a 7B model not be too long compared to the environment steps.

---

> > > > > ### Author Response · Authors · 2023-11-19
> > > > >
> > > > > Thanks for your valuable comments and quick response. We hope our following answers will clear up the doubts, and please let us know if there is any other clarification we can provide.
> > > > >
> > > > >
> > > > > **About Framework**
> > > > > Sorry for the confusion. Though we are not able to provide the ablation studies of finetuning LLMs without LoRA, **we just added a new ablation study** of the LLM-based actor-critic architecture to show that it can not only save the GPU memory but also improve stability and accelerate convergence in **Appendix G.1** (on the last page of the paper).
> > > > >
> > > > > Let us make a clarification again. We managed to optimize every module in the framework by introducing normalization and prompt design for forming the policy and designing a parameter-efficient finetuning framework for stable training. Using LoRA to train LLMs is common in supervised learning tasks. However, **applying LoRA in decision-making tasks, trained with RL was rarely explored before, which is actually not an out-of-box tool to use directly.** We also report the tricks and techniques we use to make the training process more stable in Section 4.3. We think these valuable studies and insights will also contribute to the embodied community.
> > > > >
> > > > > 1. The actor and critic models are usually two independent models in classical RLHF settings. Our novel architecture enables the actor and critic to share the same LLM backbone, which saves the GPU memory for maintaining two copies of the LLM. However, as shown in the new ablation study in Appendix G.1 (on the last page of the paper), directly applying the LoRA to the LLM backbone will introduce additional instability. The actor and critic share the same gradient from LoRA will greatly influence each other. We leverage the feature of LoRA that it does not directly update the parameters of the backbone model and use the model without LoRA to serve as the backbone of the critic model. It not only saves the effort of maintaining two copies of LLMs in the memory but also guarantees that they have independent gradients to update, making the training process more stable and efficient.
> > > > >
> > > > > 2. We also note that the dropout layers would bring additional training instabilities. Due to the randomness brought by the dropout layer, even with the same input, the output of the loglikelihood of each action is different every time, which will more frequently violate the KL divergence constraint in PPO. Therefore, we do not use dropout in our LoRA modules.
> > > > >
> > > > > 3. We observe the training instability when updating the actor multiple times with the same data. Therefore, every sampled data is discarded after training once.

---

> > > > > > ### Author Response · Authors · 2023-11-19
> > > > > >
> > > > > > **About Computational Consumption**
> > > > > > The bottlenecks of finetuning LLMs on downstream tasks are mainly two parts: 1) the inference and update speed of LLaMA-7B. 2) the feature of decision-making tasks (extremely low-quality noisy data and several seeds required to report). Compared to finetuning LLMs, the time used in interacting with environments can be almost ignored. 200K data can be collected in 10 minutes if using an agent with random policy in both Overcooked and VirtualHome.
> > > > > >
> > > > > > For the inference and update speed of LLaMA-7B, a good example for reference is the famous Alpaca LoRA, which needs several hours to finetune LLaMA-7B on the 52K Alpaca datasets. Compared to the pre-prepared high-quality data, we need to let the model interact with the environments and try to learn the optimal policy from the extremely low-quality noisy data, which usually needs far more data compared to supervised learning based methods. Our sparse reward setting makes the tasks more challenging, which also exacerbates the issue. In our experiments, our methods are trained with as much as 500K data in Overcooked and 200K data in VirtualHome, which is 10X and 4X than the Alpaca datasets, which takes at most 30 hours in a single A100 40G GPU for a single run.
> > > > > >
> > > > > > To follow the standard in RL and decision-making tasks, we report the average performance of 3 runs of different seeds of each method to show the robustness, which also dramatically increases the computational consumption, while the supervised-learning-based method usually only needs to report the performance of one run.
> > > > > >
> > > > > > ***
> > > > > >
> > > > > > **About Environments**
> > > > > > As mentioned in the **About Computational Consumption** section, the time used in interacting with environments can be almost ignored. In terms of the time used for training, there will be no significant difference among BabyAI, Overcooked and VirtualHome, with the same number of training steps, though the length of prompts and the number of actions may have some impacts.
> > > > > >
> > > > > > "BabyAI is a relatively easy environment" refers to the point that the out-of-box tasks in BabyAI are easy. We can indeed design more difficult maps and tasks by changing their source code, which requires another hard work. **However, we do not see any benefits in evaluating our methods in BabyAI given the budget constraints.**  If trained with easy tasks in BabyAI, it is not enough to show the difference between PPO and our method. If trained in hand-crafted difficult tasks, it takes even more time for our method to learn the optimal policy, due to the poor semantics, compared to Overcooked and VirtualHome. We have tried our best to select the most suitable and representative to show the performance of methods and the difference among baseline methods.

---

> > > > > > > ### Comment · Reviewer_HnZU · 2023-11-19
> > > > > > > **Final comment**
> > > > > > >
> > > > > > > Thanks for the last answers. It seems sample inefficiency on decision-making tasks (as compared to classical RLHF) prevents doing larger experiments under reasonable budgets. I increased my score to 6: I am in favor of this work to presented at a conference but think the experimental results are not enough (due to budget constraints) to make it a ground-breaking paper.

---

> > > > > > > > ### Author Response · Authors · 2023-11-19
> > > > > > > >
> > > > > > > > We greatly appreciate the reviewer's recognition. We also hope our work can be applied to various environments with sufficient computational resources.

---

### Official Review · Reviewer_fSyb · 2023-11-03

**Soundness:** 2 fair
**Presentation:** 3 good
**Contribution:** 2 fair
**Rating:** 6
**Confidence:** 3

**Summary:**

This paper provides TWOSOME (True knoWledge cOmeS frOM practicE), a Large Language Model (LLM)-based policy learning method that fine-tunes a LLM agent with a Reinforcement Learning (RL) algorithm. To address problems incurred when adopting text actions sampled from a LLM policy, this paper proposes a normalization method that divides the log probability of a text action by the number of words (or tokens). This paper demonstrates the effectiveness of TWOSOME by employing Llama-2-7B as a LLM agent and fine-tuning the LLM agent with Proximal Policy Optimization (PPO). This paper provides experiment results in Overcooked and VirtualHome, and shows that the LLM agent fine-tuned with PPO can learn better policy.

**Strengths:**

- S1. The main approach of fine-tuning a LLM agent (e.g., Llama-2-7B) with a RL algorithm (e.g., PPO) proposed in this paper seems interesting and promising. This approach can be seen as applying recent prevalent reinforcement learning with human feedback (RLHF) into decision-making tasks.

- S2. This paper provides promising initial results toward fine-tuning LLM agents with RL methods.

**Weaknesses:**

- W1. The overall approach of TWOSOME is to fine-tune a LLM agent (e.g., Llama-2-7B) with a RL algorithm (e.g., PPO). And, the unique contribution of TWOSOME can be seen as providing a normalization method for the probability distribution over text-based actions sampled from a LLM policy. This paper proposes two normalization methods: token normalization (Eq. 3) and word normalization (Eq. 4). Even though this paper proposes a promising direction towards fine-tuning a LLM agent with a RL algorithm, the main contribution seems rather marginal. The normalization method is about dividing the log probability of a varying length text-based action by the number of words (or the number of tokens). It seems to compensate a text action with larger number of words, but I am not sure that this kind of simple division is a proper way to normalize varying length text actions sampled from a LLM policy.

- W2. This paper shows that TWOSOME can learn the optimal policy in a simple environment like Overcooked. However, I am not sure that TWOSOME can properly learn optimal policy in a complex environment like VirtualHome. When showing the performance of TWOSOME in VirtualHome, the authors masks out unrelated actions to reduce the complexity of the action space.

**Questions:**

- Q1. Regarding W1, how does the word normalization help LLM agents to learn better policy? What is the intuition of the word normalization?

- Q2. Regarding W2, without manual action masking in VirtualHome, how much scores can TWOSOME achieve?

- Q3. Recently, in RLHF, some enhanced RL algorithms such as Direct Policy Optimization (DPO) and Pair-wise Proximal Policy Optimizaition (P3O) have been proposed. Can these enhance RL algorithms be applied to solving decision making tasks? And then, can we expect some performance improvements of TWOSOME?

---

> ### Author Response · Authors · 2023-11-15
>
> We thank the reviewers for their time and comments. Please see the above official comment to all the reviewers first and then our answers to your questions. We hope our answers will clear up the doubts about our work, and please let us know if there is any other clarification we can provide.
>
>
> ***
>
> Q1. How does the word normalization help LLM agents to learn better policy? What is the intuition of the word normalization?
>
> A1: For the intuition of the word normalization, we provided a detailed explanation in Section 4.1 in the main paper. Briefly speaking, **word normalization can provide a more accurate and reasonable initialized policy for the LLM agents to interact with environments**. So reasonable good actions, no matter whether they are long or short, always have higher probabilities to be sampled at the beginning of the training process, which would greatly accelerate the exploration process.
>
> We observe that the probability of each token is always less than 1. Therefore, longer action prompts tend to have lower token-level probabilities, even though the longer action prompts may be more reasonable in the current situation. So we propose the token normalization to normalize the joint probability by the number of tokens (length).
>
> For example, if we have two actions, the first action has 20 tokens. To to simple, let the probability of each token in this action be as same as 0.8, which means this action is well-written and highly related to the task and the current observation. However, the joint probability of the action is 0.8^20 = 0.011. In contrast, the second action has only two tokens, with a probability of 0.2 for each token,  which means this action is much less relative to the task and the observation, and even the relation between these two tokens is quite weak. The joint probability of the second action is 0.2 * 0.2 = 0.04. Though each token in the first action is much higher than the second action, the first action finally has much fewer chances to be chosen, compared to the second action, which is not a desirable result. With the use of the token/word normalization by taking the lengths of the action into consideration, the joint probability of the action is still 0.8 and the joint probability of the second action is still 0.2 The distribution is much more reasonable. And the first action has a higher probability of being sampled.
>
> However, we find the token normalization is slightly excessive. We observe that if a word is divided into several tokens, the first token usually has a relatively low probability, while the probabilities of the rest of the tokens tend to be remarkably high, which are often almost close to 100%. Therefore, it is more reasonable to regard the several tokens made up of one word as an integrated symbol. Then we propose our word normalization method to normalize the joint probabilities of actions by the number of words. Our experiments also demonstrate the same conclusion that TWOSOME without normalization is very unstable and fails to learn the optimal policy. Both TWOSOME with token normalization and word normalization have much better performance. And TWOSOME with word normalization converges much faster than token normalization since token normalization is slightly excessive.
>
> In Appendix F, we also provide a detailed comparison and analysis of the probabilities generated by TWOSOME without normalization and with token/word normalization across all the tasks.
>
>
> ***

---

> ### Author Response · Authors · 2023-11-15
>
> Q2. Even though this paper proposes a promising direction towards fine-tuning a LLM agent with a RL algorithm, the main contribution seems rather marginal. It seems to compensate a text action with larger number of words, but I am not sure that this kind of simple division is a proper way to normalize varying length text actions sampled from a LLM policy.
>
> A2: Thanks for your valuable comments.  We kindly remind the reviewer that the contribution of the whole framework may be underestimated. There are some previous similar works like SayCan and GLAM, but we managed to polish and optimize every module in the framework, resulting in high parameter-efficiency and better generalization and sample-efficiency. Please refer to the first point of the above official comment, **About Technical Contribution**, for a clear explanation.
>
> For the word normalization, we have introduced it in the previous question, Q1. We would not claim that our solution is the ultimate solution to the issue, but it turns out to be an easy-to-implement and feasible way to alleviate the issue, which is well acknowledged by the other three reviewers. Our method has shown significant improvement including both performance and stability according to the corresponding experiments and ablation studies.
>
> Not like the traditional natural language generation task where the output can be as long as several hundred tokens. The actions in traditional decision-making tasks are typically composed of a verb-object structure (e.g. pick up the apple), or Verb + Object + Preposition structure (e.g. put the apple on the table). They are usually within 20 tokens. Our experiments have covered the most common cases, which show that even in this range, the length of the prompts still has a significant influence on the joint probability of the actions and introduces much instability to the training process.
>
> ***
>
> Q3. I am not sure that TWOSOME can properly learn optimal policy in a complex environment like VirtualHome. When showing the performance of TWOSOME in VirtualHome, the authors masks out unrelated actions to reduce the complexity of the action space.
>
> A3: Sorry for the confusion. We intend to use action masking to prove that **compared to traditional RL methods, like PPO, it is more convenient for our method to mask actions and solve the tasks with a large action space**. When doing action masking, the network of PPO still needs to calculate the values of the masked actions, but we just do not use their values to form the policy in the final step. However, for our method, we just directly delete the masked actions from the action prompt list in the input, which also saves the computation for the values of these masked actions. Furthermore, it is also worth noting that our framework is also very flexible in extending the action space, by just directly add new actions to the action prompt list as the input, which is impossible for traditional RL methods. They need to train new agents from scratch with the new extended action space.
>
> Moreover, we argue that struggling with large action space has been one of the main challenges for all decision-making methods, which is not a unique weakness of our method. Action masking is a very common trick to deal with large action space in decision-making tasks. The action space of all the environments is also a subset of the action space of reality, where these environments also mask out unrelated actions to reduce the complexity. It is fairly reasonable to mask invalid actions and unrelated items in the action space.
>
> Finally, we want to kindly remind the reviewer that even with the use of action masking, the difficulty of VirtualHome might be underestimated. Because we adopt a sparse reward setting, only when the agent successfully finishes the task will it receive +1 reward, as shown in Figure 5, even PPO with action masking, still fails to solve the task of Entertainment. In contrast, our method, TWOSOME with word normalization manages to learn the optimal policy, which already shows that our method has the ability in a complex environment.
>
> ***
>
> Q4. Recently, in RLHF, some enhanced RL algorithms such as Direct Policy Optimization (DPO) and Pair-wise Proximal Policy Optimizaition (P3O) have been proposed. Can these enhance RL algorithms be applied to solving decision making tasks? And then, can we expect some performance improvements of TWOSOME?
>
> A4: To our understanding, compared to RLHF, both DPO and P3O attempt to directly use the preference data to train LLMs, saving the effort of training an accurate and explicit reward model or value function.  However, in the traditional decision-making setting, the reward function is already contained in the environment. We can directly use rewards to train our models instead of using preference pairs to infer the reward/value distribution. Therefore, DPO and P3O do not help in our standard RL decision-making settings.

---

> ### Author Response · Authors · 2023-11-21
>
> Dear reviewer fSyb:
>
> We just added some new results in Appendix G.2 (on the last page of the paper), to show that **even without action masking, our method, TWOSOME with word normalization, can still manage to learn the optimal policy**, which shows the superior robustness and performance of our method and demonstrates that our method has the ability to deal with large action space.
>
> We hope these new results can solve the reviewer's doubts. As the deadline for the rebuttal period is approaching, is there any other clarification we can provide?

---

> > ### Author Response · Authors · 2023-11-23
> >
> > Dear reviewer fSyb:
> >
> > We kindly remind the reviewer that the rebuttal period is going to end today. We have had some impressive discussions with other reviewers and really appreciate that our contribution is well acknowledged. We hope our clarification and new experimental results can clear up all the reviewer's doubts and confusion and receive more valuable feedbacks from the reviewer.
> >
> > Best regards,
> > Authors of Paper 3575

---

> > > ### Comment · Reviewer_fSyb · 2023-11-23
> > >
> > > Thank you for providing thoughtful responses to my comments. It helped me to understand this work more concretely. I will reiterate my review once again.

---

### Official Review · Reviewer_Xize · 2023-11-05

**Soundness:** 3 good
**Presentation:** 3 good
**Contribution:** 2 fair
**Rating:** 6
**Confidence:** 4

**Summary:**

This paper studies using a large language model (LLM) as a decision-making agent to interact with an embodied environment. Specifically, the authors proposed to train the policy with reinforcement learning (RL). First, the policy is formed by querying the LLM with each feasible action in a given state. Then, token normalization and word normalization approaches are proposed to stable the training. The proposed approach is evaluated on Overcooked and VirtualHome. The experiments show that the proposed approach outperforms baselines in terms of sample efficiency and return.

**Strengths:**

Originality and Significance:
The reviewer appreciates the motivation of the paper: Classical RL agents align well with a given environment but fail to leverage prior knowledge. On the other hand,  LLM has numerous prior knowledge while often fails to align with the given environment. This paper proposed a method to align the LLM policy with the environment and show improvement over classic RL methods.

Quality:
The paper is technically sound and the claims are supported by experiments.

Clarity:
This paper is generally well-organized and easy to follow. Some minor improvement may be required. Please see the Question section for details.

**Weaknesses:**

1. The idea of querying joint probabilities of each valid action with LLM to form a policy is explored in previous works such as GLAM (Carta et al., 2023). The reviewer found, in the current version, the credit is not clearly given to authors of previous works.

2. The reviewer has some concerns on the baselines used in the experiments. In Figure 5, aside from the ablations, the authors only consider classic PPO as a baseline. It seems unfair to directly compare an agent equipped with LLM with a classic trial-and-error PPO agent which doesn’t have access to the prior knowledge in LLM. There are many existing works that leverage LLM to form RL policies [1, 2, 3]. Particularly,  Li [1] also uses virtualHome as test environments. Comparing approaches that also leverage knowledge in LLM could make the experimental section more convincing.

3. The technical contribution of the paper is somewhat limited. Specifically, the idea of forming valid policy is proposed by previous works and LoRA for fine-tuning is a standard method. The review found the main new method is the proposed normalization approach. That to be said, the reviewer still considers the combination of existing methods might be valuable to our embodied AI community.

[1] Pre-Trained Language Models for Interactive Decision-Making, Li et al., NeurIPS 2022

[2] Do As I Can, Not As I Say: Grounding Language in Robotic Affordances, Ahn, CoRL 2022.

[3] Grounding large language models in interactive environments with online reinforcement learning, Cartra et al, 2023

**Questions:**

1. As discussed in Weakness, comparing with methods that also leverage LLM for RL agent could make the experimental section more convincing.

2. In the experiments, only two tasks from virtualHome and Overcooked are considered. Reporting results on more tasks would be helpful.

3. In Section 4.3, it reads “The critic’s MLPs use the last token of the observation prompt as input …”. Could you elaborate why only the last token of the observation prompt is used? Shouldn’t the MLP use the output of the frozen LLM?

Minor:
  4. In Eq (2), using $a_k$ in the summation is confusing.

---

> ### Author Response · Authors · 2023-11-15
>
> We thank the reviewers for their time and comments. Please see the above official comment to all the reviewers first and then our answers to your questions. We hope our answers will clear up the doubts about our work, and please let us know if there is any other clarification we can provide.
>
>
> ***
>
> Q1: The idea of querying joint probabilities of each valid action with LLM to form a policy is explored in previous works such as GLAM (Carta et al., 2023). The reviewer found, in the current version, the credit is not clearly given to authors of previous works.
>
> A1: Thanks for pointing this out. **To our best knowledge, it is SayCan that first proposed to use joint probabilities of each valid action to form a policy**. And GLAM also acknowledges this point. Here is an excerpt from GLAM in Page 5, “Finally, Ahn et al. [2022] proposed to directly use the LLM to compute the (log) probability of each action by computing the conditional probability of each token in action given the prompt p”, where Ahn et al. [2022] refers to SayCan. We also mention this in our 4.1 Section that the way we used to form the policy is very similar to SayCan without the affordance function. We will add GLAM behind SayCan, since it also uses the same way to form the policy.
>
> ***
>
> Q2: The reviewer has some concerns on the baselines used in the experiments. There are many existing works that leverage LLM to form RL policies [1, 2, 3]. Comparing approaches that also leverage knowledge in LLM could make the experimental section more convincing.
>
> [1] Pre-Trained Language Models for Interactive Decision-Making, Li et al., NeurIPS 2022
> [2] Do As I Can, Not As I Say: Grounding Language in Robotic Affordances, Ahn, CoRL 2022.
> [3] Grounding large language models in interactive environments with online reinforcement learning, Cartra et al, 2023
>
> A2: Thanks for your pointing out. Please refer to the third point of the above official comment, **About Baselines Selection**, for a clear explanation. **Actually, we do compare our method with SayCan[2] and GLAM[3]**. Because we need to make some modifications and apply them to our tasks and domains, and we found that they also served as our ablation studies, for consistency, we call SayCan “TWOSOME without finetuning” and GLAM “TWOSOME without action prompt normalization” or directly “TWOSOME without normalization”.
>
> **As for [1], it is actually a purely supervised-learning-based method**. Instead of explicitly calculating the probabilities of each action to form standard RL policies, they directly generate actions using LLMs.  The loss they show in Equation 1, is exactly the same as the autoregressive loss. They just formulate the loss into a policy-like format. There is no difference between autoregressive supervised learning and imitation learning if actions are the only output of LLMs. And the authors also do not claim they are an RL-based method. In their paper, they also rely on pre-collected high-quality expert datasets. We do not think that they directly compete with our method. We can first apply their methods to do offline training if provided with a high-quality dataset, and then use our method to do online training by interacting with the environments.
>
>
> ***
>
> Q3: The technical contribution of the paper is somewhat limited. Specifically, the idea of forming valid policy is proposed by previous works and LoRA for fine-tuning is a standard method. The review found the main new method is the proposed normalization approach. That to be said, the reviewer still considers the combination of existing methods might be valuable to our embodied AI community.
>
> A3: We appreciate that the reviewer acknowledges our contribution to the embodied AI community. Please refer to the first point of the above official comment, **About Technical Contribution**, for a clear explanation.
>
>
> ***
>
>
> Minor Q4:  In Eq (2), using ak in the summation is confusing.
>
> A4: We greatly thank the reviewer for catching the typo. We have corrected it in the latest version.

---

### Official Review · Reviewer_FDRX · 2023-11-09

**Soundness:** 3 good
**Presentation:** 3 good
**Contribution:** 3 good
**Rating:** 6
**Confidence:** 3

**Summary:**

- TWOSOME is a general online framework that allows LLMs to be used as decision-making agents in embodied environments.

- TWOSOME uses a two-stage approach to align LLMs with environments: first, it queries the joint probabilities of each valid action with LLMs to form behaviour policies; and second, it enhances the stability and robustness of the policies using normalisation methods and prompt design principles.

- TWOSOME exhibits significantly better sample efficiency and performance compared to other methods in both classical decision-making environment, Overcooked, and simulated household environment, VirtualHome.

- TWOSOME shows superior generalization ability to unseen tasks and does not result in a significant loss of the LLMs' original ability.

**Strengths:**

Strengths:

- Proposes a novel framework, TWOSOME, for aligning LLMs with embodied environments.

- TWOSOME is a general framework that can be applied to a wide range of decision-making tasks.

- TWOSOME exhibits significantly better sample efficiency and performance compared to other methods.

- TWOSOME shows superior generalization ability to unseen tasks.

- TWOSOME does not result in a significant loss of the LLMs' original ability.


Overall, the paper proposes a promising new framework for aligning LLMs with embodied environments. The framework is well-motivated and the experimental results are good.

**Weaknesses:**

Weaknesses:

- The paper does not provide a theoretical analysis of the proposed framework.

- The paper does not evaluate the performance of TWOSOME on a wider range of tasks and environments.

- The paper does not discuss the potential limitations of the proposed framework.

- The paper does not introduce anything novel, rather just combines existing components together to generalise to new tasks (apart from word normalisation which is fairly trivial).

- The paper does not compare to other LLM fine tuning baselines which generalise to new unseen tasks.

**Questions:**

-

---

> ### Author Response · Authors · 2023-11-15
>
> We thank the reviewers for their time and comments. Please see the above official comment to all the reviewers first and then our answers to your questions. We hope our answers will clear up the doubts about our work, and please let us know if there is any other clarification we can provide.
>
> ***
>
> Q1: The paper does not provide a theoretical analysis of the proposed framework.
>
> A1: Thanks for pointing it out. We indeed do not provide much theoretical analysis of the proposed framework because the whole framework is still basically under the traditional online actor-critic (PPO) framework, where the convergence of PPO should also apply. Though LLMs are usually transformer-based architecture, the last layer of LLMs is still an MLP layer to map the values generated by Transformer blocks to the probabilities of each token in the vocabulary.  We use the values to form the policies, which have no essential difference from the traditional RL agents, which usually have some CNN/MLP layers and use the output of the last MLP layer to form the policies. Finally, LoRA only updates part of the parameters, which may indeed prevent RL agents from learning the optimal policies. Empirically, LoRA shows superior performance not only in our work but also in tremendous CV/NLP works. We agree that it is non-trivial to provide a theoretical analysis of LoRA, especially in RL settings, but it is beyond the scope of this work.
>
> ***
>
> Q2: The paper does not evaluate the performance of TWOSOME on a wider range of tasks and environments.
>
> A2: Thanks for your valuable comments. We fully understand the reviewers’ wishes to see more results in different scenarios. Please refer to the second point of the above official comment, **About Evaluations on More Tasks and Environments** , for a clear explanation.
>
> ***
>
> Q3: The paper does not discuss the potential limitations of the proposed framework.
>
> A3: Thanks for pointing it out. Due to the limitation of pages, we do not provide a detailed analysis of the potential limitations. However, **we do mention our limitations in the final Discussion and Conclusion Section**, ‘However, our method suffers a major limitation that training a PPO agent from scratch is far faster and cheaper than finetuning an LLM. TWOSOME needs to feed all the valid actions to the LLMs for every action sampling, resulting in multiple times the amount of computation and a small batch size.’  Another limitation is that because we use an LLM as the backbone of our framework, we need to convert the observation into text and feed it into LLM, which inevitably causes information loss. The future work will be replacing the LLM with a VLM to directly use the image as input. We will put them in a separate section called Limitations and provide a more detailed discussion in the camera-ready version as there is more space provided.
>
>
> ***
>
> Q4: The paper does not introduce anything novel, rather just combines existing components together to generalise to new tasks (apart from word normalisation which is fairly trivial).
>
> A4: Thanks for your valuable comments. We kindly remind the reviewer that the contribution of the whole framework may be underestimated. There are some previous similar works like SayCan and GLAM, but we managed to polish and optimize every module in the framework, resulting in high parameter-efficiency and better generalization and sample-efficiency. Please refer to the first point of the above official comment, **About Technical Contribution**, for a clear explanation.
>
>
> ***
>
> Q5: The paper does not compare to other LLM fine tuning baselines which generalise to new unseen tasks.
>
> A5: Thanks for your valuable comments. To our best knowledge, only GLAM uses RL to train LLMs in embodied environments, which also serves as one of our main baselines. For the other supervised-learning-based methods, we do not think that they directly compete with our method. We can first apply their methods to do offline training if provided with a high-quality dataset, and then use our method to do online training by interacting with the environments. Please refer to the third point of the above official comment, **About Baselines Selection**, for a clear explanation.

---

### Author Response · Authors · 2023-11-15
**Response to All Reviewers (Part 1)**

We appreciate that all the reviewers agree that our paper is generally well-organized and easy to follow. Here we attempt to solve some common questions and confusions proposed by the reviewers.

**1. About Technical Contribution**
We appreciate that our word normalization is acknowledged as a unique contribution by all the reviewers. However, the contribution of the whole framework might be underestimated. Although there are some previous similar works like SayCan and GLAM, we managed to polish and optimize every module in the framework, resulting in high parameter-efficiency and better generalization and sample-efficiency. We are also grateful that some reviewers believe that these would be valuable to the embodied AI community.

**1) Parameter-efficient Fine-tuning Architecture**
1. As described in GLAM’s paper on Page 34, **GLAM requires 8 NVIDIA A100 80GB GPUs to train their framework with a small encoder-decoder LLM, Flan-T5 780M.** In Section 5.1, we also mentioned that our parameter-efficient framework allows us to run all the experiments using **a single A100 40G GPU with a 9X larger modern decoder-only LLM, LLaMA 7B**, with a similar length of the observation prompts (about 50 words) and larger action space (GLAM has 6 actions. TWOSOME has 6, 8, 10 and 17 actions in the four settings).  8 Nvidia A100 80GB GPUs is a relatively high bar that prevents most researchers and labs from conducting similar experiments. Our framework significantly lowers the bar with only one A100 40G GPU needed and uses the most popular LLM, LLaMA as our backbone, which would greatly encourage the community to conduct the following research.

2. It is worth noting that the parameter efficiency of our framework is not only brought by the LoRA but also by **our novel LLM-based actor-critic architecture**, where the actor and critic share the same LLM backbone but without any mutual influence on each other, while they are usually two independent models in classical RLHF settings. We leverage the feature of LoRA that it does not directly update the parameters of the backbone model and use the model without LoRA to serve as the backbone of the critic model. It saves the effort of maintaining two copies of LLMs in the memory and also guarantees that they have independent gradients to update, making the training process more stable and efficient. By only maintaining one LLM backbone, we successfully trained TWOSOME in one A100 40G GPU.

**2) General and Efficient Framework**
We appreciate that some reviewers think our proposed normalization methods are interesting and non-trivial. All the previous works, like SayCan and GLAM fail to observe and solve this potential critical issue. Though the implementation of the normalizations is straightforward, it requires careful observation of the token-level probabilities and pinpoints the issue. Empirical experiments show that our method can handle various lengths of action prompts and increase stability and sample efficiency. We successfully generalize the framework to various settings.


Moreover, GLAM does not fully utilize the capabilities of LLMs and fails to observe a critical advantage of the framework that besides the prior knowledge introduced by LLMs, our framework provides a practical and easy-to-implement way to modify the initialization of the policy and assign more probabilities to certain desirable actions by designing suitable prompts. This non-trivial feature has been pursued by RL agents for a long time.

Here is an example of the observation prompt GLAM used on Page 30. “You carry a blue ball, You see a wall 4 steps forward, You see a wall 2 steps left, You see a grey key 1 step right and 1 step forward, You see a red box 2 steps forward”. This kind of rough prompt design prevents them from achieving better performance.

We argue that the most appropriate application scenarios for TWOSOME should be the cases where agents are equipped with high-level(macro) actions like SayCan, instead of primitive actions, which usually have less meaningful semantics, resulting in underutilizing the capabilities of LLMs. That is also the reason why we focus on the macro-action setting.

***

---

> ### Author Response · Authors · 2023-11-15
> **Response to All Reviewers (Part 2)**
>
> **2. About Evaluations on More Tasks and Environments**
> Our methods do not rely on any assumptions or features of the environments and tasks, which are compatible with any standard RL decision-making settings. We have tried our best to select the most representative environments and tasks to show the generalization of our method, a classical sequential decision-making environment, Overcooked, and a simulated household environment, VirtualHome with more actions and items available.
>
> One of our baselines, GLAM, only tested their methods in a simple grid world environment, BabyAI with easy tasks, such as going somewhere or picking up something. **Our environments are much more complex and practical**. And our well-designed tasks are a combination of several of their tasks. For example, in our VirtureHome task, the agent needs to find chips and milk, get close to them, grab them, carry them to the living room, put them on the coffee table, turn on the TV, get close to the sofa, and sit on the sofa. We fully understand the reviewers’ wishes to see more results in different scenarios. There are also some previous impactful works[1][2], only evaluated their methods in Overcooked[1] and VirtualHome[2]. We believe our current environments and tasks are enough to show the robustness and generalization of our methods.
>
> Besides, we want to kindly remind the reviewers that **the computational consumption of our experiments might be underestimated**. We have three fine-tuning-based methods to compare, TWOSOME without normalization and with token/word normalization. Each method needs to be evaluated on four different tasks in the two environments. And according to RL standard, the performance of each method should be reported on the average of at least three seeds to show robustness. Thus, to obtain the results shown in Figure 5, we have to finetune at least 3X4X3=36 LLaMA-7B models without taking hyperparameter tuning into account. **It is difficult for RL-based methods to get evaluated in dozens of tasks like supervised-learning-based methods since they only need to train the model once with a pre-prepared high-quality expert dataset, while we need to collect and learn the noisy low-quality data from scratch.**
>
> ***
>
> **3. About Baselines Selection**
>
> We find that some reviewers may get confused about the baselines we use and miss the discussions in the extended related work in Appendix A. Due to the limitation of pages in the main paper, we had to put more discussions in the Appendix. We will move them to the main paper in the camera-ready version since more space will be provided.
>
> As introduced in Section 5.1, we compare the performance among five methods, “PPO (adapted from CleanRL (Huang et al., 2022a)), TWOSOME without finetuning (similar to SayCan (Brohan et al., 2023) with affordance value set to 1), TWOSOME without action prompt normalization (similar to GLAM (Carta et al., 2023) under decoder-only architecture and high-level actions) and TWOSOME with token normalization or word normalization.”
> **So we do adopt SayCan and GLAM as our baselines.** Because we need to make some modifications and apply them to our tasks and domains, and we found that they can also serve as our ablation studies, for consistency, we call SayCan “TWOSOME without finetuning” and GLAM “TWOSOME without action prompt normalization” or directly “TWOSOME without normalization”.
>
> As for the other supervised-learning-based methods, like [2] and [3]. They usually require pre-collected high-quality expert datasets. It is also the key difference between reinforcement learning and supervised learning. In our assumption, the agent does not have any specific prior knowledge of the environment and it needs to interact with the environment by itself, which is usually a more generalized and realistic way. Moreover, our method is not contradictory to the supervised learning methods like [2] and [3]. A high-quality dataset can always improve performance. We can first train the LLM with the provided dataset by supervised learning for offline training and then use our method for online training. Because LLMs are originally trained by supervised learning. It is easy to shift between supervised learning and reinforcement learning. It is another advantage of our method, compared to traditional RL methods and supervised-learning-based methods. **So these supervised-learning-based methods do not directly compete with our method.**
>
>
> [1] Strouse, D. J., et al. "Collaborating with humans without human data." Advances in Neural Information Processing Systems 34 (2021): 14502-14515.
> [2] Xiang, Jiannan, et al. "Language Models Meet World Models: Embodied Experiences Enhance Language Models." Advances in Neural Information Processing Systems 36 (2023).
> [3] Li, Shuang, et al. "Pre-trained language models for interactive decision-making." Advances in Neural Information Processing Systems 35 (2022): 31199-31212.

---

> ### Author Response · Authors · 2023-11-19
> **Response to All Reviewers (Part 3)**
>
> To all reviewers:
>
> We have added two new ablation studies in Appendix G.
>
> 1. In Appendix G.1  (on the last but one page of the paper), the new experiments show that our novel parameter-efficient architecture can not only dramatically save the GPU memory but also improve stability and accelerate convergence.
> 2. In Appendix G.2 (on the last page of the paper), the new experiments show that even without using the action masking in both VirtualHome tasks, our best method, TWOSOME with word normalization can still manage to learn the optimal policy, which shows the superior robustness and performance of our method and demonstrate that our method has the ability to deal with large action space.

---

### Meta-Review · Area_Chair_hZwK · 2023-12-09

**Metareview:**

This paper tackles the important problem of how to better ground LLMs as decision-making agents in embodied environments and trained through RL. Reviewers are generally in agreement that it’s a promising approach and makes a novel contribution through the proposed word normalization, yielding solid results. The main concerns that I find convincing related to whether this was too incremental of a contribution, as it mainly built on previous works such as LoRA and SayCan / GLAM, and whether they needed to evaluate on a wider range of tasks, beyond just Overcooked and VirtualHome. Reviewer fSyb also pointed out that the results on VirtualHome relied on masking out unrelated actions to reduce the complexity of the action space.

During the rebuttal period, the authors added additional experiments and responses that seemed to address many of the initial concerns, such as including an additional experiment without action masking. After discussion, reviewers all converged on borderline accept recommendations, but none seemed particularly enthusiastic about acceptance. For instance, reviewer HnZU commented “I am in favor of this work to presented at a conference but think the experimental results are not enough (due to budget constraints) to make it a ground-breaking paper” while fSyb continued to express a desire to see a more formal derivation for why the word normalization approach is the appropriate solution.

Overall I think this paper has great potential. I am inclined to accept, as the results seem pretty strong, and further it’s commendable that they were able to implement all experiments on a lower compute budget. However, I note that including the following would make for a much stronger paper: 1) additional environments and fine-tuning tasks, and 2) a more rigorous or formal proof for why word normalization is correct, which I urge the authors to consider for the camera-ready version.

**Justification For Why Not Higher Score:**

As indicated above, there are two main weaknesses that keep this paper from being a very strong accept: 1) additional environments and fine-tuning tasks, and 2) a more rigorous or formal proof for why word normalization is correct.

**Justification For Why Not Lower Score:**

Overall there was a consensus to accept.

---

### Decision · Program_Chairs · 2024-01-16

Accept (poster)